# Learning Curves for Noisy Heterogeneous Feature-Subsampled Ridge Ensembles

**Benjamin S. Ruben**[1],   **Cengiz Pehlevan**[2,3,4]

[1]Biophysics Graduate Program
[2]Center for Brain Science, [3]John A. Paulson School of Engineering and Applied Sciences, [4]Kempner
Institute for the Study of Natural and Artificial Intelligence,
Harvard University
Cambridge, MA 02138
benruben@g.harvard.edu, cpehlevan@seas.harvard.edu

## Abstract

Feature bagging is a well-established ensembling method which aims to reduce prediction variance by combining predictions of many estimators trained on subsets or projections of features. Here, we develop a theory of feature-bagging in noisy least-squares ridge ensembles and simplify the resulting learning curves in the special case of equicorrelated data. Using analytical learning curves, we demonstrate that subsampling shifts the double-descent peak of a linear predictor. This leads us to introduce heterogeneous feature ensembling, with estimators built on varying numbers of feature dimensions, as a computationally efficient method to mitigate double-descent. Then, we compare the performance of a feature-subsampling ensemble to a single linear predictor, describing a trade-off between noise amplification due to subsampling and noise reduction due to ensembling. Our qualitative insights carry over to linear classifiers applied to image classification tasks with realistic datasets constructed using a state-of-the-art deep learning feature map.

## 1   Introduction

Ensembling methods are ubiquitous in machine learning practice [1]. A class of ensembling methods (known as attribute bagging [2] or the random subspace method [3]) is based on feature subsampling [2–6], where predictors are independently trained on subsets of the features, and their predictions are combined to achieve a stronger prediction. The random forest method is a popular example [3, 7].

While commonly used in practice, a theoretical understanding of ensembling via feature subsampling is not well developed. Here, we provide an analysis of this technique in the linear ridge regression setting. Using methods from statistical physics [8–12], we obtain analytical expressions for typical-case generalization error in linear ridge ensembles (proposition 1), and simplify these expressions in the special case of equicorrelated data with isotropic feature noise (proposition 2). The result provides a powerful tool to quickly probe the generalization error of ensembled regression under a rich set of conditions. In section 3, we study the behavior of a single feature-subsampling regression model. We observe that subsampling shifts the location of a predictor's sample-wise double-descent peak [13–15]. This motivates section 4, where we study ensembles built on predictors which are heterogeneous in the number of features they access, as a method to mitigate double-descent. We demonstrate this method's efficacy in a realistic image classification task. In section 5 we apply our theory to the trade-off between ensembling and subsampling in resource-constrained settings. We

characterize how a variety of factors influence the optimal ensembling strategy, finding a particular significance to the level of noise in the predictions made by ensemble members.

In summary, we make the following contributions:

- Using the replica trick from statistical physics [8, 11], we derive the generalization error of ensembled least-squares ridge regression in a general setting, and simplify the resulting expressions in the tractable special case where features are equicorrelated.
- We demonstrate benefits of heterogeneous ensembling as a robust and computationally efficient regularizer for mitigating double-descent with analytical theory and in a realistic image classification task.
- We describe the ensembling-subsampling trade-off in resource-constrained settings, and characterize the effect of label noise, feature noise, readout noise, regularization, sample size and task structure on the optimal ensembling strategy.

**Related works:** A substantial body of work has elucidated the behavior of linear predictors for a variety of feature maps [14, 16–30]. Several recent works have extended this research to characterize the behavior of ensembled regression using solvable models [24, 31–33]. Additional recent works study the performance of ridge ensembles with example-wise subsampling [34, 35] and simultaneous subsampling of features and examples [32], finding that subsampling behaves as an implicit regularization. Methods from statistical physics have long been used for machine learning theory [10–12, 26, 27, 30, 36, 37]. Relevant work in this domain include [38] which studied ensembling by data-subsampling in linear regression.

# 2   Learning Curves for Ensembled Ridge Regression

We consider noisy ensembled ridge regression in the setting where ensemble members are trained independently on masked versions of the available features. We derive our main analytical formula for generalization error of ensembled linear regression, as well as analytical expressions for generalization error in the special case of equicorrelated features with isotropic noise.

## 2.1   Problem Setup

Consider a training set $\mathcal{D} = \{\bar{\psi}^\mu, y^\mu\}_{\mu=1}^P$ of size $P$. The training examples $\bar{\psi}^\mu \in \mathbb{R}^M$ are drawn from a Gaussian distribution with Gaussian feature noise: $\bar{\psi}^\mu = \psi^\mu + \sigma^\mu$, where $\psi^\mu \sim \mathcal{N}(0, \Sigma_s)$ and $\sigma^\mu \sim \mathcal{N}(0, \Sigma_0)$. Data and noise are drawn i.i.d. so that $\mathbb{E}\left[\psi^\mu \psi^{\nu\top}\right] = \delta_{\mu\nu} \Sigma_s$ and $\mathbb{E}\left[\sigma^\mu \sigma^{\nu\top}\right] = \delta_{\mu\nu} \Sigma_0$. Labels are generated from a noisy teacher function $y^\mu = \frac{1}{\sqrt{M}} w^{*\top} \psi^\mu + \epsilon^\mu$ where $\epsilon^\mu \sim \mathcal{N}(0, \zeta^2)$. Label noises are drawn i.i.d. so that $\mathbb{E}[\epsilon^\mu \epsilon^\nu] = \delta_{\mu\nu} \zeta^2$.

We seek to analyze the quality of predictions which are averaged over an ensemble of ridge regression models, each with access to a subset of the features. We consider $k$ linear predictors with weights $\hat{w}_r \in \mathbb{R}^{N_r}$, $r = 1, \dots, k$. Critically, we allow $N_r \neq N_{r'}$ for $r \neq r'$, which allows us to introduce *structural* heterogeneity into the ensemble of predictors. A forward pass of the model is given as:

$$f(\psi) = \frac{1}{k} \sum_{r=1}^k f_r(\psi), \qquad f_r(\psi) = \frac{1}{\sqrt{N_r}} \hat{w}_r^\top A_r (\psi + \sigma) + \xi_r. \tag{1}$$

The model's prediction $f(\psi)$ is an average over $k$ linear predictors. The "measurement matrices" $A_r \in \mathbb{R}^{N_r \times M}$ act as linear masks restricting the information about the features available to each member of the ensemble. Subsampling may be implemented by choosing the rows of each $A_r$ to coincide with the rows of the identity matrix – the row indices corresponding to indices of the sampled features. The feature noise $\sigma \sim \mathcal{N}(0, \Sigma_0)$ and the readout noises $\xi_r \sim \mathcal{N}(0, \eta_r^2)$, are drawn independently at the execution of each forward pass of the model. Note that while the feature noise is shared across the ensemble, readout noise is drawn independently for each readout: $\mathbb{E}[\xi_r \xi_{r'}] = \delta_{rr'} \eta_r^2$.

The weight vectors are trained separately in order to minimize an ordinary least-squares loss function with ridge regularization:

$$\hat{\boldsymbol{w}}_r = \arg\min_{\boldsymbol{w}_r \in \mathbb{R}^{N_r}} \left[ \sum_{\mu=1}^{P} \left( \frac{1}{\sqrt{N_r}} \boldsymbol{w}_r^\top \boldsymbol{A}_r \bar{\boldsymbol{\psi}}^\mu + \xi_r^\mu - y^\mu \right)^2 + \lambda_r |\boldsymbol{w}_r^2| \right] \tag{2}$$

Here $\{\xi_r^\mu\}$ represents the readout noise which is present during training, and independently drawn: $\xi_r^\mu \sim \mathcal{N}(0, \eta_r^2)$, $\mathbb{E}[\xi_r^\mu \xi_r^\nu] = \eta_r^2 \delta_{\mu\nu}$. As a measure of model performance, we consider the generalization error, given by the mean-squared-error (MSE) on ensemble-averaged prediction:

$$E_g(\mathcal{D}) = \mathbb{E}_{\psi, \sigma, \{\xi_r\}} \left[ \left( f(\boldsymbol{\psi}) - \frac{1}{\sqrt{M}} \boldsymbol{w}^{*\top} \boldsymbol{\psi} \right)^2 \right] \tag{3}$$

Here, the expectation is over the data distribution and noise: $\boldsymbol{\psi} \sim \mathcal{N}(0, \boldsymbol{\Sigma}_s)$, $\boldsymbol{\sigma} \sim \mathcal{N}(0, \boldsymbol{\Sigma}_0)$, $\xi_r \sim \mathcal{N}(0, \eta_r^2)$. The generalization error depends on the particular realization of the dataset $\mathcal{D}$ through the learned weights $\{\hat{\boldsymbol{w}}^*\}$. We may decompose the generalization error as follows:

$$E_g(\mathcal{D}) = \frac{1}{k^2} \sum_{r,r'=1}^{k} E_{rr'}(\mathcal{D}) \tag{4}$$

$$E_{rr'}(\mathcal{D}) \equiv \frac{1}{M} \left[ \left( \frac{1}{\sqrt{\nu_{rr}}} \boldsymbol{A}_r^\top \hat{\boldsymbol{w}}_r - \boldsymbol{w}^* \right)^\top \boldsymbol{\Sigma}_s \left( \frac{1}{\sqrt{\nu_{r'r'}}} \boldsymbol{A}_{r'}^\top \hat{\boldsymbol{w}}_{r'} - \boldsymbol{w}^* \right) \right.$$
$$\left. + \frac{1}{\sqrt{\nu_{rr}\nu_{r'r'}}} \hat{\boldsymbol{w}}_r^\top \boldsymbol{A}_r \boldsymbol{\Sigma}_0 \boldsymbol{A}_{r'}^\top \hat{\boldsymbol{w}}_{r'} + M\delta_{rr'}\eta_r^2 \right] \tag{5}$$

Computing the generalization error of the model is then a matter of calculating $E_{rr'}$ in the cases where $r = r'$ and $r \neq r'$. In the asymptotic limit we consider, we expect that the generalization error concentrates over randomly drawn datasets $\mathcal{D}$.

## 2.2 Main Result

We calculate the generalization error using the replica trick from statistical physics, and present the calculation in Appendix F. The result of our calculation is stated in proposition 1.

**Proposition 1.** *Consider the ensembled ridge regression problem described in Section 2.1. Consider the asymptotic limit where $M, P, \{N_r\} \to \infty$ while the ratios $\alpha = \frac{P}{M}$ and $\nu_{rr} = \frac{N_r}{M}$, $r = 1, \ldots, k$ remain fixed. Define the following quantities:*

$$\tilde{\boldsymbol{\Sigma}}_{rr'} \equiv \frac{1}{\sqrt{\nu_{rr}\nu_{r'r'}}} \boldsymbol{A}_r [\boldsymbol{\Sigma}_s + \boldsymbol{\Sigma}_0] \boldsymbol{A}_{r'}^\top \tag{6}$$

$$\boldsymbol{G}_r \equiv \boldsymbol{I}_{N_r} + \hat{q}_r \tilde{\boldsymbol{\Sigma}}_{rr} \tag{7}$$

$$\gamma_{rr'} \equiv \frac{\alpha}{M(\lambda_r + q_r)(\lambda_{r'} + q_{r'})} \operatorname{tr}\left[ \boldsymbol{G}_r^{-1} \tilde{\boldsymbol{\Sigma}}_{rr'} \boldsymbol{G}_{r'}^{-1} \tilde{\boldsymbol{\Sigma}}_{r'r} \right] \tag{8}$$

*Then the terms of the average generalization error (eq. 5) may be written as:*

$$\langle E_{rr'}(\mathcal{D}) \rangle_{\mathcal{D}} = \frac{\gamma_{rr'}\zeta^2 + \delta_{rr'}\eta_r^2}{1 - \gamma_{rr'}} + \frac{1}{1 - \gamma_{rr'}} \left( \frac{1}{M} \boldsymbol{w}^{*\top} \boldsymbol{\Sigma}_s \boldsymbol{w}^* \right)$$
$$- \frac{1}{M(1 - \gamma_{rr'})} \boldsymbol{w}^{*\top} \boldsymbol{\Sigma}_s \left[ \frac{1}{\nu_{rr}} \hat{q}_r \boldsymbol{A}_r^\top \boldsymbol{G}_r^{-1} \boldsymbol{A}_r + \frac{1}{\nu_{r'r'}} \hat{q}_{r'} \boldsymbol{A}_{r'}^\top \boldsymbol{G}_{r'}^{-1} \boldsymbol{A}_{r'} \right] \boldsymbol{\Sigma}_s \boldsymbol{w}^* \tag{9}$$
$$+ \frac{\hat{q}_r \hat{q}_{r'}}{M(1 - \gamma_{rr'})} \frac{1}{\sqrt{\nu_{rr}\nu_{r'r'}}} \boldsymbol{w}^{*\top} \boldsymbol{\Sigma}_s \boldsymbol{A}_r^\top \boldsymbol{G}_r^{-1} \tilde{\boldsymbol{\Sigma}}_{rr'} \boldsymbol{G}_{r'}^{-1} \boldsymbol{A}_{r'} \boldsymbol{\Sigma}_s \boldsymbol{w}^*$$

*where the pairs of order parameters $\{q_r, \hat{q}_r\}$ for $r = 1, \ldots, K$, satisfy the following self-consistent saddle-point equations*

$$\hat{q}_r = \frac{\alpha}{\lambda_r + q_r}, \qquad q_r = \frac{1}{M} \operatorname{tr}\left[ \boldsymbol{G}_r^{-1} \tilde{\boldsymbol{\Sigma}}_{rr} \right]. \tag{10}$$

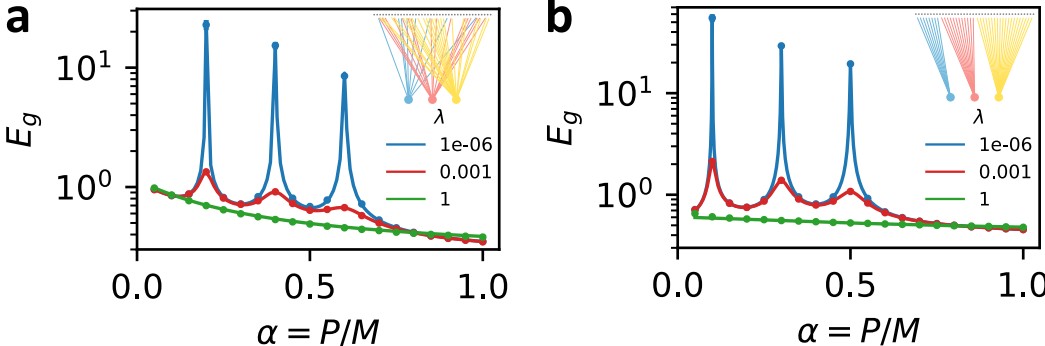

Figure 1: Comparison of numerical and theoretical learning curves for ensembled linear regression. Circles represent numerical results averaged over 100 trials; lines indicate theoretical predictions. Error bars represent the standard error of the mean but are often smaller than the markers. (a) Testing of proposition 1 with $M = 2000$, $[\boldsymbol{\Sigma}_s]_{ij} = .8^{|i-j|}$, $[\boldsymbol{\Sigma}_0]_{ij} = \frac{1}{10}(0.3)^{|i-j|}$, $\zeta = 0.1$, and all $\eta_r = 0.2$ and $\lambda_r = \lambda$ (see legend). $k = 3$ linear predictors access fixed, randomly selected (with replacement) subsets of the features with fractional sizes $\nu_{rr} = 0.2, 0.4, 0.6$. Fixed ground-truth weights $\boldsymbol{w}^*$ are drawn from an isotropic Gaussian distribution. (b) Testing of proposition 2 with $M = 5000$, $s = 1$, $c = 0.6$, $\omega^2 = 0.1$, $\zeta = 0.1$, all $\eta_r = 0.1$, and all $\lambda_r = \lambda$ (see legend). Ground truth weights sampled as in eq. 11 with $\rho = 0.3$. Feature subsets accessed by each readout are mutually exclusive (inset) with fractional sizes $\nu_{rr} = 0.1, 0.3, 0.5$.

*Proof.* We calculate the terms in the generalization error using the replica trick, a standard but non-rigorous method from the statistical physics of disordered systems. The full derivation may be found in the Appendix F. When the matrices $\boldsymbol{A}_r (\boldsymbol{\Sigma}_s + \boldsymbol{\Sigma}_0) \boldsymbol{A}_r^\top$, $r = 1, \ldots, k$ have bounded spectra, this result may be obtained by extending the results of [31] to include readout noise, as shown in Appendix G. □

We make the following remarks:

*Remark* 1. Implicit in this theorem is the assumption that the relevant matrix contractions and traces which appear in the generalization error (eq. 9) and the surrounding definitions tend to a well-defined limit which remains $\mathcal{O}(1)$ as $M \to \infty$.

*Remark* 2. This result applies for any (well-behaved) linear masks $\{\boldsymbol{A}_r\}$. We will focus on the case where each $\boldsymbol{A}_r$ implements subsampling of an extensive fraction $\nu_{rr}$ of the features.

*Remark* 3. When $k = 1$, our result reduces to the generalization error of a single ridge regression model, as studied in refs. [36, 39].

*Remark* 4. We include "readout noise" which independently corrupts the predictions of each ensemble member. This models sources of variation between ensemble members not otherwise accounted for. For example, ensembles of deep networks will vary due to random initialization of parameters [24, 31, 36]. Readout noise is more directly present in physical neural networks, such as an analog neural networks [40] or biological neural circuits[41] due to their inherent stochasticity.

In Figure 1a, we confirm the result of the general calculation by comparing with numerical experiments using a synthetic dataset with $M = 2000$ highly structured features (see caption for details). $k = 3$ readouts see random, fixed subsets of features. Theory curves are calculated by solving the fixed-point equations 10 numerically for the chosen $\boldsymbol{\Sigma}_s$, $\boldsymbol{\Sigma}_0$ and $\{\boldsymbol{A}_r\}_{r=1}^k$ then evaluating eq. 9.

## 2.3 Equicorrelated Data

Our general result allows the freedom to tune many important parameters of the learning problem: the correlation structure of the dataset, the number of ensemble members, the scales of noise, etc. However, the derived expressions are rather opaque. In order to better understand the phenomena captured by these expressions, we examine the following special case:

**Proposition 2.** *In the setting of section 2.1 and proposition 1, consider the following special case:*

$$\boldsymbol{w}^* = \sqrt{1-\rho^2}\mathbb{P}_\perp \boldsymbol{w}_0^* + \rho \mathbf{1}_M \tag{11}$$

$$\boldsymbol{w}_0^* \sim \mathcal{N}(0, \boldsymbol{I}_M) \tag{12}$$

$$\boldsymbol{\Sigma}_s = s\left[(1-c)\boldsymbol{I}_M + c\mathbf{1}_M\mathbf{1}_M^\top\right] \tag{13}$$

$$\boldsymbol{\Sigma}_0 = \omega^2 \boldsymbol{I}_M \tag{14}$$

*with $c \in [0,1], \rho \in [-1,1]$. Label and readout noises $\zeta, \eta_r \geq 0$ are permitted. Here $\mathbb{P}_\perp = \boldsymbol{I}_M - \frac{1}{M}\mathbf{1}_M\mathbf{1}_M^\top$ is a projection matrix which removes the component of $\boldsymbol{w}_0^*$ which is parallel to $\mathbf{1}_M$. The matrices $\{\boldsymbol{A}_r\}_{r=1}^k$ have rows consisting of distinct one-hot vectors so that each of the $k$ readouts has access to a subset of $N_r = \nu_{rr}M$ features. For $r \neq r'$, denote by $n_{rr'}$ the number of neurons sampled by both $\boldsymbol{A}_r$ and $\boldsymbol{A}_{r'}$ and let $\nu_{rr'} \equiv n_{rr'}/M$ remain fixed as $M \to \infty$.*

*Define the following quantities:*

$$a \equiv s(1-c) + \omega^2 \qquad S_r \equiv \frac{\hat{q}_r}{\nu_{rr} + a\hat{q}_r}, \qquad \gamma_{rr'} \equiv \frac{a^2 \nu_{rr'} S_r S_{r'}}{\alpha} \tag{15}$$

*The terms of the decomposed generalization error may then be written:*

$$\langle E_{rr'}\rangle_{\mathcal{D},\boldsymbol{w}_0^*} = \frac{1}{1-\gamma_{rr'}}\left((1-\rho^2)I_{rr'}^0 + \rho^2 I_{rr'}^1\right) + \frac{\gamma_{rr'}\zeta^2 + \delta_{rr'}\eta_r^2}{1-\gamma_{rr'}} \tag{16}$$

*where we have defined*

$$I_{rr'}^0 \equiv s(1-c)\left(1 - s(1-c)\nu_{rr}S_r - s(1-c)\nu_{r'r'}S_{r'} + as(1-c)\nu_{rr'}S_r S_{r'}\right) \tag{17}$$

$$I_{rr'}^1 \equiv \begin{cases} \frac{s(1-c)(\nu_{rr'}-\nu_{rr}\nu_{r'r'})+\omega^2\nu_{rr'}}{\nu_{rr}\nu_{r'r'}} & \text{if } 0 < c \leq 1 \\ I_{rr'}^0 & \text{if } c = 0 \end{cases} \tag{18}$$

*and where $\{q_r, \hat{q}_r\}$ may be obtained analytically as the solution (with $q_r > 0$) to:*

$$q_r = \frac{a\nu_{rr}}{\nu_{rr} + a\hat{q}_r} \qquad , \qquad \hat{q}_r = \frac{\alpha}{\lambda_r + q_r} \tag{19}$$

*In the "ridgeless" limit where all $\lambda_r \to 0$, we may make the following simplifications:*

$$S_r \to \frac{2\alpha}{a\left(\alpha + \nu_{rr} + |\alpha - \nu_{rr}|\right)}, \quad \gamma_{rr'} \to \frac{4\alpha\nu_{rr'}}{\left(\alpha + \nu_{rr} + |\alpha - \nu_{rr}|\right)\left(\alpha + \nu_{r'r'} + |\alpha - \nu_{r'r'}|\right)} \tag{20}$$

*Proof.* Simplifying the fixed-point equations and generalization error formulas in this special case is an exercise in linear algebra. The main tools used are the Sherman-Morrison formula [42] and the fact that the data distribution is isotropic in the features so that the form of $\tilde{\boldsymbol{\Sigma}}_{rr}$ and $\tilde{\boldsymbol{\Sigma}}_{rr'}$ depend only on the subsampling and overlap fractions $\nu_{rr}, \nu_{r'r'}, \nu_{rr'}$. To aid in computing the necessary matrix contractions we developed a custom Mathematica package which handles block matrices of symbolic dimension, with blocks containing matrices of the form $\boldsymbol{M} = c_1 \boldsymbol{I} + c_2 \mathbf{1}\mathbf{1}^\top$. This package and the Mathematica notebook used to derive these results are available online (see Appendix B)  □

In this tractable special case, $c \in [0, 1]$ is a parameter which tunes the strength of correlations between features of the data. When $c = 0$, the features are independent, and when $c = 1$ the features are always equivalent. $s$ sets the overall scale of the features and the "Data-Task alignment" $\rho$ tunes the alignment of the ground truth weights with the special direction in the covariance matrix (analogous to "task-model" alignment [14, 27]). A table of parameters is provided in Appendix A. In Figure 1b, we test these results by comparing the theoretical expressions for generalization error with the results of numerical experiments, finding perfect agreement.

With an analytical formula for the generalization error, we can compute the optimal regularization parameters $\lambda_r$ which minimize the generalization error. These may, in general, depend on both $r$ and the sample size $\alpha$. Rather than minimizing the error of the ensemble, we may minimize the generalization error of predictions made by the ensemble members independently. We find that this "locally optimal" regularization, denoted $\lambda^*$, is independent of $\alpha$, generalizing results from [14, 43] to correlated data distributions (see Appendix H.3).

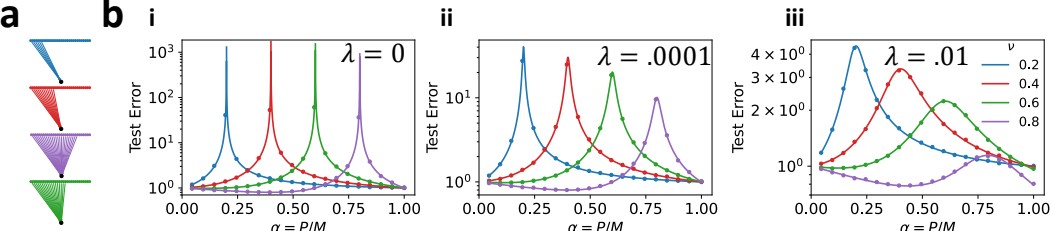

Figure 2: Subsampling alters the location of the double-descent peak of a linear predictor. (a) Illustrations of subsampled linear predictors with varying subsampling fraction $\nu$. (b) Comparison between experiment and theory for subsampling linear regression on equicorrelated datasets. We choose task parameters as in proposition 2 with $c = \omega = \zeta = \eta = 0$, $s = 1$, and (i) $\lambda = 0$, (ii) $\lambda = 10^{-4}$, (iii) $\lambda = 10^{-2}$. All learning curves are for a single linear predictor $k = 1$ with subsampling fraction $\nu$ shown in legend. Circles show results of numerical experiment. Lines are analytical prediction.

# 3 Subsampling shifts the double-descent peak of a linear predictor

Consider a single linear regressor ($k = 1$) which connects to a subset of $N = \nu M$ features in the equicorrelated data setting of proposition 2. Also setting $c = 0$, $s = 1$, and $\eta_r = \omega = 0$ and taking the limit $\lambda \to 0$ the generalization error reads:

$$\langle E_g \rangle_{\mathcal{D}, \boldsymbol{w}^*} = \left\{ \begin{array}{ll} \frac{\nu}{\nu - \alpha} \left[ (1 - \nu) + \frac{1}{\nu}(\alpha - \nu)^2 \right] + \frac{\alpha}{\nu - \alpha} \zeta^2, & \text{if } \alpha < \nu \\ \frac{\alpha}{\alpha - \nu} [1 - \nu] + \frac{\nu}{\alpha - \nu} \zeta^2, & \text{if } \alpha > \nu \end{array} \right\} \tag{21}$$

We thus see that double descent can arise from two possible sources of variance: explicit label noise (if $\zeta > 0$) or implicit label noise induced by feature subsampling ($\nu < 1$). As $E_g \sim (\alpha - \nu)^{-1}$, generalization error diverges when sample size is equal to the number of sampled features. Intuitively, this occurs because subsampling changes the number of parameters of the regression model, and thus its interpolation threshold. To demonstrate this, we plot the learning curves for subsampled linear regression on equicorrelated data in Figure 2. At small finite ridge the test error no longer diverges when $\alpha = \nu$, but still displays a distinctive peak.

# 4 Heterogeneous connectivity mitigates double-descent

Double-descent – over-fitting to noise in the training set near a model's interpolation threshold – poses a serious risk in practical machine-learning applications [22]. Cross-validating the regularization strength against the training set is the canonical approach to avoiding double-descent [17, 43], but in practice requires a computationally expensive parameter sweep and prior knowledge of the task. In situations where computational resources are limited or hyperparameters are fixed prior to specification of the task, it is natural to seek an alternative solution. Considering again the plots in Figure 2(b), we observe that at any value of $\alpha$, the double-descent peak can be avoided with an acceptable choice of the subsampling fraction $\nu$. This suggests another strategy to mitigate double descent: heterogeneous ensembling. Ensembling over predictors with a heterogeneous distribution of interpolation thresholds, we may expect that when one predictor fails due to over-fitting, the other members of the ensemble compensate with accurate predictions.

In Figure 3, we show that heterogeneous ensembling can guard against double-descent. We define two ensembling strategies: in homogeneous ensembling, each of $k$ readouts connects a fraction $\nu_{rr} = 1/k$ features. In heterogeneous ensembling, the number of features connected by each of the $k$ readouts are drawn from a Gamma distribution $\Gamma_{k,\sigma}(\nu)$ with mean $1/k$ and standard deviation $\sigma$ (see Fig. 3b) then re-scaled to sum to 1 (see Appendix C for details). All feature subsets are mutually exclusive ($\nu_{rr'} = 0$ for $r \neq r'$). Homogeneous and heterogeneous ensembling are illustrated for $k = 10$ in Figs. 3 a.i and 3 a.ii respectively. We test this hypothesis using eq. 16 in 3c. At small regularization

($\lambda = .001$), we find that heterogeneity of the distribution of subsampling fractions ($\sigma > 0$) lowers the double-descent peak of an ensemble of linear predictors, while at larger regularization ($\lambda = 0.1$), there is little difference between homogeneous and heterogeneous learning curves. The asymptotic ($\alpha \to \infty$) error is unaffected by the presence of heterogeneity in the degrees of connectivity, which can be seen as the coincidence of the triangular markers in Fig. 3c, as well as from the $\alpha \to \infty$ limit of eq. 16 (see Appendix H.5). Fig. 3c also shows the learning curve of a single linear predictor with no feature subsampling and optimal regularization. We see that the feature-subsampling ensemble appreciably outperforms the fully-connected model when $c = 0.8$ and $\eta = 0.5$, suggesting the important roles of data correlations and readout noise in determining the optimal readout strategy. These roles are further explored in section 5 and fig 4.

We also test the effect of heterogeneous ensembling in the a realistic classification task. Specifically, we train ensembles of linear classifiers to predict the labels of imagenet [44] images corresponding to 10 different dog breeds (the "Imagewoof" task [45]) from their top-hidden-layer representations in a pre-trained ResNext deep network [46] (see Appendix E for details). We characterize the statistics of the resulting $M = 2048$-dimensional feature set in Fig. S1. This "ResNext-Imagewoof" classification task has multiple features which make it amenable to learning with a feature-subsampling ensemble. First, the ResNext features have a high degree of redundancy [47], allowing classification to be performed accurately using only a fraction of the available features (see Fig. 3d and S1c). Second, when classifications of multiple predictors are combined by a majority vote, there is a natural upper bound on the influence of a single erring ensemble member (unlike in regression where predictions can diverge). Calculating learning curves for the imagewoof classification task using homogeneous ensembles, we see sharp double-descent peaks in an ensemble of size $k$ when $P = M/k$ (Fig. 3e.i). Using a heterogeneous ensemble mitigates this catastrophic over-fitting, leading to monotonically decreasing error without regularization (Fig. 3e.ii). A single linear predictor with a tuned regularization of $\lambda = 0.1$ performs only marginally better than the heterogeneous feature-subsampling ensemble with $k = 16$ or $k = 32$. This suggests heterogeneous ensembling can be an effective alternative to regularization in real-world classification tasks using pre-trained deep learning feature maps.

Note that training a feature-subsampling ensemble also benefits from improved computational complexity. Training an estimator of dimension $N_r$ involves, in the worst case, inverting an $N_r \times N_r$ matrix, which requires $\mathcal{O}(N_r^3)$ operations. Setting $N_r = M/k$, we see that the number of operations required to train an ensemble of $k$ predictors scales as $\mathcal{O}(k^{-2})$.

# 5 Correlations, Noise, and Task Structure Dictate the Ensembling-Subsampling Trade-off

In resource-constrained settings, one must decide between training a single large predictor or an ensemble of smaller predictors. When the number of weights is constrained, ensembling may benefit generalization by averaging over multiple predictions, but at the expense of each prediction incorporating fewer features. Intuitively, the presence of correlations between features limits the penalty incurred by subsampling, as measurements from a subset of features will also confer information about the unsampled features. The equicorrelated data model of proposition 2 permits a solvable toy model for these competing effects. We consider the special case of ensembling over $k$ readouts, each connecting the same fraction $\nu_{rr} = \nu = 1/k$ of all features. For simplicity, we set $\nu_{rr'} = 0$ for $r \neq r'$. We asses the learning curves of this toy model in both the ridgeless limit $\lambda \to 0$ where double-descent has a large effect on test error, and at 'locally optimal' regularization $\lambda = \lambda^*$ for which double-descent is eliminated. In these special cases, one can write the generalization error in the following forms (see Appendix H.4 for derivation):

$$E_g(k, s, c, \eta, \omega, \zeta, \rho, \alpha, \lambda = 0) = s(1 - c)\mathcal{E}(k, \rho, \alpha, H, W, Z) \tag{22}$$
$$E_g(k, s, c, \eta, \omega, \zeta, \rho, \alpha, \lambda = \lambda^*) = s(1 - c)\mathcal{E}^*(k, \rho, \alpha, H, W, Z) \tag{23}$$

where we have defined the effective noise-to-signal ratios:

$$H \equiv \frac{\eta^2}{s(1 - c)}, \quad W = \frac{\omega^2}{s(1 - c)}, \quad Z = \frac{\zeta^2}{s(1 - c)} \tag{24}$$

Therefore, given fixed parameters $s, c, \rho, \alpha$, the value $k^*$ which minimizes error depends on the noise scales, $s$, and $c$ only through the ratios $H, W$ and $Z$:

$$k^*_{\lambda=0}(H, W, Z, \rho, \alpha) \equiv \arg\min_{k \in \mathbb{N}} E_g(k) = \arg\min_{k \in \mathbb{N}} \mathcal{E}(k, \rho, \alpha, H, W, Z) \qquad (25)$$

$$k^*_{\lambda=\lambda^*}(H, W, Z, \rho, \alpha) \equiv \arg\min_{k \in \mathbb{N}} E_g(k) = \arg\min_{k \in \mathbb{N}} \mathcal{E}^*(k, \rho, \alpha, H, W, Z) \qquad (26)$$

In Fig. 4a, we plot these reduced errors curves $\mathcal{E}$, $\mathcal{E}^*$ as a function of $\alpha$ for varying ensemble sizes $k$ and reduced readout noise scales $H$. At zero regularization learning curves diverge at their interpolation threshold. At locally optimal regularization $\lambda = \lambda^*$, learning curves decrease monotonically with sample size. Increasing readout noise $H$ raises generalization error more sharply for smaller $k$. In Fig. 4b we plot the optimal $k^*$ in various two-dimensional slices of parameter space in which $\rho$ is fixed and $W = Z = 0$ while $\alpha$ and $H$ vary. The resulting phase diagrams may be divided into three regions. In the signal-dominated phase a single fully-connected readout is optimal ($k^* = 1$). In an intermediate phase, $1 < k^* < \infty$ minimizes error. And in a noise-dominated phase $k^* = \infty$. At zero regularization, we have determined an analytical expression for the boundary between the intermediate and noise-dominated phases (see Appendix H.4.1 and dotted lines in Figs 4.b,c,d). The signal-dominated, intermediate, and noise-dominated phases persist when $\lambda = \lambda^*$, removing the effects of double descent. In all panels, an increase in H causes an increase in $k^*$. This can occur because of a decrease in the signal-to-readout noise ratio $s/\eta^2$, or through an increase in the correlation strength $c$. An increase in $\rho$ also leads to an increase in $k^*$, indicating that ensembling is more effective for easier tasks. Figs 4c,d show analogous phase diagrams where $W$ or $Z$ are varied. Signal-dominated, intermediate, and noise-dominated regimes are visible in the resulting phase diagrams at zero regularization. However, when optimal regularization is used, $k^* = 1$ is always optimal. The presence of regions where $k^* > 1$ can thus be attributed to double-descent at sub-optimal regularization or to the presence of readout noise which is independent across predictors. We chart the parameter-space of the reduced errors and optimal ensemble size $k^*$ extensively in Appendix I. We plot learning curves for the "ResNext-Imagewoof" ensembled linear classification task with varying strength of readout noise in Fig. 4e, and phase diagrams of optimal ensemble size $k$ in Fig. 4f, finding similar behavior to the toy model. See Figs. S3, S4, S5 and Appendix E.4.3 for further discussion.

# 6 Conclusion

In this paper, we provided a theory of feature-subsampled linear ridge regression. We identified the special case in which features of the data are "equicorrelated" as a minimal toy model to explore the combined effects of subsampling, ensembling, and different types of noise on generalization error. The resulting learning curves displayed two potentially useful phenomena.

First, we demonstrated that heterogeneous ensembling can mitigate over-fitting, reducing or eliminating the double-descent peak of an under-regularized model. In most machine learning applications, the size of the dataset is known at the outset and suitable regularization may be determined to mitigate double descent, either by selecting a highly over-parameterized model [22] or by cross-validation techniques (see for example [17]). However, in contexts where a single network architecture is designed for an unknown task or a variety of tasks with varying structure and noise levels, heterogeneous ensembling may be used to smooth out the perils of double-descent.

Next, we described a trade-off between noise reduction due to ensembling and noise amplification due to subsampling in a resource-constrained setting where the total number of weights is fixed. Our analysis suggests that ensembling is particularly useful in neural networks with an inherent noise. Physical neural networks, such as analog neural networks[40] and biological neural circuits [41] present such a resource-constrained environments where intrinsic noise is a significant issue.

Much work remains to achieve a full understanding of the interactions between data correlations, readout noise, and ensembling. In this work, we have given a thorough treatment of the convenient special case where features are equicorrelated. Future work should analyze subsampling and ensembling for codes with realistic correlation structure, such as the power-law spectra (see Fig. S1) [27, 30, 48, 49] and sparse activation patterns [50].

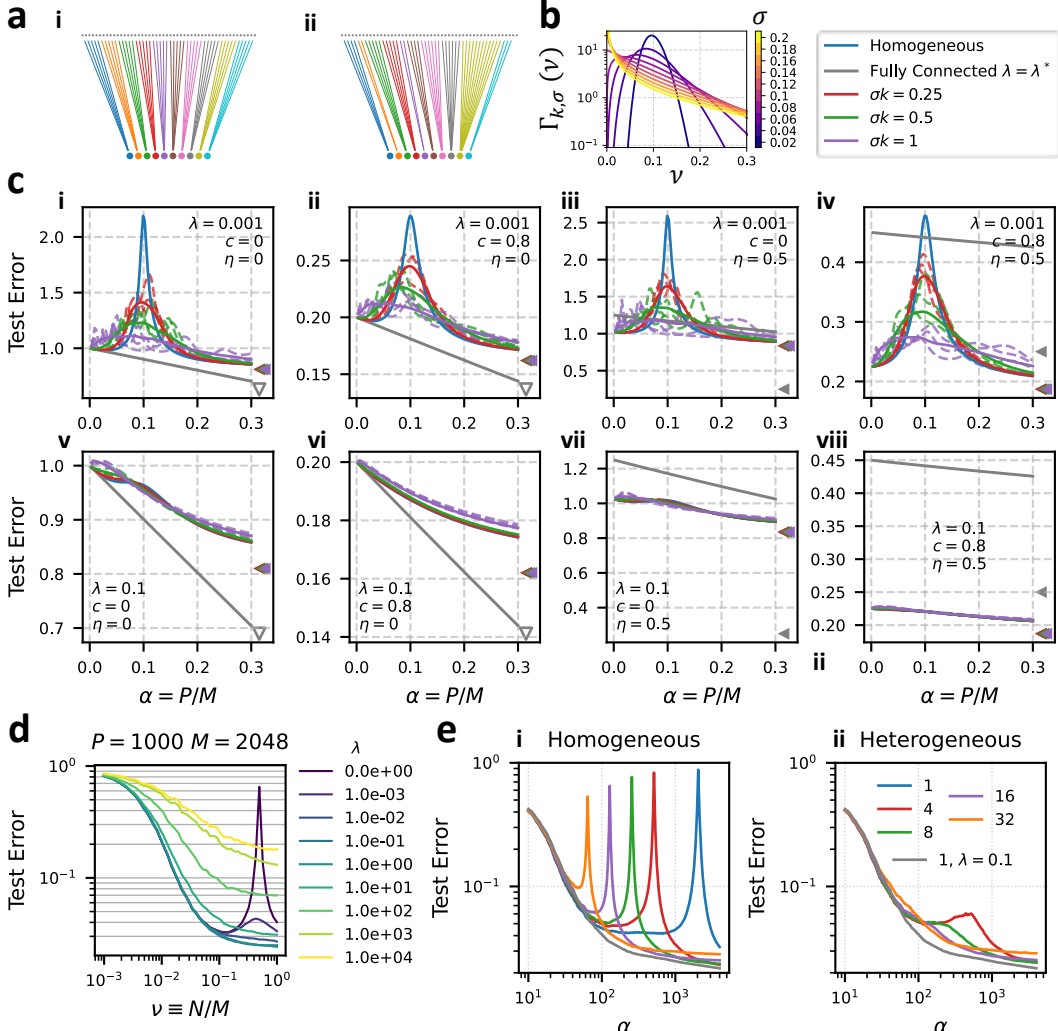

Figure 3: Heterogeneous ensembling mitigates double-descent. (a) We compare (i) homogeneous ensembling, in which $k$ readouts connect to the same fraction $\nu = 1/k$ of features, and (ii) heterogeneous ensembling (b) In heterogeneous ensembling subsampling fractions are drawn i.i.d. from $\Gamma_{k,\sigma}(\nu)$, shown here for $k = 10$, then re-scaled to sum to 1. (c) Generalization Error Curves for Homogeneous and Heterogeneous ensembling with $k = 10$, $\zeta = 0$, $\rho = 0.3$ and indicated values of $\lambda$, $c$, and $\eta$. Blue: homogeneous subsampling. Red, green, and purple show heterogeneous subsampling with $\sigma = 0.25/k, 0.5/k, 1/k$ respectively. Dashed lines show learning curves for 3 particular realizations of $\{\nu_{11}, \ldots, \nu_{kk}\}$. Solid curves show the average over 100 realizations. Gray shows the learning curve for a single linear readout with $\nu = 1$ and optimal regularization (eq. 193). Triangular marks show the asymptotic generalization error ($\alpha \to \infty$), with downward-pointing gray triangles indicating an asymptotic error of zero. (d,e) Generalization error of linear classifiers applied to the imagewoof dataset with ResNext features averaged over 100 trials. (d) $P = 100$, $k = 1$ varying subsampling fraction $\nu$ and regularization $\lambda$ (legend). (e) Generalization error of (i) homogeneous and (ii) heterogeneous (with $\sigma = 0.75/k$) ensembles of classifiers. Legend indicates $k$ values. $\lambda = 0$ except for gray curves, where $\lambda = 0.1$

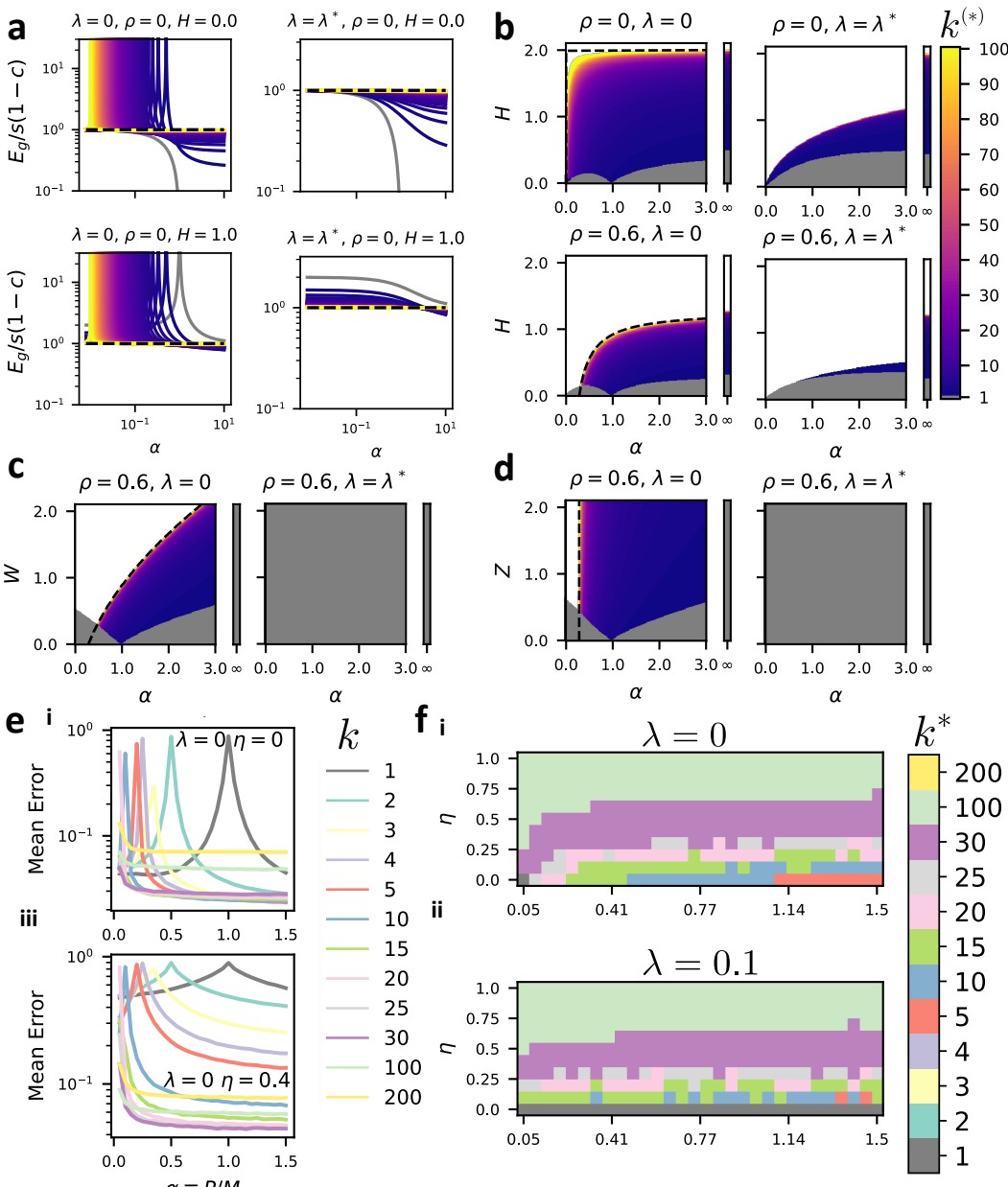

Figure 4: Task parameters dictate the ensembling-subsampling trade-off: (a-d) In the setting of proposition 2 in the special case where all $\nu_{rr'} = \frac{1}{k}\delta_{rr'}$ so that feature subsets are mutually exclusive and the total number of weights is conserved. (a) We plot the reduced generalization errors $\mathcal{E}$ (for $\lambda = 0$, using eq. 22) and $\mathcal{E}^*$ (for $\lambda = \lambda^*$ using eq. 23) of linear ridge ensembles of varying size $k$ with $\rho = 0$ and $H = 0, 1$ (values indicated above plots). Grey lines indicate $k = 1$, dashed black lines $k \to \infty$, and intermediate $k$ values by the colorbar. (b) We plot optimal ensemble size $k^*$ (eqs. 25, 26) in the parameter space of sample size $\alpha$ and reduced readout noise scale $H$ setting $W = Z = 0$. Grey indicates $k^* = 1$ and white indicates $k^* = \infty$, with intermediate values given by the colorbar. Appended vertical bars show $\alpha \to \infty$. Dotted black lines show the analytical boundary between the intermediate and noise-dominated phases given by eq. 214. (c) optimal readout $k^*$ phase diagrams as in (b) but showing $W$-dependence with $H = Z = 0$. (d) optimal readout $k^*$ phase diagrams as in (b) but showing $Z$-dependence with $H = W = 0$. (e) Learning curves for feature-subsampling ensembles of linear classifiers combined using a majority vote rule on the imagewoof classification task (see Appendix E). As in (a-d) we set $\nu_{rr'} = \frac{1}{k}\delta_{rr'}$. Error is calculated as the probability of incorrectly classifying a test example. $\lambda$ and $\eta$ values are indicated in each panel. (f) Numerical phase diagrams showing the value of $k$ which minimizes test error in the parameter space of sample size $P$ and readout noise scale $\eta$, with regularization (i) $\lambda = 0$ (pseudoinverse rule) (ii) $\lambda = 0.1$.

# 7    Acknowledgements

CP is supported by NSF Award DMS-2134157, NSF CAREER Award IIS-2239780, and a Sloan Research Fellowship. This work has been made possible in part by a gift from the Chan Zuckerberg Initiative Foundation to establish the Kempner Institute for the Study of Natural and Artificial Intelligence. BSR was also supported by the National Institutes of Health Molecular Biophysics Training Grant NIH/ NIGMS T32 GM008313. We thank Jacob Zavatone-Veth and Blake Bordelon for thoughtful discussion and comments on this manuscript.

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

# A    Table of Parameters from Proposition 2 and Figures 2,3,4

| | |
|---|---|
| Data Scale | $s$ |
| Correlation Strength | $c$ |
| Data-Task Alignment | $\rho$ |
| Readout Noise Scale | $\eta$ |
| Feature noise scale | $\omega$ |
| Label Noise Scale | $\zeta$ |
| Ensemble Size | $k$ |
| Subsampling Fractions | $\nu_{rr}$    $r = 1, \ldots, K$ |
| Subsampling Overlap Fractions | $\nu_{rr'}$    $r \neq r'$ |
| Sample Size | $P \to \infty$ |
| Data Dimensionality | $M \to \infty$ |
| Sample Complexity | $\alpha \equiv \frac{P}{M}$    (finite) |
| Effective Noise-To-Signal Ratios | $H \equiv \frac{\eta^2}{s(1-c)}, W \equiv \frac{\omega^2}{s(1-c)}, Z \equiv \frac{\zeta^2}{s(1-c)}$ |

# B    Code Availability and Compute

All Code used in this paper has been made available online (see https://github.com/benruben87/Learning-Curves-for-Heterogeneous-Feature-Subsampled-Ridge-Ensembles.git). This includes code used to perform numerical experiments, calculate theoretical learning curves, and produce plots as well as the custom Mathematica libraries used to simplify the generalization error in the special case of equicorrelated data. The compute time required to do all of the calculations in this paper is approximately 3 GPU days.

# C    Homogeneous and Heterogeneous Subsampling

In this section, we describe the homoegeneous and heterogeneous subsampling strategies, as used in this work, in detail. This sampling method was used in calculating the theoretical loss curves for heterogeneous subsampling experiments seen in main text Fig. 3, and in the heterogeneous subsampling experiments applied to the ResNext-features-based image classification task in Figs. 3e, S2.

In homogeneous ensembling, the subsampling fractions $\nu_{rr} = N_r/M$ are chosen as $\nu_{rr} = 1/k$ for all $r = 1, \ldots, k$. In heterogeneous ensembling, the subsampling fractions $\{\nu_{11}, \ldots, \nu_{kk}\}$ are generated according to the following statistical process:

1. Each fraction $\nu_{rr}$ is drawn independently as $\nu_{rr} \sim \Gamma_{k,\sigma}$, where a $\Gamma_{k,\sigma}$ represents a Gamma distribution with mean $\frac{1}{k}$ and variance $\sigma^2$.

2. The fractions are re-scaled in order to sum to 1: $\nu_{rr} \to \nu_{rr}/(\nu_1 + \cdots + \nu_k)$

Equivalently, the subsampling fractions are drawn from a Dirichlet distribution parameterized by the ensemble size $k$ and a chosen variance $\sigma$ as $(\nu_{11}, \ldots, \nu_{kk}) \sim \text{Dir}((\sigma k)^{-2}, \ldots, (\sigma k)^{-2})$ [51].

In main text Fig. 3c, we combine this sampling strategy with theoretical learning curves for the equicorrelated data model in a quasi-numerical experiment. At each trial of the experiment, we draw a particular realization of the subsampling fractions $\{\nu_{11}, \ldots, \nu_{kk}\}$, then use the analytical expression (eq. 16) to calculate the resulting learning curve. Dotted lines show the loss curves for 3 single trials, corresponding to three particular realizations of the subsampling fractions. The solid lines show the average over 100 trials.

Note that we have defined our own convention for the parameterization of the $\Gamma$ distribution in which the inverse of the mean and the standard deviation are specified. In terms of the standard "shape" and "scale" parameters, we have:

$$\Gamma_{k,\sigma} \equiv \Gamma\left(\text{shape } = (k\sigma)^{-2}, \text{scale } = k\sigma^2\right) \tag{27}$$

# D   Numerical Linear Regression with Synthetic Datasets

Numerical experiments were performed using the PyTorch library [52]. The code used to perform numerical experiments and generate plots has been made publicly available (see section B).

In numerical regression experiments, synthetic datasets with label noise are constructed as described in section 2.1, drawing data randomly from multivariate Gaussian distributions and adding label noise (see "DatasetMaker.py" in available code). Representing the training set in terms of a data matrix $\boldsymbol{\Psi} \in \mathbb{R}^{M \times P}$ in which column $\mu$ consist of the training point $\boldsymbol{\psi}_\mu$, and the labels with a column vector $\boldsymbol{y}$ such that $\boldsymbol{y}_\mu = y_\mu$, the learned weights are calculated as:

$$\hat{\boldsymbol{w}} = \boldsymbol{\Psi} \left( \boldsymbol{\Psi}^\top \boldsymbol{\Psi} + \lambda \boldsymbol{I}_p \right)^{-1} \boldsymbol{y} \tag{28}$$

In the ridgeless case, a pseudoinverse is used:

$$\hat{\boldsymbol{w}} = \boldsymbol{\Psi}^\dagger \boldsymbol{y} \tag{29}$$

# E   Ensembled Linear Classification of Imagenet Images

In this section, we provide the details of numerical experiments which demonstrate that qualitative insights gained from our analysis of the linear regression task with Gaussian data carries over to a practical machine learning task. In particular, we apply ensembles of linear classifiers to datasets constructed using a pre-trained ResNext [46] a specific type of Convolutional Neural Network (CNN).

## E.1   Dataset Construction

To construct the dataset, we start with a set of $n$ images $\{\boldsymbol{x}^\mu\}_{\mu=1}^n$ from a subset of $C = 10$ classes of the imagenet dataset. For each image $\boldsymbol{x}^\mu$, we obtain a corresponding feature vector $\boldsymbol{\psi}^\mu \in \mathbb{R}^M$ as the last-hidden-layer activation of the ResNext [46], which has been pre-trained on the imagenet classification task [44]. The architecture we use produces $M = 2048$ features per image. These features will serve as the data input to the downstream linear classifier. The corresponding labels $\boldsymbol{y}^\mu \in \mathbb{R}^C$ are one-hot vectors.

We construct two datasets using this method, using images from the "Imagenette" and "Imagewoof" datasets [45]. For the "Imagenette" task, we a construct a training set of size $n_{tr} = 9469$ and a test set of size $n_{test} = 3925$ containing features corresponding to images from 10 unrelated classes (tench, English springer, cassette player, chain saw, church, French horn, garbage truck, gas pump, golf ball, parachute). For the "Imagewoof" task, we a construct a training set of size $n_{tr} = 9025$ and a test set of size $n_{test} = 3929$ containing features corresponding to images of 10 different dog breeds (Australian terrier, Border terrier, Samoyed, Beagle, Shih-Tzu, English foxhound, Rhodesian ridgeback, Dingo, Golden retriever, Old English sheepdog). The imagewoof classification task is naturally more difficult. The statistics of the resulting datasets are described in Fig. S1, where we plot the data-data covariance matrix, feature-feature covariance matrix, and the eigenvalue spectrum for both the "Imagenette" and "Imagewoof" tasks.

## E.2   Model Training

At training time a dataset of $P$ examples is constructed: $\mathcal{D} = \{\boldsymbol{\psi}^\mu, \boldsymbol{y}^\mu\}_{\mu=1}^P$. We represent the training set with a "design matrix" $\boldsymbol{\Phi} \in \mathbb{R}^{P \times M}$ and a label matrix $\boldsymbol{Y} \in \{0,1\}^{P \times C}$. The loss function for each ensemble member is generalized to multi-class regression. With ridge regularization, the objective for each ensemble member $r$ becomes:

$$\hat{\boldsymbol{W}}_r = \underset{\boldsymbol{W}_r \in \mathbb{R}^{N_r \times C}}{\arg\min} \left[ \sum_{\mu=1}^P \| \frac{1}{\sqrt{N_r}} \boldsymbol{A}_r \bar{\boldsymbol{\psi}}^\mu \boldsymbol{W}_r + \boldsymbol{\xi}_r^\mu - \boldsymbol{y}^\mu \|_2^2 + \lambda_r \| \boldsymbol{W}_r \|_F^2 \right],$$

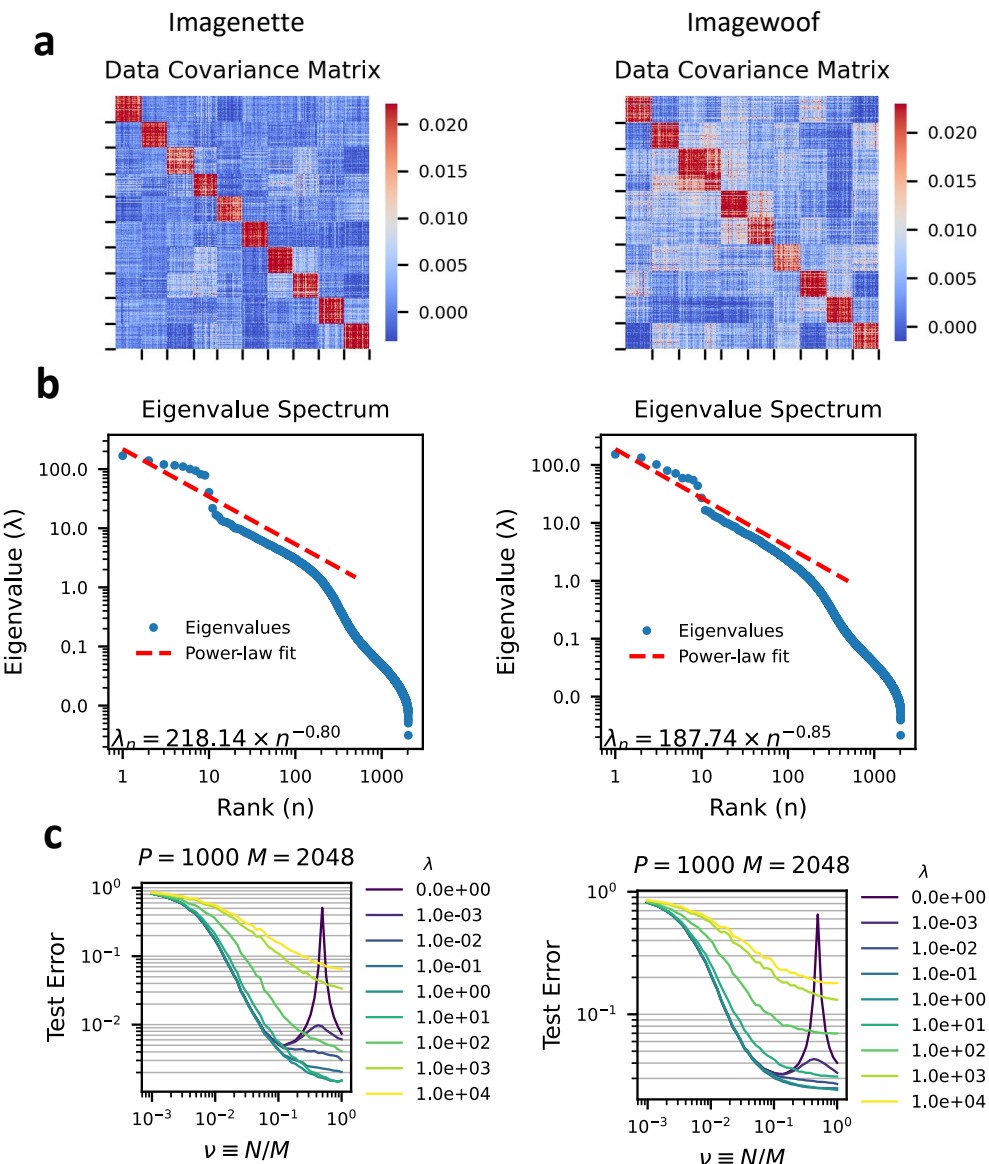

Figure S1: In numerical experiments, we train linear classifiers to predict labels of imagenet images based on their last-hidden-layer representations in a pre-trained RexNext deep learning architecture [46]. Here, we show the structure of the datasets constructed using the ResNext feature map for the Imagenette task (left), which consists of categorizing images from 10 unrelated categories, and the Imagewoof task (right), which consists of categorizing images from 10 different dog breeds. (a) Gram matrix of the centered ResNext features defined as $\frac{1}{P}\left(\mathbf{\Phi} - \bar{\mathbf{\Phi}}\right)^{\top}\left(\mathbf{\Phi} - \bar{\mathbf{\Phi}}\right)$ for data matrix $\mathbf{\Phi} \in \mathbb{R}^{P \times M}$ where $P$ is the total size of the dataset. Dataset is sorted by label and tick marks show the boundaries between classes. (b) The covariance eigenspectrum of the ResNext features is well described by a power law decay. (c) Generalization error of Linear classification with a single linear predictor with access to a fraction $\nu = N/M$ of the ResNext features averaged over 100 trials (see discussion in section E.4.1)

where $\| \cdot \|_F$ denotes the Frobenius norm and where the readout noise vector $\boldsymbol{\xi}_r^\mu \sim \mathcal{N}(0, \eta^2 \boldsymbol{I}_C)$ is a $C$ - dimensional readout noise which corrupts the model's prediction. The weights $\hat{\boldsymbol{W}}_r$ can be determined in closed form as follows:

$$\hat{\boldsymbol{W}}_r = \frac{1}{\sqrt{N_r}} \left( \frac{1}{N_r} \boldsymbol{A}_r \boldsymbol{\Phi}^\top \boldsymbol{\Phi} \boldsymbol{A}_r^\top + \lambda_r \boldsymbol{I} \right)^{-1} \left( \boldsymbol{A}_r \boldsymbol{\Phi}^\top (\boldsymbol{Y} - \boldsymbol{\Xi}_r) \right).$$

where we have defined $[\boldsymbol{\Xi}_r]_{\mu c} = [\boldsymbol{\xi}_r^\mu]_c$. When $\lambda = 0$ we instead use a pseudoinverse rule. In all experiments presented here, we use measurement matrices $\boldsymbol{A}_r$ which implement an ordinary subsampling of the features, so that the rows and columns of $\boldsymbol{A}_r$ consist of one-hot vectors.

## E.3    Model Prediction

Once trained, the learned weights may be used to predict the label of a new example $\boldsymbol{\psi}$ as follows. For $r = 1, \ldots, k$ we calculate the prediction of each ensemble member by first assigning each class a "score". The scores of predictor $r$ are stored in a vector $\boldsymbol{f}_r \in \mathbb{R}^C$:

$$\boldsymbol{f}_r(\boldsymbol{\psi}) = \frac{1}{\sqrt{N_r}} \boldsymbol{A}_r \boldsymbol{\psi} \hat{\boldsymbol{W}}_r + \boldsymbol{\xi}_r$$

Where $\boldsymbol{\xi}_r \sim \mathcal{N}(0, \eta^2 \boldsymbol{I}_C)$ is drawn randomly at model evaluation. Each ensemble member's "vote" corresponds to the class with the largest score. The prediction of the ensemble is then calculated as a majority vote of the ensemble members. Generalization error is then calculated as the probability of misclassifying an example from the test set.

## E.4    Linear Classification Experiments

We apply the described majority-vote linear classifier ensembles to the ResNext-Imagenette and ResNext-Imagewoof tasks in three different experiments.

### E.4.1    Reduncancy of ResNext Features

In the first experiment, we investigate the performance of a single linear classifier ($k = 1$) as the fraction of features $\nu$ which it has access to varies. We set $P = 1000$ and vary $\nu$ over 50 values on a logarithmic scale from $10^{-3}$ to 1. We also vary the regularization strength over 0 and a logarithmic scale from $10^{-3}$ to $10^4$. We average over 100 trials. At each trial the particular subset of $P = 1000$ training examples and the particular subset of $\nu * M$ features is randomly re-sampled. We find that ResNext features are highly redundant – classification accuracy is very robust to subsampling of the features. For example, in the Imagewoof classification task with the best regularization tested ($\lambda = 0.1$), test error increases from about 1% to about 2% as the subsampling fraction decreases from $\nu = 1$ to $\nu = 0.1$ (meaning 90% of the features are ignored) (see Fig. 3d). Similarly, for the imagenette task, test error increases from about 0.1% to about 0.2% as the subsampling fraction decreases from $\nu = 1$ to $\nu = 0.1$ (see Fig. S1c). Code used to run these experiments may be found in the folder "DeepNet_Subsamp" in the GitHub repository.

### E.4.2    Heterogeneous Ensembling Mitigates Double-Descent

In the second experiment, we compare learning curves for homogeneous ensembling and heterogeneous ensembling applied to the ResNext-Imagewoof classification task. In each trial, we train ensembles of $k = \{1, 4, 8, 16, 32\}$ linear predictors whose subsampling fractions $\{\nu_{rr}, \ldots, \nu_{kk}\}$ are assigned either with Homogeneous ensembling or with Heterogeneous Ensembling with $\sigma = 0.75/k$ C. After subsampling fraction are assigned, the training set is randomly shuffled. We iterate over 50 sample sizes $P$ logarithmically distributed from 400 to 4000, and then add the values $M/k$ for each $k$ to the list of $P$ values. We repeat for 100 trials for both $\lambda_r = 0$ (pseudoinverse rule) and $\lambda = 0.1$,

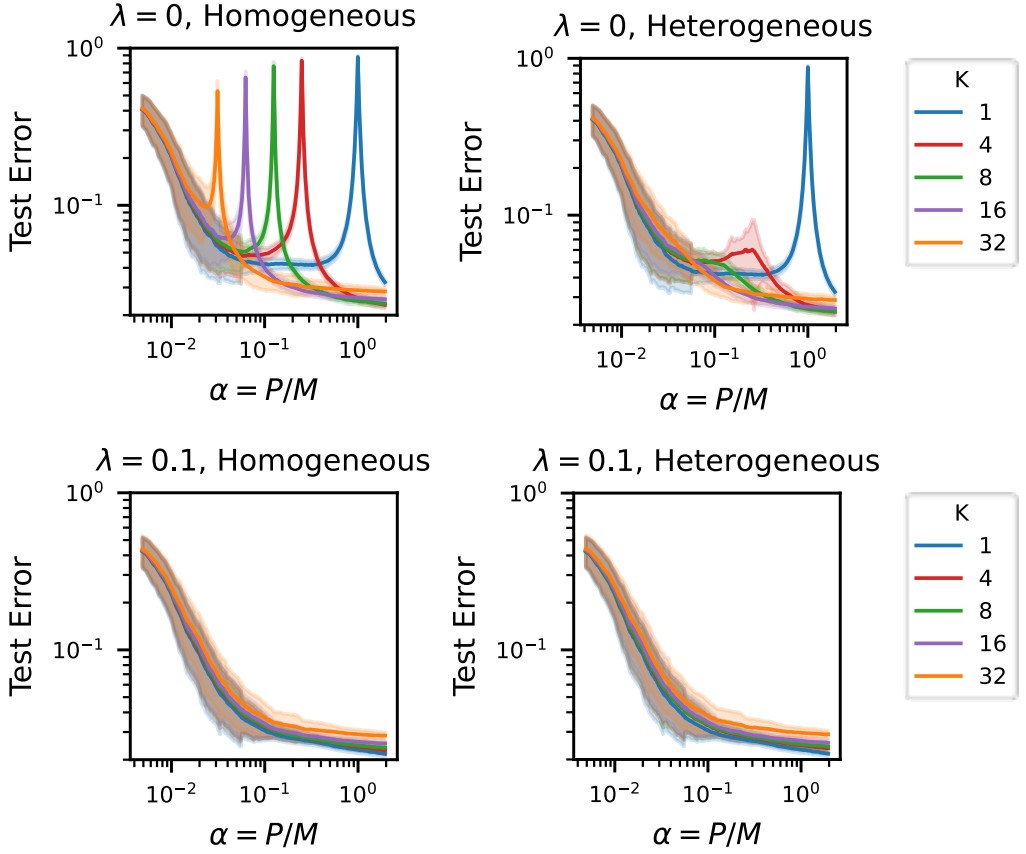

Figure S2: Results of the experiment described in section E.4.2. Learning curves for ResNext-Imagenewoof classification task in linear classifier ensembles. In Homogeneous ensembling (left) all $\nu_{rr} = 1/k$ with $k$ indicated by the legend. In heterogeneous ensembling $\nu_{rr}$ are drawn from a Dirichlet distribution as described in section C, with $\sigma = 3/(4k)$. We use regularization $\lambda = 0$ (top) and $\lambda = 0.1$ (bottom). $\lambda = 0, 0.1$. Lines represent an average over 100 trials, shaded regions show standard deviation. We set $\nu_{rr'} = 0$ for $r \neq r'$.

which was found to mitigate double-descent by the parameter sweep in Fig. S1c. In main text Fig. 3e, we plot the mean learning curves over 100 trials. In Fig. S2 we show standard deviation over the 100 trials as shaded error bars. When $\lambda = 0$, heterogeneous ensembling mitigates double-descent, leading to a monotonically decreasing learning curve for sufficiently high $k$. When $\lambda = 0.1$, homogeneous and heterogeneous ensembles of size $k$ perform similarly. Code used to run these experiments may be found in the folder "DeepNet_HomVHet" in the GitHub repository.

### E.4.3 Readout Noise Encourages Ensembling

In the third experiment, we test the effect of a readout noise which is independent across the members of the ensemble on generalization error. We do this by sweeping over the readout noise scale $\eta$ as defined in sections E.2, E.3. For $\lambda = 0, 0.1$ and $\nu = 0, .1, \ldots, 1$ we compute the learning curves of linear predictors with $k = 1, 2, 3, 4, 5, 10, 15, 20, 25, 30, 100, 200$ and all $\nu_{rr} = 1/k$, averaged over 50 trials. These learning curves are shown in Fig. S3 for both the ResNext-Imagenette and ResNext-Imagewoof task. In Figs. S4a and S5a, we plot the value $k^*$ which minimizes error as pase diagrams in the parameter space of $\alpha$ and $\eta$, analogous to the phase diagrams in Fig. 4b. We see that the qualitative shape of these phase diagrams is similar to the equicorrelated model. The differences may be attributed to the nonlinear nature of the classification task. Furthermore, we find

that, in general the optimal $k^*$ tends to be higher with the Imagenette dataset than with the Imagewoof dataset, in agreement with our finding in the equicorrelated regression model that as $\rho$ increases (making the classification task easier), optimal ensemble size tends to increase (Fig. 4b and S6, S7, S8). In Fig S3 we see that there are often a number of $k$ values for which test error is at or near to its lowest. To quantify this, we also plot diagrams of the minimum and maximum values of $k$ that are within an small margin $\epsilon$ of the minimum measured error. For the ResNext-Imagenette task, we use $\epsilon = 0.001$ and for ResNext-Imagewoof $\epsilon = 0.01$. We see that there is a wide array of $k$ values which bring error near-to-minimum in practice (Figs S4b,c, S5b,c). We also plot the minimum achieved error, and the difference between minimum errors at $\lambda = 0.1, 0$ (Figs. S4d, S5d). Code used to run these experiments may be found in the folder "DeepNet_PD" in the GitHub repository.

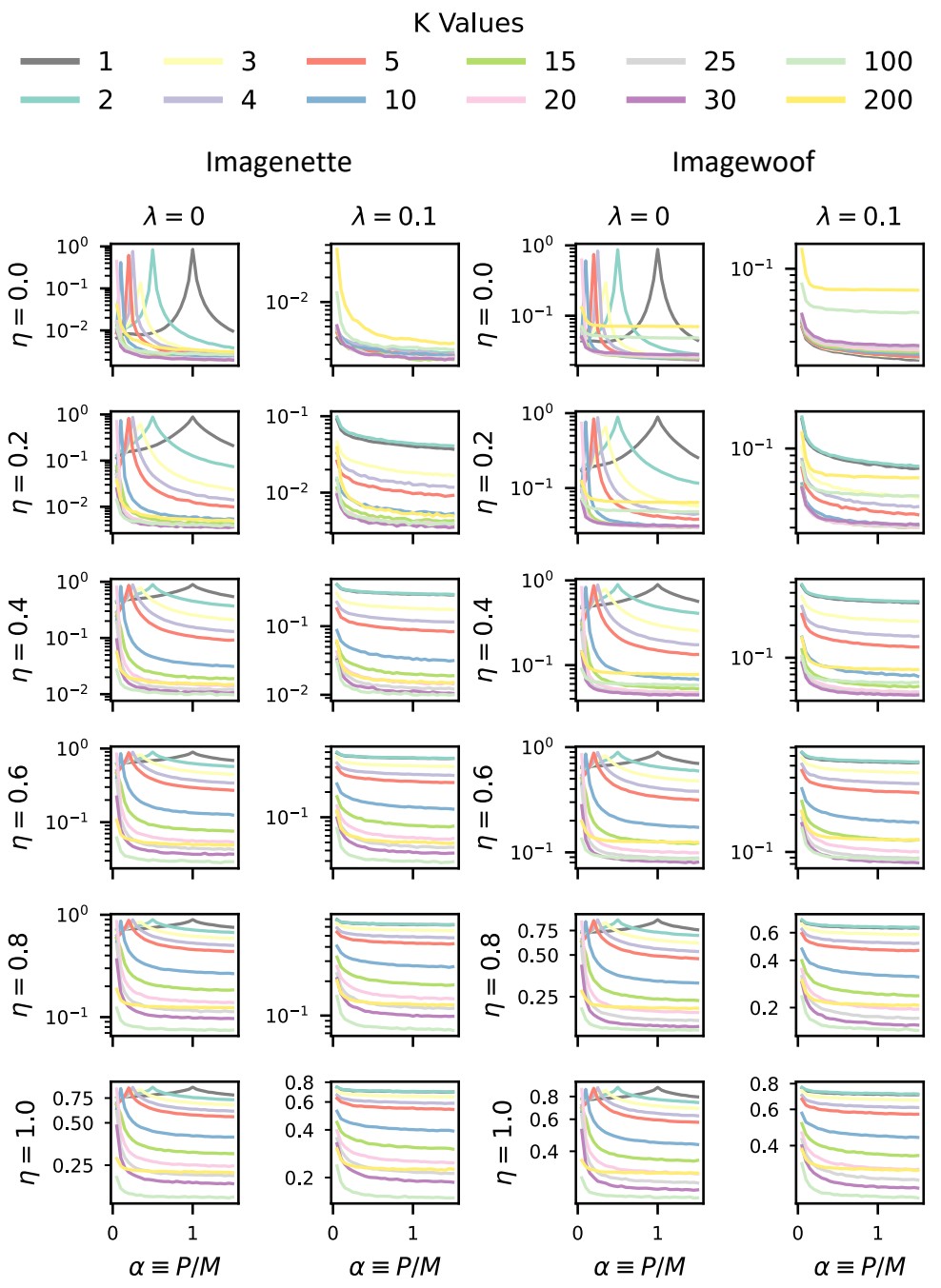

Figure S3: Learning curves for ensembles of linear classifiers with homogeneous subsampling for $\lambda = 0, 0.1$ and readout noise $\eta$ values indicated in the figure. Results are averaged over 50 trials. Experiments are described in section E.4.3

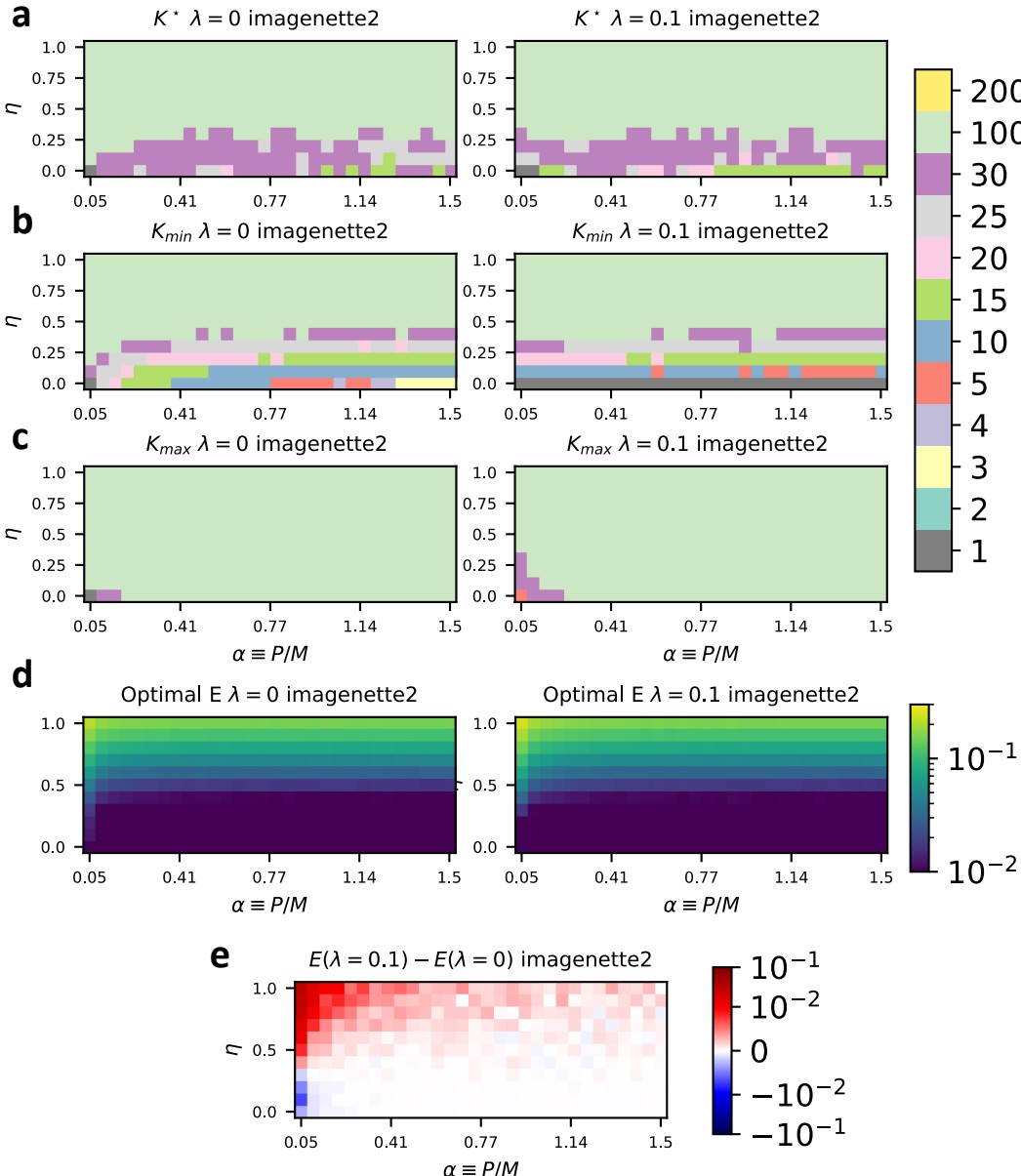

Figure S4: Diagrams described in section E.4.3 for the ResNext-Imagenette experiment. Using learning curves in Fig. S3 (and for additional values of $\eta$ not shown there) , we plot (a) Optimal $k^*$ in the parameter space of $\alpha$ and $\eta$, (b) Minimum value of $k$ for which error is within a tolerance $\epsilon = .001$ of its value for $k^*$, (b) Maximum value of $k$ for which error is within a tolerance $\epsilon = .001$ of its value for $k^*$, (d) the value of the minimum error $E(k^*)$, and (e) the difference between this optimal error for $\lambda = 0.1$ and $\lambda = 0$.

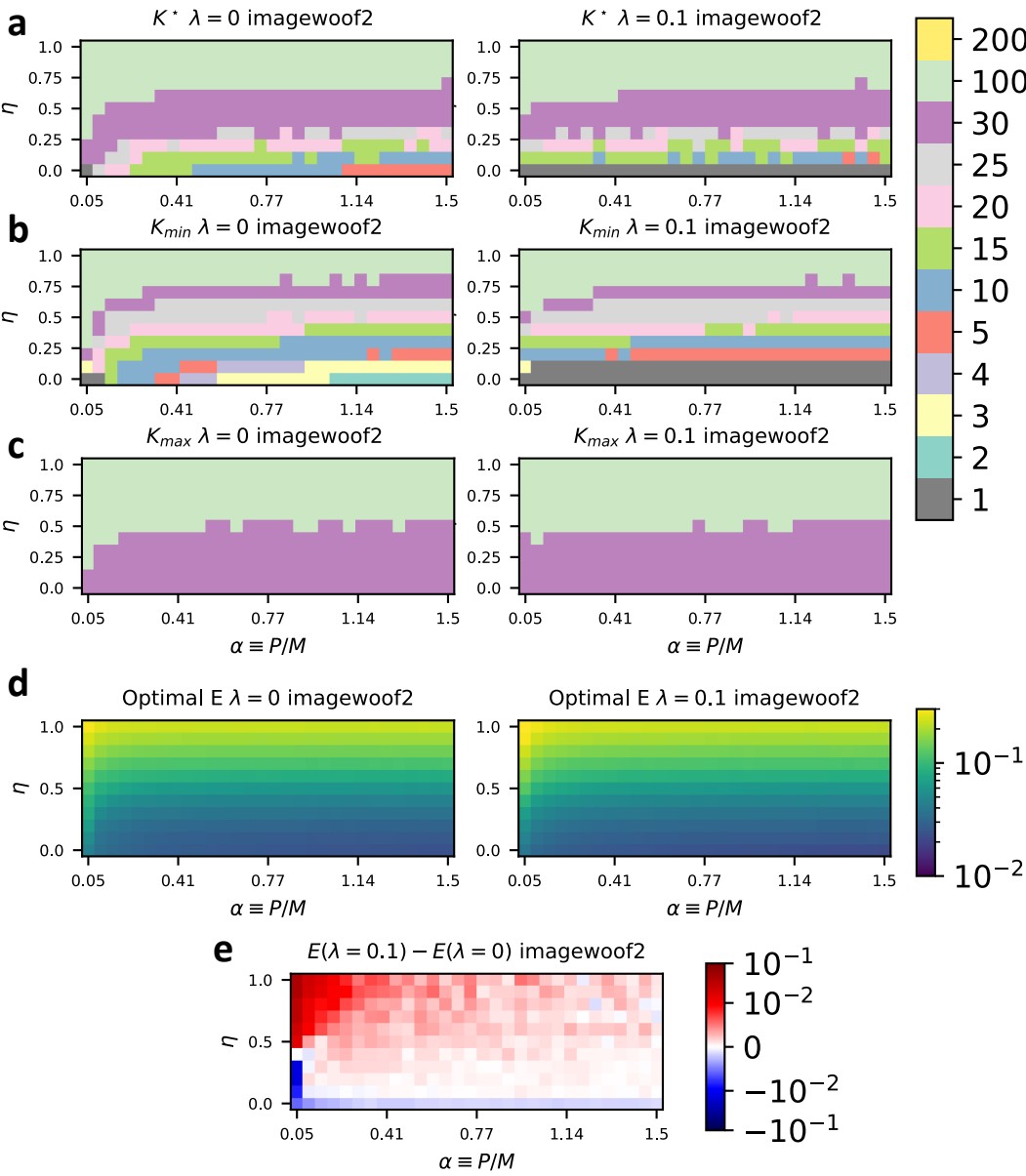

Figure S5: Diagrams described in section E.4.3 for the ResNext-Imagewoof experiment. Using learning curves in Fig. S3 (and for additional values of $\eta$ not shown there) , we plot (a) Optimal $k^*$ in the parameter space of $\alpha$ and $\eta$, (b) Minimum value of $k$ for which error is within a tolerance $\epsilon = .001$ of its value for $k^*$, (b) Maximum value of $k$ for which error is within a tolerance $\epsilon = .001$ of its value for $k^*$, (d) the value of the minimum error $E(k^*)$, and (e) the difference between this optimal error for $\lambda = 0.1$ and $\lambda = 0$.

# F  Generalization error of ensembled linear regression from the replica trick

Here we use the Replica Trick from statistical physics to derive analytical expressions for $E_{rr'}$. We treat the cases where $r = r'$ and $r \neq r'$ separately. Following a statistical mechanics approach, we calculate the average generalization error over a Gibbs measure with inverse temperature $\beta$;

$$Z = \int \prod_r d\boldsymbol{w} \exp\left(-\frac{\beta}{2}\sum_r E_t^r - \frac{M\beta}{2}\sum_{r,r'} J_{rr'} E_{rr'}(\boldsymbol{w_r}, \boldsymbol{w_{r'}})\right) \tag{30}$$

$$E_t^r = \sum_{\mu=1}^{P}\left(\frac{1}{\sqrt{N_r}}\boldsymbol{w}_r^\top \boldsymbol{A}_r \bar{\boldsymbol{\psi}}_\mu + \xi_r - y_\mu\right)^2 + \lambda|\boldsymbol{w}_r^2| \tag{31}$$

In the limit where $\beta \to \infty$ the gibbs measure will concentrate around the weight vector $\hat{\boldsymbol{w}}_r$ which minimizes the regularized loss function. The replica trick relies on the following identity:

$$\langle\log(Z[\mathcal{D}])\rangle_{\mathcal{D}} = \lim_{n\to 0}\frac{1}{n}\log\left(\langle Z^n\rangle_{\mathcal{D}}\right) \tag{32}$$

where $\langle\cdot\rangle_{\mathcal{D}}$ represents an average over all quenched disorder in the system. In this case, quenched disorder – the disorder which is fixed prior to and throughout training of the weights – consists of the selected training examples along with their feature noise and label noise: $\mathcal{D} = \{\boldsymbol{\psi}_\mu, \boldsymbol{\sigma}^\mu, \epsilon^\mu\}_{\mu=1}^P$. The calculation proceeds by first computing the average of the replicated partition function assuming $n$ is a positive integer. Then, in a non-rigorous but standard step, we analytically extend the result to $n \to 0$.

## F.1  Diagonal Terms

We start by calculating $E_{rr}$ for some fixed choice of $r$. This derivation partially follows section D.3 from [36], with the addition of readout noise and label noise. Noting that the diagonal terms of the generalization error $E_{rr}$ only depend on the learned weights $\boldsymbol{w}_r$, and the loss function separates over the readouts, we may consider the Gibbs measure over only these weights:

$$Z = \int d\boldsymbol{w}_r \exp\left(-\frac{\beta}{2\lambda}E_r^t - \frac{JM\beta}{2}E_{rr}(\boldsymbol{w}_r)\right) \tag{33}$$

$$\langle Z^n\rangle_{\mathcal{D}} = \int \prod_a d\boldsymbol{w}_r^a \mathbb{E}_{\{\boldsymbol{\psi}_\mu, \boldsymbol{\sigma}^\mu, \epsilon^\mu\}}$$

$$\exp\left(-\frac{\beta M}{2\lambda}\sum_{\mu,a}\frac{1}{M}\left[\frac{1}{\sqrt{\nu_{rr}}}\boldsymbol{w}_r^{a\top}\boldsymbol{A}_r\left(\boldsymbol{\psi}_\mu + \boldsymbol{\sigma}^\mu\right) - \boldsymbol{w}^{*\top}\boldsymbol{\psi}_\mu - \sqrt{M}(\epsilon^\mu - \xi_r^\mu)\right]^2\right. \tag{34}$$

$$\left. -\frac{\beta}{2}\sum_a |\boldsymbol{w}_r^a|^2 - \frac{JM\beta}{2}\sum_a E_{rr}(\boldsymbol{w}^a)\right)$$

Next we must perform the averages over quenched disorder. We first integrate over $\{\boldsymbol{\psi}_\mu, \boldsymbol{\sigma}^\mu, \xi_r^\mu, \epsilon^\mu\}_{\mu=1}^P$. Noting that the scalars

$$h_\mu^{ra} \equiv \frac{1}{\sqrt{M}}\left[\frac{1}{\sqrt{\nu_{rr}}}\boldsymbol{w}_r^{a\top}\boldsymbol{A}_r\left(\boldsymbol{\psi}_\mu + \boldsymbol{\sigma}^\mu\right) - \boldsymbol{w}^{*\top}\boldsymbol{\psi}_\mu - \sqrt{M}(\epsilon^\mu - \xi_r^\mu)\right]$$

are Gaussian random variables (when conditioned on $A_r$) with mean zero and covariance:

$$\langle h_\mu^{ra} h_\nu^{rb} \rangle = \delta_{\mu\nu} Q_{ab}^{rr} \tag{35}$$

$$Q_{ab}^{rr} = \frac{1}{M} \left[ \left( \frac{1}{\sqrt{\nu_{rr}}} \boldsymbol{w}_r^{a\top} \boldsymbol{A}_r - \boldsymbol{w}^{*\top} \right) \boldsymbol{\Sigma}_s \left( \frac{1}{\sqrt{\nu_{rr}}} \boldsymbol{A}_r^\top \boldsymbol{w}_r^b - \boldsymbol{w}^* \right) \right.$$
$$\left. + \frac{1}{\nu_{rr}} \boldsymbol{w}_r^{a\top} \boldsymbol{A}_r \boldsymbol{\Sigma}_0 \boldsymbol{A}_r^\top \boldsymbol{w}_r^b + M(\zeta^2 + \eta_r^2) \right] \tag{36}$$

To perform this integral we re-write in terms of $\{\boldsymbol{H}_\mu^r\}_{\mu=1}^P$, where

$$\boldsymbol{H}_\mu^r = \begin{bmatrix} h_\mu^{r1} \\ h_\mu^{r2} \\ \vdots \\ h_\mu^{rn} \end{bmatrix} \in \mathbb{R}^n \tag{37}$$

$$\langle Z^n \rangle_{\mathcal{D}} = \int \prod_a d\boldsymbol{w}_r^a \mathbb{E}_{\{\psi_\mu, \boldsymbol{\sigma}^\mu, \epsilon^\mu\}} \exp\left( -\frac{\beta}{2\lambda} \sum_\mu \boldsymbol{H}_\mu^{r\top} \boldsymbol{H}_\mu^r - \frac{\beta}{2} \sum_a |\boldsymbol{w}_r^a|^2 - \frac{JM\beta}{2} \sum_a E_{rr}(\boldsymbol{w}^a) \right) \tag{38}$$

Integrating over the $\boldsymbol{H}_\mu^r$ we get:

$$\langle Z^n \rangle_{\mathcal{D}} = \int \prod_a d\boldsymbol{w}_r^a \exp\left( -\frac{P}{2} \log \det\left( \boldsymbol{I}_n + \frac{\beta}{\lambda} \boldsymbol{Q}^{rr} \right) - \frac{\beta}{2} \sum_a |\boldsymbol{w}_r^a|^2 - \frac{JM\beta}{2} \sum_a E_{rr}(\boldsymbol{w}_r) \right) \tag{39}$$

Next we integrate over $\boldsymbol{Q}^r$ and add constraints. We use the following identity:

$$1 = \prod_{ab'} \int dQ_{ab}^{rr} \delta\left( Q_{ab}^{rr} - \frac{1}{M} \left[ \left( \frac{1}{\sqrt{\nu_{rr}}} \boldsymbol{w}_r^{a\top} \boldsymbol{A}_r - \boldsymbol{w}^{*\top} \right) \boldsymbol{\Sigma}_s \left( \frac{1}{\sqrt{\nu_{rr}}} \boldsymbol{A}_r^\top \boldsymbol{w}_r^b - \boldsymbol{w}^* \right) \right.\right.$$
$$\left.\left. + \frac{1}{\nu_{rr}} \boldsymbol{w}_r^{a\top} \boldsymbol{A}_r \boldsymbol{\Sigma}_0 \boldsymbol{A}_r^\top \boldsymbol{w}_r^b + M(\zeta^2 + \eta_r^2) \right] \right) \tag{40}$$

Using the Fourier representation of the delta function, we get:

$$1 = \prod_{ab} \int \frac{1}{4\pi i/M} dQ_{ab}^{rr} d\hat{Q}_{ab}^{rr} \exp\left( \frac{M}{2} \hat{Q}_{ab}^{rr} \left( Q_{ab}^{rr} - \frac{1}{M} \left[ \left( \frac{1}{\sqrt{\nu_{rr}}} \boldsymbol{w}_r^{a\top} \boldsymbol{A}_r - \boldsymbol{w}^{*\top} \right) \boldsymbol{\Sigma}_s \left( \frac{1}{\sqrt{\nu_{rr}}} \boldsymbol{A}_r^\top \boldsymbol{w}_r^b - \boldsymbol{w}^* \right) \right.\right.\right.$$
$$\left.\left.\left. + \frac{1}{\nu_{rr}} \boldsymbol{w}_r^{a\top} \boldsymbol{A}_r \boldsymbol{\Sigma}_0 \boldsymbol{A}_r^\top \boldsymbol{w}_r^b + M(\zeta^2 + \eta_r^2) \right] \right) \right) \tag{41}$$

Inserting this identity into the replicated partition function and substituting $E_{rr}(\boldsymbol{w}_r^a) = Q_{aa}^{rr} - \zeta^2$ we find:

$$\langle Z^n \rangle_{\mathcal{D}} \propto$$

$$\int \prod_{ab} dQ^{rr}_{ab} d\hat{Q}^{rr}_{ab} \exp\left( -\frac{P}{2} \log \det\left( \boldsymbol{I}_n + \frac{\beta}{\lambda} \boldsymbol{Q}^{rr} \right) + \frac{1}{2} \sum_{ab} M \hat{Q}^{rr}_{ab} Q^{rr}_{ab} - \frac{JM\beta}{2} \sum_a (Q^{rr}_{aa} - \zeta^2) \right)$$

$$\int \prod_a d\boldsymbol{w}^a_r \exp\left( -\frac{\beta}{2} \sum_a |\boldsymbol{w}^a_r|^2 - \frac{1}{2} \sum_{ab} \hat{Q}^{rr}_{ab} \left[ \left( \frac{1}{\sqrt{\nu_{rr}}} \boldsymbol{w}^{a\top}_r \boldsymbol{A}_r - \boldsymbol{w}^{*\top} \right) \boldsymbol{\Sigma}_s \left( \frac{1}{\sqrt{\nu_{rr}}} \boldsymbol{A}^\top_r \boldsymbol{w}^b_r - \boldsymbol{w}^* \right) \right. \right.$$

$$\left. \left. + \frac{1}{\nu_{rr}} \boldsymbol{w}^{a\top}_r \boldsymbol{A}_r \boldsymbol{\Sigma}_0 \boldsymbol{A}^\top_r \boldsymbol{w}^b_r + M(\zeta^2 + \eta^2_r) \right] \right)$$

$$\tag{42}$$

In order to perform the Gaussian integral over the $\{\boldsymbol{w}^a_r\}$, we unfold over the replica index $a$. We first define the following:

$$\boldsymbol{w}^{\cdot}_r \equiv \begin{bmatrix} \boldsymbol{w}^1_r \\ \vdots \\ \boldsymbol{w}^n_r \end{bmatrix} \tag{43}$$

$$T^r \equiv \beta \boldsymbol{I}_n \otimes \boldsymbol{I}_{N_r} + \hat{\boldsymbol{Q}}^{rr} \otimes \left( \frac{1}{\nu_{rr}} \boldsymbol{A}_r (\boldsymbol{\Sigma}_s + \boldsymbol{\Sigma}_0) \boldsymbol{A}^\top_r \right) \tag{44}$$

$$V^r \equiv (\hat{\boldsymbol{Q}}^{rr} \otimes \boldsymbol{I}_{N_r})(\boldsymbol{1}_n \otimes \frac{1}{\sqrt{\nu_{rr}}} \boldsymbol{A}_r \boldsymbol{\Sigma}_s \boldsymbol{w}^*) \tag{45}$$

We then have for the integral over $w$

$$\int d\boldsymbol{w}^{\cdot}_r \exp\left( -\frac{1}{2} \boldsymbol{w}^{\cdot\top}_r T^r \boldsymbol{w}^{\cdot}_{r'} + V^{r\top} \boldsymbol{w}^{\cdot}_r \right) \tag{46}$$

$$= \exp\left( \frac{1}{2} V^{r\top} (T^r)^{-1} V^r - \frac{1}{2} \log \det(T^r) \right) \tag{47}$$

We can finally write the replicated partition function as:

$$\langle Z^n \rangle_{\mathcal{D}} \propto$$

$$\int \prod_{ab} dQ^{rr}_{ab} d\hat{Q}^r_{ab} \exp\left( -\frac{P}{2} \log \det\left( \boldsymbol{I}_n + \frac{\beta}{\lambda} \boldsymbol{Q}^{rr} \right) + \frac{1}{2} \sum_{ab} M \hat{Q}^{rr}_{ab} Q^{rr}_{ab} - \frac{JM\beta}{2} \sum_a (Q^{rr}_{aa} - \zeta^2) \right)$$

$$\exp\left( \frac{1}{2} V^{r\top} (T^r)^{-1} V^r - \frac{1}{2} \log \det(T^r) - \frac{1}{2} \sum_{ab} \hat{Q}^{rr}_{ab} (M(\zeta^2 + \eta^2_r) + \boldsymbol{w}^{*\top} \boldsymbol{\Sigma}_s \boldsymbol{w}^*) \right)$$

$$\tag{48}$$

We now make the following replica-symmetric ansatz:

$$Q^{rr}_{ab} = \beta^{-1} q \delta_{ab} + q_0 \tag{49}$$

$$\hat{Q}^{rr}_{ab} = \beta \hat{q} \delta_{ab} + \beta^2 \hat{q}_0 \tag{50}$$

which is well-motivated because the loss function is convex. We will verify that the chosen scalings are self-consistent in the zero-temperature limit where $\beta \to \infty$. We may then rewrite the partition function as follows:

$$\langle Z^n \rangle_{\mathcal{D}} = \exp\left(-\frac{nM}{2}\mathfrak{g}\left[q, q_0, \hat{q}, \hat{q}_0\right]\right) \tag{51}$$

Where the effective action is written:

$$\mathfrak{g}\left[q, q_0, \hat{q}, \hat{q}_0\right] = \alpha\left[\log(1+\frac{q}{\lambda}) + \frac{\beta q_0}{\lambda+q}\right] - (q\hat{q} + \beta q\hat{q}_0 + \beta q_0\hat{q}) + \beta J\left[(\beta^{-1}q + q_0) - \zeta^2\right]$$

$$-\frac{\beta}{\nu_{rr}M}\hat{q}^2\boldsymbol{w}^{*\top}\boldsymbol{\Sigma}_s\boldsymbol{A}_r^\top\boldsymbol{G}^{-1}\boldsymbol{A}_r\boldsymbol{\Sigma}_s\boldsymbol{w}^* + \frac{1}{M}\left[\log\det(\boldsymbol{G}) + \beta\hat{q}_0\operatorname{tr}[\boldsymbol{G}^{-1}\tilde{\boldsymbol{\Sigma}}]\right] + \beta\hat{q}\left(\zeta^2 + \eta_r^2 + \frac{1}{M}\boldsymbol{w}^{*\top}\boldsymbol{\Sigma}_s\boldsymbol{w}^*\right) \tag{52}$$

Where $\boldsymbol{G} \equiv \boldsymbol{I}_{N_r} + \hat{q}\tilde{\boldsymbol{\Sigma}}$ and $\tilde{\boldsymbol{\Sigma}} \equiv \frac{1}{\nu_{rr}}\boldsymbol{A}_r(\boldsymbol{\Sigma}_s + \boldsymbol{\Sigma}_0)\boldsymbol{A}_r^\top$

To determine the values of the order parameters we set the derivatives of $\mathfrak{g}\left[q, q_0, \hat{q}, \hat{q}_0\right]$, evaluated at zero source ($J = 0$), to zero:

$$\frac{\partial\mathfrak{g}}{\partial q_0} = 0 = \frac{\alpha\beta}{\lambda+q} - \beta\hat{q} \qquad\qquad \Rightarrow \hat{q} = \frac{\alpha}{\lambda+q} \tag{53}$$

$$\frac{\partial\mathfrak{g}}{\partial\hat{q}_0} = 0 = -\beta q + \frac{\beta}{M}\operatorname{tr}\left[\boldsymbol{G}^{-1}\tilde{\boldsymbol{\Sigma}}\right] \qquad\qquad \Rightarrow q = \frac{1}{M}\operatorname{tr}\left[\boldsymbol{G}^{-1}\tilde{\boldsymbol{\Sigma}}\right] \tag{54}$$

$$\frac{\partial\mathfrak{g}}{\partial q} = 0 = \frac{\alpha}{\lambda+q} - \frac{\alpha q_0\beta}{(\lambda+\beta q)^2} - \hat{q} - \beta\hat{q}_0 \qquad\qquad \Rightarrow \hat{q}_0 = -\frac{\alpha q_0}{(\lambda+\beta q)^2} \tag{55}$$

$$\frac{\partial\mathfrak{g}}{\partial\hat{q}} = 0 = -q - \beta q_0 + \beta\zeta^2 + \beta\eta_r^2 + \frac{1}{M}\operatorname{tr}\left[\boldsymbol{G}^{-1}\tilde{\boldsymbol{\Sigma}}\right] - \frac{\beta\hat{q}_0}{M}\operatorname{tr}\left[\left(\boldsymbol{G}^{-1}\tilde{\boldsymbol{\Sigma}}\right)^2\right]$$

$$+\frac{\beta}{M}\boldsymbol{w}^{*\top}\left[\boldsymbol{\Sigma}_s - \frac{2}{\nu_{rr}}\hat{q}\boldsymbol{\Sigma}_s\boldsymbol{A}_r^\top\boldsymbol{G}^{-1}\boldsymbol{A}_r\boldsymbol{\Sigma}_s + \frac{1}{\nu_{rr}}\hat{q}^2\boldsymbol{\Sigma}_s\boldsymbol{A}_r^\top\boldsymbol{G}^{-1}\tilde{\boldsymbol{\Sigma}}\boldsymbol{G}^{-1}\boldsymbol{A}_r\boldsymbol{\Sigma}_s\right]\boldsymbol{w}^* \tag{56}$$

$$\Rightarrow q_0 = \frac{1}{1-\gamma}\left(\zeta^2 + \eta_r^2 + \frac{1}{M}\boldsymbol{w}^{*\top}\left[\boldsymbol{\Sigma}_s - \frac{2}{\nu_{rr}}\hat{q}\boldsymbol{\Sigma}_s\boldsymbol{A}_r^\top\boldsymbol{G}^{-1}\boldsymbol{A}_r\boldsymbol{\Sigma}_s + \frac{1}{\nu_{rr}}\hat{q}^2\boldsymbol{\Sigma}_s\boldsymbol{A}_r^\top\boldsymbol{G}^{-1}\tilde{\boldsymbol{\Sigma}}\boldsymbol{G}^{-1}\boldsymbol{A}_r\boldsymbol{\Sigma}_s\right]\boldsymbol{w}^*\right) \tag{57}$$

where $\gamma \equiv \frac{\alpha}{M\kappa^2}\operatorname{tr}\left[(\boldsymbol{G}^{-1}\tilde{\boldsymbol{\Sigma}})^2\right]$ and $\kappa \equiv \lambda + q$. We retroactively confirm that the chosen $\beta$-scalings are correct by noticing that the saddle-point equations permit a solution where all order parameters remain $\mathcal{O}(1)$ as $\beta \to \infty$. The error is given as:

$$E_{rr} = \lim_{\beta\to\infty}\frac{1}{\beta}\frac{\partial}{\partial J}\mathfrak{g}[q, q_0, \hat{q}, \hat{q}_0] = \lim_{\beta\to\infty}\beta^{-1}q + q_0 - \zeta^2 = q_0 - \zeta^2 \tag{58}$$

Where $q, q_0, \hat{q}, \hat{q}_0$ are the solutions to the saddle-point equations 53, 54, 55, 57. Substituting eq. 57 for $q_0$, we obtain:

$$E_{rr} = \frac{1}{1-\gamma}\frac{1}{M}\boldsymbol{w}^{\star\top}\left[\boldsymbol{\Sigma}_s - \frac{2}{\nu_{rr}}\hat{q}\boldsymbol{\Sigma}_s\boldsymbol{A}_r^\top\boldsymbol{G}^{-1}\boldsymbol{A}_r\boldsymbol{\Sigma}_s + \frac{1}{\nu_{rr}}\hat{q}^2\boldsymbol{\Sigma}_s\boldsymbol{A}_r^\top\boldsymbol{G}^{-1}\tilde{\boldsymbol{\Sigma}}\boldsymbol{G}^{-1}\boldsymbol{A}_r\boldsymbol{\Sigma}_s\right]\boldsymbol{w}^* + \frac{\gamma\zeta^2 + \eta_r^2}{1-\gamma} \tag{59}$$

$$= \frac{1}{1-\gamma}\frac{1}{M}\boldsymbol{w}^{\star\top}\left[\boldsymbol{\Sigma}_s - \frac{1}{\nu_{rr}}\hat{q}\boldsymbol{\Sigma}_s\boldsymbol{A}_r^\top\boldsymbol{G}^{-1}\boldsymbol{A}_r\boldsymbol{\Sigma}_s - \frac{1}{\nu_{rr}}\hat{q}\boldsymbol{\Sigma}_s\boldsymbol{A}_r^\top\boldsymbol{G}^{-2}\boldsymbol{A}_r\boldsymbol{\Sigma}_s\right]\boldsymbol{w}^* + \frac{\gamma\zeta^2 + \eta_r^2}{1-\gamma} \tag{60}$$

## F.2 Off-Diagonal Terms

We now calculate $E_{rr'}$ for $r \neq r'$. We now must consider the joint Gibbs Measure over $\boldsymbol{w}_r$ and $\boldsymbol{w}_{r'}$:

$$Z = \int d\boldsymbol{w}_r d\boldsymbol{w}_{r'} \exp\left(-\frac{\beta}{2\lambda}(E_r^t + E_{r'}^t) - \frac{JM\beta}{2}E_{rr'}(\boldsymbol{w}_r, \boldsymbol{w}_{r'})\right) \tag{61}$$

$$\tag{62}$$

$$\langle Z^n \rangle_{\mathcal{D}} = \int \prod_a d\boldsymbol{w}_r^a d\boldsymbol{w}_{r'}^a \mathbb{E}_{\{\psi_\mu, \boldsymbol{\sigma}^\mu, \epsilon^\mu\}}$$
$$\exp\left(-\frac{\beta M}{2\lambda}\sum_{\mu,a}\frac{1}{M}\left[\left(h_\mu^{ra}\right)^2 + \left(h_\mu^{r'a}\right)^2\right] - \frac{\beta}{2}\sum_a \left[|\boldsymbol{w}_r^a|^2 + |\boldsymbol{w}_{r'}^a|^2\right] - \frac{JM\beta}{2}\sum_a E_{rr'}(\boldsymbol{w}_r^a, \boldsymbol{w}_{r'}^a)\right) \tag{63}$$

Where the $h_\mu^{ra}$ are defined as before. Next we must perform the averages over quenched disorder. To do so, we note that the $h_\mu^{ra}$ are Gaussian random variables with covariance structure:

$$\langle h_\mu^{ra} h_\nu^{r'b} \rangle = \delta_{\mu\nu} Q_{ab}^{rr'} \tag{64}$$

$$Q_{ab}^{rr'} = \frac{1}{M}\left[\left(\frac{1}{\sqrt{\nu_{rr}}}\boldsymbol{w}_r^{a\top}\boldsymbol{A}_r - \boldsymbol{w}^{*\top}\right)\boldsymbol{\Sigma}_s\left(\frac{1}{\sqrt{\nu_{r'r'}}}\boldsymbol{A}_{r'}^\top\boldsymbol{w}_{r'}^b - \boldsymbol{w}^*\right) \right.$$
$$\left. + \frac{1}{\sqrt{\nu_{rr}\nu_{r'r'}}}\boldsymbol{w}_r^{a\top}\boldsymbol{A}_r\boldsymbol{\Sigma}_0\boldsymbol{A}_{r'}^\top\boldsymbol{w}_{r'}^b + M\zeta^2\right] \tag{65}$$

To perform this integral we re-write in terms of $\{\boldsymbol{H}_\mu\}_{\mu=1}^P$, where

$$\boldsymbol{H}_\mu = \begin{bmatrix}\boldsymbol{H}_\mu^r \\ \boldsymbol{H}_\mu^{r'}\end{bmatrix} \in \mathbb{R}^{2n} \tag{66}$$

$$\langle Z^n \rangle_{\mathcal{D}} = \int \prod_a d\boldsymbol{w}_r^a d\boldsymbol{w}_{r'}^a \mathbb{E}_{\{\psi_\mu, \boldsymbol{\sigma}^\mu, \epsilon^\mu\}}$$
$$\exp\left(-\frac{\beta}{2\lambda}\sum_\mu \boldsymbol{H}_\mu^\top \boldsymbol{H}_\mu - \frac{\beta}{2}\sum_a \left[|\boldsymbol{w}_r^a|^2 + |\boldsymbol{w}_{r'}^a|^2\right] - \frac{JM\beta}{2}\sum_a E_{rr'}(\boldsymbol{w}_r^a, \boldsymbol{w}_{r'}^a)\right) \tag{67}$$

Integrating over $\boldsymbol{H}_\mu$ we get:

$$\langle Z^n \rangle_{\mathcal{D}} = \int \prod_a d\boldsymbol{w}_r^a d\boldsymbol{w}_{r'}^a$$
$$\exp\left(-\frac{P}{2}\log\det\left(\boldsymbol{I}_{2n} + \frac{\beta}{\lambda}\boldsymbol{Q}\right) - \frac{\beta}{2}\sum_a \left[|\boldsymbol{w}_r^a|^2 + |\boldsymbol{w}_{r'}^a|^2\right] - \frac{JM\beta}{2}\sum_a E_{rr}(\boldsymbol{w}_r^a, \boldsymbol{w}_{r'}^a)\right) \tag{68}$$

Where we have defined the matrix $\boldsymbol{Q}$ so that:

$$\boldsymbol{Q} = \begin{bmatrix}\boldsymbol{Q}^{rr} & \boldsymbol{Q}^{rr'} \\ \boldsymbol{Q}^{rr'} & \boldsymbol{Q}^{r'r'}\end{bmatrix} \tag{69}$$

Next we integrate over $\boldsymbol{Q}$ and add constraints. We use the following identity:

$$1 = \prod_{ab} \int dQ_{ab}^{rr'} \delta \left( Q_{ab}^{rr'} - \frac{1}{M} \left[ \left( \frac{1}{\sqrt{\nu_{rr}}} \boldsymbol{w}_r^{a\top} \boldsymbol{A}_r - \boldsymbol{w}^{*\top} \right) \boldsymbol{\Sigma}_s \left( \frac{1}{\sqrt{\nu_{r'r'}}} \boldsymbol{A}_{r'}^{\top} \boldsymbol{w}_{r'}^{b} - \boldsymbol{w}^{*} \right) \right. \right.$$
$$\left. \left. + \frac{1}{\sqrt{\nu_{rr}\nu_{r'r'}}} \boldsymbol{w}_r^{a\top} \boldsymbol{A}_r \boldsymbol{\Sigma}_0 \boldsymbol{A}_{r'}^{\top} \boldsymbol{w}_{r'}^{b} + M\zeta^2 \right] \right) \tag{70}$$

Using the Fourier representation of the delta function, we get:

$$1 = \prod_{ab} \int \frac{1}{4\pi i/M} dQ_{ab}^{rr'} d\hat{Q}_{ab}^{rr'} \exp \left( \frac{M}{2} \hat{Q}_{ab}^{rr'} \left( Q_{ab}^{rr'} - \frac{1}{M} \left[ \left( \frac{1}{\sqrt{\nu_{rr}}} \boldsymbol{w}_r^{a\top} \boldsymbol{A}_r - \boldsymbol{w}^{*\top} \right) \boldsymbol{\Sigma}_s \left( \frac{1}{\sqrt{\nu_{r'r'}}} \boldsymbol{A}_{r'}^{\top} \boldsymbol{w}_{r'}^{b} - \boldsymbol{w}^{*} \right) \right. \right. \right.$$
$$\left. \left. \left. + \frac{1}{\nu_{rr}} \boldsymbol{w}_r^{a\top} \boldsymbol{A}_r \boldsymbol{\Sigma}_0 \boldsymbol{A}_{r'}^{\top} \boldsymbol{w}_{r'}^{b} + M\zeta^2 \right] \right) \right) \tag{71}$$

Inserting this identity and the corresponding statements for $Q_{ab}^{rr}$ and $Q_{ab}^{r'r'}$ into the replicated partition function and substituting $E_{rr'}(\boldsymbol{w}^a) = Q_{aa}^{rr'} - \zeta^2$ we find:

$$\langle Z^n \rangle_{\mathcal{D}} \propto \int \prod_{abr_1r_2} dQ_{ab}^{r_1r_2} d\hat{Q}_{ab}^{r_1r_2}$$

$$\exp \left( -\frac{P}{2} \log \det \left( \boldsymbol{I}_{2n} + \frac{\beta}{\lambda} \boldsymbol{Q} \right) + \frac{1}{2} \sum_{abr_1r_2} M \hat{Q}_{ab}^{r_1r_2} Q_{ab}^{r_1r_2} - \frac{JM\beta}{2} \sum_a (Q_{aa}^{rr'} - \zeta^2) \right)$$

$$\int \prod_a d\boldsymbol{w}_r^a d\boldsymbol{w}_{r'}^a \exp \left( -\frac{\beta}{2} \sum_a \left[ |\boldsymbol{w}_r^a|^2 + |\boldsymbol{w}_{r'}^a|^2 \right] - \frac{1}{2} \sum_{abr_1r_2} \hat{Q}_{ab}^{r_1r_2} \left[ \left( \frac{1}{\sqrt{\nu_{r_1}}} \boldsymbol{w}_{r_1}^{a\top} \boldsymbol{A}_{r_1} - \boldsymbol{w}^{*\top} \right) \boldsymbol{\Sigma}_s \left( \frac{1}{\sqrt{\nu_{r_2}}} \boldsymbol{A}_{r_2}^{\top} \boldsymbol{w}_{r_2}^{b} - \boldsymbol{w}^{*} \right) \right. \right.$$

$$\left. \left. + \frac{1}{\sqrt{\nu_{r_1}\nu_{r_2}}} \boldsymbol{w}_{r_1}^{a\top} \boldsymbol{A}_{r_1} \boldsymbol{\Sigma}_0 \boldsymbol{A}_{r_2}^{\top} \boldsymbol{w}_{r_2}^{b} + M\zeta^2 \right] \right) \tag{72}$$

Where sums over $r_1$ and $r_2$ run over $\{r, r'\}$.

In order to perform the Gaussian integral over the $\{\boldsymbol{w}_r^a\}$, we unfold in two steps. We first define the following:

$$\boldsymbol{w}_r^{\cdot} \equiv \begin{bmatrix} \boldsymbol{w}_r^1 \\ \vdots \\ \boldsymbol{w}_r^n \end{bmatrix} \tag{73}$$

$$[\hat{\boldsymbol{Q}}^{rr'}]_{ab} \equiv \hat{Q}_{ab}^{rr'} \tag{74}$$

$$\tilde{\boldsymbol{\Sigma}}_{rr'} \equiv \frac{1}{\sqrt{\nu_{rr}\nu_{r'r'}}} \boldsymbol{A}_r [\boldsymbol{\Sigma}_s + \boldsymbol{\Sigma}_0] \boldsymbol{A}_{r'}^{\top} \tag{75}$$

$$T^{rr'} \equiv \beta \delta_{rr'} \boldsymbol{I}_n \otimes \boldsymbol{I}_{N_r} + \hat{\boldsymbol{Q}}^{rr'} \otimes \tilde{\boldsymbol{\Sigma}}_{rr'} \tag{76}$$

Unfolding over the replica indices, we then get:

$$\langle Z^n \rangle_{\mathcal{D}} \propto \int \prod_{abr_1r_2} dQ_{ab}^{r_1r_2} d\hat{Q}_{ab}^{r_1r_2}$$

$$\exp\left(-\frac{P}{2}\log\det\left(\boldsymbol{I}_{2n} + \frac{\beta}{\lambda}\boldsymbol{Q}\right) + \frac{1}{2}\sum_{abr_1r_2} M\hat{Q}_{ab}^{r_1r_2}Q_{ab}^{r_1r_2} - \frac{JM\beta}{2}\sum_a (Q_{aa}^{rr'} - \zeta^2)\right)$$

$$\exp\left(-\frac{1}{2}\sum_{abr_1r_2} \hat{Q}_{ab}^{r_1r_2}\left(\boldsymbol{w}^{*\top}\boldsymbol{\Sigma}_s\boldsymbol{w}^* + M\zeta^2\right)\right)$$

$$\int d\boldsymbol{w}_r^{\cdot} d\boldsymbol{w}_{r'}^{\cdot} \exp\left(-\frac{1}{2}\sum_{r_1r_2} \boldsymbol{w}_{r_1}^{\cdot\top} T^{r_1r_2}\boldsymbol{w}_{r_2} + \sum_{r_1r_2}\left[(\hat{\boldsymbol{Q}}^{r_1r_2}\otimes\boldsymbol{I}_{N_{r_1}})(\boldsymbol{1}_n\otimes\frac{1}{\sqrt{\nu_{r_1}}}\boldsymbol{A}_{r_1}\boldsymbol{\Sigma}_s\boldsymbol{w}^*)\right]^{\top}\boldsymbol{w}_{r_1}\right)$$

$$\tag{77}$$

Note that the dimensionality of $T^{r_1r_2}$ varies for different choices of $r_1$ and $r_2$. Next, we unfold over the two readouts:

$$\boldsymbol{w} \equiv \begin{bmatrix} \boldsymbol{w}_r^{\cdot} \\ \boldsymbol{w}_{r'}^{\cdot} \end{bmatrix} \tag{78}$$

$$T \equiv \begin{bmatrix} T^{rr} & T^{rr'} \\ T^{r'r} & T^{r'r'} \end{bmatrix} \tag{79}$$

$$V \equiv \begin{bmatrix} \left((\hat{\boldsymbol{Q}}^{rr} + \hat{\boldsymbol{Q}}^{rr'})\otimes\boldsymbol{I}_{N_r}\right)\left(\boldsymbol{1}_n\otimes\frac{1}{\sqrt{\nu_{rr}}}\boldsymbol{A}_r\boldsymbol{\Sigma}_s\boldsymbol{w}^*\right) \\ \left((\hat{\boldsymbol{Q}}^{r'r'} + \hat{\boldsymbol{Q}}^{r'r})\otimes\boldsymbol{I}_{N_{r'}}\right)\left(\boldsymbol{1}_n\otimes\frac{1}{\sqrt{\nu_{r'r'}}}\boldsymbol{A}_{r'}\boldsymbol{\Sigma}_s\boldsymbol{w}^*\right) \end{bmatrix} \tag{80}$$

The integral over w then becomes:

$$\int d\boldsymbol{w}\exp\left(-\frac{1}{2}\boldsymbol{w}^{\top}T\boldsymbol{w} + V^{\top}\boldsymbol{w}\right) \propto \exp\left(\frac{1}{2}V^{\top}T^{-1}V - \frac{1}{2}\log\det T\right) \tag{81}$$

We are now ready to make a replica-symmetric ansatz. The order parameter that we wish to constrain is $Q_{ab}^{rr'}$. Overlaps go between the weights from different replicas of the system as well as different readouts. The scale of the overlap between two measurements depends on their overlap with each other and with the principal components of the data distribution. An ansatz which is replica-symmetric but makes no assumptions about the overlaps between different measurements is as follows:

$$Q_{ab}^{r_1r_2} = \beta^{-1}q^{r_1r_2}\delta_{ab} + Q^{r_1r_2} \tag{82}$$

$$\hat{Q}_{ab}^{r_1r_2} = \beta\hat{q}^{r_1r_2}\delta_{ab} + \beta^2\hat{Q}^{r_1r_2} \tag{83}$$

Next step is to plug the RS ansatz into the free energy and simplify. To make calculations more transparent, we re-label the paramters in the RS ansatz as follows:

$$\boldsymbol{Q}^{rr} = \beta^{-1}q\boldsymbol{I} + Q\boldsymbol{1}\boldsymbol{1}^{\top} \tag{84}$$

$$\boldsymbol{Q}^{r'r'} = \beta^{-1}r\boldsymbol{I} + R\boldsymbol{1}\boldsymbol{1}^{\top} \tag{85}$$

$$\boldsymbol{Q}^{rr'} = \beta^{-1}c\boldsymbol{I} + C\boldsymbol{1}\boldsymbol{1}^{\top} \tag{86}$$

$$\hat{\boldsymbol{Q}}^{rr} = \beta\hat{q}\boldsymbol{I} + \beta^2\hat{Q}\boldsymbol{1}\boldsymbol{1}^{\top} \tag{87}$$

$$\hat{\boldsymbol{Q}}^{r'r'} = \beta\hat{r}\boldsymbol{I} + \beta^2\hat{R}\boldsymbol{1}\boldsymbol{1}^{\top} \tag{88}$$

$$\hat{\boldsymbol{Q}}^{rr'} = \beta\hat{c}\boldsymbol{I} + \beta^2\hat{C}\boldsymbol{1}\boldsymbol{1}^{\top} \tag{89}$$

In order to simplify $\log \det \left(\lambda \boldsymbol{I}_{2n} + \beta \boldsymbol{Q}\right)$, we note that this is a symmetric 2-by-2-block matrix, where each block commutes with all other blocks. We may then use [53]'s result to simplify.

$$\log \det \left(\lambda \boldsymbol{I}_{2n} + \beta \boldsymbol{Q}\right) = n \left[\log \left((\lambda + q)(\lambda + r) - c^2\right) + \beta \frac{(\lambda + q)R + (\lambda + r)Q - 2cC}{(\lambda + q)(\lambda + r) - c^2}\right] + \mathcal{O}(n^2) \tag{90}$$

$$\sum_{abr_1r_2} \hat{\boldsymbol{Q}}_{ab}^{r_1r_2} \boldsymbol{Q}_{ab}^{r_1r_2} = n \left[\left(q\hat{q} + \beta \hat{q}Q + \beta q\hat{Q}\right) + \left(r\hat{r} + \beta \hat{r}R + \beta r\hat{R}\right) + 2\left(c\hat{c} + \beta \hat{c}C + \beta c\hat{C}\right)\right] + \mathcal{O}(n^2) \tag{91}$$

$$\sum_a \left(\boldsymbol{Q}_{aa}^{rr'} - \zeta^2\right) = n \left[\frac{1}{\beta}c + C - \zeta^2\right] + \mathcal{O}(n^2) \tag{92}$$

$$\sum_{abr_1r_2} \hat{\boldsymbol{Q}}_{ab}^{r_1r_2} = \beta n \left[\hat{q} + \hat{r} + 2\hat{c}\right] + \mathcal{O}(n^2) \tag{93}$$

$$\log \det(T) = n \left[\log(\beta) + \log \det \begin{bmatrix} \boldsymbol{G}_{rr} & \boldsymbol{G}_{rr'} \\ \boldsymbol{G}_{r'r} & \boldsymbol{G}_{r'r'} \end{bmatrix} + \beta \operatorname{tr} \left(\begin{bmatrix} \boldsymbol{G}_{rr} & \boldsymbol{G}_{rr'} \\ \boldsymbol{G}_{r'r} & \boldsymbol{G}_{r'r'} \end{bmatrix}^{-1} \begin{bmatrix} \hat{Q}\tilde{\boldsymbol{\Sigma}}_{rr} & \hat{C}\tilde{\boldsymbol{\Sigma}}_{rr'} \\ \hat{C}\tilde{\boldsymbol{\Sigma}}_{r'r} & \hat{R}\tilde{\boldsymbol{\Sigma}}_{r'r'} \end{bmatrix}\right)\right] + \mathcal{O}(n^2) \tag{94}$$

where $\boldsymbol{G}_{rr} = \boldsymbol{I}_{N_r} + \hat{q}\tilde{\boldsymbol{\Sigma}}_{rr}$ $\quad \boldsymbol{G}_{r'r'} = \boldsymbol{I}_{N_{r'}} + \hat{r}\tilde{\boldsymbol{\Sigma}}_{r'r'}$ $\quad \boldsymbol{G}_{rr'} = \hat{c}\tilde{\boldsymbol{\Sigma}}_{rr'}$ $\quad \boldsymbol{G}_{r'r} = \hat{c}\tilde{\boldsymbol{\Sigma}}_{r'r}$ $\tag{95}$

$$V^{\top}T^{-1}V = n\beta \boldsymbol{w}^{*\top} \begin{bmatrix} \frac{1}{\sqrt{\nu_{rr}}}(\hat{q} + \hat{c})\boldsymbol{A}_r\boldsymbol{\Sigma}_s \\ \frac{1}{\sqrt{\nu_{r'r'}}}(\hat{r} + \hat{c})\boldsymbol{A}_{r'}\boldsymbol{\Sigma}_s \end{bmatrix}^{\top} \begin{bmatrix} \boldsymbol{G}_{rr} & \boldsymbol{G}_{rr'} \\ \boldsymbol{G}_{r'r} & \boldsymbol{G}_{r'r'} \end{bmatrix}^{-1} \begin{bmatrix} \frac{1}{\sqrt{\nu_{rr}}}(\hat{q} + \hat{c})\boldsymbol{A}_r\boldsymbol{\Sigma}_s \\ \frac{1}{\sqrt{\nu_{r'r'}}}(\hat{r} + \hat{c})\boldsymbol{A}_{r'}\boldsymbol{\Sigma}_s \end{bmatrix} \boldsymbol{w}^* + \mathcal{O}(n^2) \tag{96}$$

Collecting these terms, we may write the replicated partition function as follows:

$$\langle Z^n \rangle_{\mathcal{D}} = \exp \left(-\frac{nM}{2}\mathfrak{g}\left[q, Q, r, R, c, C, \hat{q}, \hat{Q}, \hat{r}, \hat{R}, \hat{c}, \hat{C}\right]\right) \tag{97}$$

Where the free energy is written:

$$\mathfrak{g}\left[q, Q, r, R, c, C, \hat{q}, \hat{Q}, \hat{r}, \hat{R}, \hat{c}, \hat{C}\right] = \tag{98}$$

$$\alpha \left[\log \left((\lambda + q)(\lambda + r) - c^2\right) + \beta \frac{(\lambda + q)R + (\lambda + r)Q - 2cC}{(\lambda + q)(\lambda + r) - c^2}\right] \tag{99}$$

$$- \left[\left(q\hat{q} + \beta \hat{q}Q + \beta q\hat{Q}\right) + \left(r\hat{r} + \beta \hat{r}R + \beta r\hat{R}\right) + 2\left(c\hat{c} + \beta \hat{c}C + \beta c\hat{C}\right)\right] \tag{100}$$

$$+ J(c + \beta C - \beta \zeta^2) \tag{101}$$

$$+ \beta \left[\hat{q} + \hat{r} + 2\hat{c}\right]\left(\frac{1}{M}\boldsymbol{w}^{*\top}\boldsymbol{\Sigma}\boldsymbol{w}^* + \zeta^2\right) \tag{102}$$

$$- \frac{1}{M}\beta \boldsymbol{w}^{*\top} \begin{bmatrix} \frac{1}{\sqrt{\nu_{rr}}}(\hat{q} + \hat{c})\boldsymbol{A}_r\boldsymbol{\Sigma}_s \\ \frac{1}{\sqrt{\nu_{r'r'}}}(\hat{r} + \hat{c})\boldsymbol{A}_{r'}\boldsymbol{\Sigma}_s \end{bmatrix}^{\top} \boldsymbol{G}^{-1} \begin{bmatrix} \frac{1}{\sqrt{\nu_{rr}}}(\hat{q} + \hat{c})\boldsymbol{A}_r\boldsymbol{\Sigma}_s \\ \frac{1}{\sqrt{\nu_{r'r'}}}(\hat{r} + \hat{c})\boldsymbol{A}_{r'}\boldsymbol{\Sigma}_s \end{bmatrix} \boldsymbol{w}^* \tag{103}$$

$$+ \frac{1}{M}\left[\log(\beta) + \log \det \boldsymbol{G} + \beta \operatorname{tr}\left(\boldsymbol{G}^{-1} \begin{bmatrix} \hat{Q}\tilde{\boldsymbol{\Sigma}}_{rr} & \hat{C}\tilde{\boldsymbol{\Sigma}}_{rr'} \\ \hat{C}\tilde{\boldsymbol{\Sigma}}_{r'r} & \hat{R}\tilde{\boldsymbol{\Sigma}}_{r'r'} \end{bmatrix}\right)\right] \tag{104}$$

where we have defined $G \equiv \begin{bmatrix} G_{rr} & G_{rr'} \\ G_{r'r} & G_{r'r'} \end{bmatrix}$

The saddle-point equations for the replica-diagonal order parameters are:

$$\frac{\partial \mathfrak{g}}{\partial Q} = 0 = \beta \frac{\alpha(\lambda + r)}{(\lambda + q)(\lambda + r) - c^2} - \beta \hat{q} \tag{105}$$

$$\frac{\partial \mathfrak{g}}{\partial \hat{Q}} = 0 = -\beta q + \beta \frac{1}{M} \operatorname{tr} \left( G^{-1} \begin{bmatrix} \tilde{\Sigma}_{rr} & 0 \\ 0 & 0 \end{bmatrix} \right) \tag{106}$$

$$\frac{\partial \mathfrak{g}}{\partial R} = 0 = \beta \frac{\alpha(\lambda + q)}{(\lambda + q)(\lambda + r) - c^2} - \beta \hat{r} \tag{107}$$

$$\frac{\partial \mathfrak{g}}{\partial \hat{R}} = 0 = -\beta r + \beta \frac{1}{M} \operatorname{tr} \left( G^{-1} \begin{bmatrix} 0 & 0 \\ 0 & \tilde{\Sigma}_{r'r'} \end{bmatrix} \right) \tag{108}$$

$$\frac{\partial \mathfrak{g}}{\partial C} = 0 = -\beta \frac{2\alpha c}{(\lambda + q)(\lambda + r) - c^2} - 2\beta \hat{c} + \beta J \tag{109}$$

$$\frac{\partial \mathfrak{g}}{\partial \hat{C}} = 0 = -2\beta c + \beta \frac{1}{M} \operatorname{tr} \left( G^{-1} \begin{bmatrix} 0 & \tilde{\Sigma}_{rr'} \\ \tilde{\Sigma}_{r'r} & 0 \end{bmatrix} \right) \tag{110}$$

Note that when $J = 0$, the saddle point equations 109, 110 are solved by setting $c = \hat{c} = 0$, and in this case the remaining saddle-point equations decouple over the readouts (as expected for independently trained ensemble members) giving: For readout $r$:

$$0 = \frac{\alpha}{(\lambda + q)} - \hat{q} \tag{111}$$

$$0 = -q + \frac{1}{M} \operatorname{tr} \left( G_{rr}^{-1} \tilde{\Sigma}_{rr} \right) \tag{112}$$

and for readout $r'$:

$$0 = \frac{\alpha}{(\lambda + r)} - \hat{r} \tag{113}$$

$$0 = -r + \frac{1}{M} \operatorname{tr} \left( G_{r'r'}^{-1} \tilde{\Sigma}_{r'r'} \right) \tag{114}$$

These are equivalent to the saddle-point equations for a single readout given in equation 53, 54 as expected for independently trained readouts. It is physically sensible that $c = 0$ when $J = 0$, because at zero source, there is no term in the replicated system energy function which would distinguish the overlap between two readouts from the same replica from the overlap between two readouts in separate replicas (we expect that the total overlap between readouts is non-zero, as we may still have $C > 0$).

The saddle-point equations obtained by setting the derivatives $\frac{\partial \mathfrak{g}}{\partial q} = \frac{\partial \mathfrak{g}}{\partial \hat{q}} = \frac{\partial \mathfrak{g}}{\partial r} = \frac{\partial \mathfrak{g}}{\partial \hat{r}} = 0$ will similarly decouple to recover two copies of the diagonal case 57 55. We will not re-write the expressions here as they are not necessary to determine the off-diagonal error term $E_{rr'}$.

The remaining saddle-point equations are obtained by setting $\frac{\partial \mathfrak{g}}{\partial c} = \frac{\partial \mathfrak{g}}{\partial \hat{c}} = 0$

$$\frac{\partial \mathfrak{g}}{\partial c}\bigg|_{c=\hat{c}=J=0} = -\frac{2\alpha\beta C}{(\lambda + q)(\lambda + r)} - 2\beta\hat{C} \quad \Rightarrow \quad \hat{C} = -\frac{\alpha C}{(\lambda + q)(\lambda + r)} \tag{115}$$

$$\frac{\partial \mathfrak{g}}{\partial \hat{c}}\bigg|_{c=\hat{c}=J=0} = 0 = -2\beta C + 2\beta \left( \frac{1}{M} \boldsymbol{w}^{*\top} \boldsymbol{\Sigma}_s \boldsymbol{w}^* + \zeta^2 \right)$$
$$- \frac{2\beta}{M} \boldsymbol{w}^{*\top} \boldsymbol{\Sigma}_s \left[ \frac{1}{\nu_{rr}} \hat{q} \boldsymbol{A}_r^\top \boldsymbol{G}_{rr}^{-1} \boldsymbol{A}_r + \frac{1}{\nu_{r'r'}} \hat{r} \boldsymbol{A}_{r'}^\top \boldsymbol{G}_{r'r'}^{-1} \boldsymbol{A}_{r'} \right] \boldsymbol{\Sigma}_s \boldsymbol{w}^*$$
$$+ \frac{2\beta \hat{q} \hat{r}}{M} \frac{1}{\sqrt{\nu_{rr} \nu_{r'r'}}} \boldsymbol{w}^{*\top} \boldsymbol{\Sigma}_s \boldsymbol{A}_r^\top \boldsymbol{G}_{rr}^{-1} \tilde{\boldsymbol{\Sigma}}_{rr'} \boldsymbol{G}_{r'r'}^{-1} \boldsymbol{A}_{r'} \boldsymbol{\Sigma}_s \boldsymbol{w}^* \tag{116}$$
$$- \frac{2\hat{C}\beta}{M} \operatorname{tr} \left[ \boldsymbol{G}_{rr}^{-1} \tilde{\boldsymbol{\Sigma}}_{rr'} \boldsymbol{G}_{r'r'}^{-1} \tilde{\boldsymbol{\Sigma}}_{r'r} \right]$$

Solving equations 115 and 116 for $C$, we obtain:

$$C = \frac{1}{1-\gamma} \zeta^2 + \frac{1}{1-\gamma} \left( \frac{1}{M} \boldsymbol{w}^{*\top} \boldsymbol{\Sigma}_s \boldsymbol{w}^* \right)$$
$$- \frac{1}{M(1-\gamma)} \boldsymbol{w}^{*\top} \boldsymbol{\Sigma}_s \left[ \frac{1}{\nu_{rr}} \hat{q} \boldsymbol{A}_r^\top \boldsymbol{G}_{rr}^{-1} \boldsymbol{A}_r + \frac{1}{\nu_{r'r'}} \hat{r} \boldsymbol{A}_{r'}^\top \boldsymbol{G}_{r'r'}^{-1} \boldsymbol{A}_{r'} \right] \boldsymbol{\Sigma}_s \boldsymbol{w}^* \tag{117}$$
$$+ \frac{1}{M(1-\gamma)} \hat{q} \hat{r} \frac{1}{\sqrt{\nu_{rr} \nu_{r'r'}}} \boldsymbol{w}^{*\top} \boldsymbol{\Sigma}_s \boldsymbol{A}_r^\top \boldsymbol{G}_{rr}^{-1} \tilde{\boldsymbol{\Sigma}}_{rr'} \boldsymbol{G}_{r'r'}^{-1} \boldsymbol{A}_{r'} \boldsymbol{\Sigma}_s \boldsymbol{w}^*$$

$$\text{where} \quad \gamma \equiv \frac{\alpha}{(\lambda+q)(\lambda+r)} \operatorname{tr} \left[ \boldsymbol{G}_{rr}^{-1} \tilde{\boldsymbol{\Sigma}}_{rr'} \boldsymbol{G}_{r'r'}^{-1} \tilde{\boldsymbol{\Sigma}}_{r'r} \right] \tag{118}$$

We can obtain the generalization error as:

$$E_{rr'} = \lim_{\beta \to \infty} \frac{1}{\beta} \frac{\partial}{\partial J} \mathfrak{g} \left[ q, Q, r, R, c, C, \hat{q}, \hat{Q}, \hat{r}, \hat{R}, \hat{c}, \hat{C} \right] = \lim_{\beta \to \infty} \frac{1}{\beta} (c + \beta C - \beta \zeta^2) = C - \zeta^2 \tag{119}$$

We may then simplify the expression for the $E_{rr'}$ error as follows:

$$E_{rr'} = \frac{\gamma}{1-\gamma} \zeta^2 + \frac{1}{1-\gamma} \left( \frac{1}{M} \boldsymbol{w}^{*\top} \boldsymbol{\Sigma}_s \boldsymbol{w}^* \right)$$
$$- \frac{1}{M(1-\gamma)} \boldsymbol{w}^{*\top} \boldsymbol{\Sigma}_s \left[ \frac{1}{\nu_{rr}} \hat{q} \boldsymbol{A}_r^\top \boldsymbol{G}_{rr}^{-1} \boldsymbol{A}_r + \frac{1}{\nu_{r'r'}} \hat{r} \boldsymbol{A}_{r'}^\top \boldsymbol{G}_{r'r'}^{-1} \boldsymbol{A}_{r'} \right] \boldsymbol{\Sigma}_s \boldsymbol{w}^* \tag{120}$$
$$+ \frac{1}{M(1-\gamma)} \hat{q} \hat{r} \frac{1}{\sqrt{\nu_{rr} \nu_{r'r'}}} \boldsymbol{w}^{*\top} \boldsymbol{\Sigma}_s \boldsymbol{A}_r^\top \boldsymbol{G}_{rr}^{-1} \tilde{\boldsymbol{\Sigma}}_{rr'} \boldsymbol{G}_{r'r'}^{-1} \boldsymbol{A}_{r'} \boldsymbol{\Sigma}_s \boldsymbol{w}^*$$

Re-labeling the order parameters: $\hat{q} \to \hat{q}_r$, $\hat{r} \to \hat{q}_{r'}$, $\gamma \to \gamma_{rr'}$ and $\boldsymbol{G}_{rr} \to \boldsymbol{G}_r$, we obtain the result given in the main text.

# G   Derivation of Proposition 1 from [31]

In the case where the data and noise covariance matrices $\boldsymbol{\Sigma}_s$ and $\boldsymbol{\Sigma}_0$ have bounded spectra, our main result may be derived using Theorem 4.1 from Loureiro et. al. [31], with a few additional arguments to incorporate a readout noise which varies across ensemble members, and to allow for the presence of label noise in the training set but not at test time. Rather than reproducing their very lengthy statements here, we direct the reader to theorem 4.1 and corollary 4.2 in [31].

## G.1 Altered Expectations at Evaluation

In [31], labels are generated at both training and test time through the same statistical process $y \sim P_y^0(y|\nu)$ where $\nu = \frac{\langle x|\theta \rangle}{\sqrt{d}}$. However, their results may be easily extended to the case where data are generated through a different statistical process for the training set and at evaluation. This will allow the application of their results to the case where label noise is present during training but not at evaluation as studied in this work. We therefore introduce separate distributions of the labels $y$ at training and evaluation. During training, we still have $y \sim P_y^0(y|\nu)$. At evaluation, we put $y \sim P_y^g(y|\nu)$. This leads to the updated formula:

$$\mathbb{E}_{(y,x)} \left[ \varphi \left( y, \frac{\langle\langle \hat{W} \mid U \rangle\rangle}{\sqrt{p}} \right) \right] \xrightarrow{\text{P}} \mathbb{E}_{(\nu,\mu)} \left[ \int dy P_y^g(y|\nu) \varphi(y, \mu) \right] \tag{121}$$

when the expectation is over data-label pairs $(y, x)$ at *evaluation*.

## G.2 Rigorous Proof of Proposition 1

We now restate the the problem setup of our main theorem using notation consistent with [31]. We study generalization error in an ensemble of estimators $\{\hat{w}_k\}$, $k = 1, \ldots, K$. We say $\hat{w}_k \in \mathbb{R}^p$ for all $k = 1, \ldots, K$. The weights are trained independently such that each minimizes a ridge loss function:

$$\hat{w}_k = \arg\min_w \sum_{\mu=1}^n \left( y^\mu - \frac{1}{\sqrt{N_k}} u_k^\top w - \xi_k^\mu \right)^2 + \lambda_k |w|^2 \qquad k = 1, \ldots, K$$

$$= \arg\min_w \sum_{\mu=1}^n \left( y^\mu - \frac{1}{\sqrt{p}} u_k^\top \left( \frac{1}{\sqrt{\nu_{kk}}} w \right) - \xi_k^\mu \right)^2 + \nu_{kk} \lambda_k \left| \frac{1}{\sqrt{\nu_{kk}}} w \right|^2 \tag{122}$$

So that our results will correspond to the results of [31] after re-scaling the regularizations $\lambda_k \to \nu_{kk} \lambda_k$. The training labels are drawn as $y^\mu \sim P_y^0(y|\frac{1}{\sqrt{p}} \theta^\top x^\mu)$ where $P_y^0(y|x) = \mathcal{N}(x, \zeta^2)$, $\xi_k^\mu \sim \mathcal{N}(0, \eta_r^2)$ and $\theta$ represent the ground truth weights. For $k = 1, \ldots, k$ we have a "measurement matrix" $A_k \in \mathbb{R}^{N_k \times d}$, and we may set $d = p \geq \max_k N_k$ (so that $\gamma = \frac{d}{p} = 1$). The feature vectors $u_k(x)$ are then drawn as

$$u_k(x) = \begin{bmatrix} A_k(x + \sigma) \\ 0_{(p-N_k)} \end{bmatrix} = \bar{A}_k(x + \sigma)$$

where $x \sim \mathcal{N}(0, \Sigma_s)$ and $\sigma \sim \mathcal{N}(0, \Sigma_0)$. For convenience, we have defined the auxilary matrices

$$\bar{A}_k \equiv \begin{bmatrix} A_k \\ 0_{(p-N_k) \times p} \end{bmatrix} \in \mathbb{R}^{p \times p}$$

By constructing the feature vectors as p-dimensional vectors with only $N_k$ non-zero components, we may apply the results of [31] while preserving structural heterogeneity (we may have $N_k \neq N_{k'}$ for $k \neq k'$) as is present in our main result. Because $[u_k]_i = 0$ for all $i > N_k$, these auxiliary dimensions will not affect model predictions or generalization error. We then have $\mu_k = \frac{1}{\sqrt{p}} u_k^\top \hat{w}_k$, and labels are generated at evaluation according to :

$$y \sim P_y^g(y|\frac{1}{K} \sum_k \mu_k) \text{ where } P_y^g(y|x) = \mathcal{N}(x, \frac{1}{K^2} \sum_k \eta_k^2) \tag{123}$$

The generalization error may then be decomposed as $E_g = \frac{1}{K^2} \sum_{k,k'} E_{kk'}$ where $E_{kk'} = \mathbb{E}_{(y,x)} [(y - \mu_k)(y - \mu_{k'})]$. We will apply eq. 121 separately to calculate $E_{rr'}$ in the cases where $r \neq r'$ and $r = r'$.

### G.2.1 Off-Diagonal Terms

Eq. 121 cannot be used to calculate $E_{rr'}$ directly in the case where $r \neq r'$ due to the presence of noises $\xi_k^\mu$ which vary over elements of the ensemble (indexed by $k$) in the loss function (eq. 122). We may argue, however, that the presence of readout noise noise has no effect on the expected value of $E_{rr'}$ when $r \neq r'$, then apply eq. 121 with $\eta_k = \eta_{k'} = 0$. To see this, we examine the analytical form of the minimizer of the loss function (eq, 122):

$$\hat{\boldsymbol{w}}_k = \boldsymbol{U}_k^\top \left( \boldsymbol{U}_k \boldsymbol{U}_k^\top + \lambda_k \boldsymbol{I} \right)^{-1} (\boldsymbol{y} - \boldsymbol{\xi}_k) \tag{124}$$

where we have defined the design matrices $[\boldsymbol{U}_k]_{i\mu} = \frac{1}{\sqrt{p}}[\boldsymbol{u}_k(\boldsymbol{x}^\mu)]_i$ and the vectors $[\boldsymbol{y}]_\mu = y^\mu$ and $[\boldsymbol{\xi}_k]_\mu = \xi_k^\mu$. We then have

$$
\begin{aligned}
E_{kk'} =& \left( y - \frac{1}{\sqrt{p}} \boldsymbol{y}^\top \left( \boldsymbol{U}_k \boldsymbol{U}_k^\top + \lambda_k \boldsymbol{I} \right)^{-1} \boldsymbol{U}_k \right) \left( y - \frac{1}{\sqrt{p}} \boldsymbol{y}^\top \left( \boldsymbol{U}_{k'} \boldsymbol{U}_{k'}^\top + \lambda_{k'} \boldsymbol{I} \right)^{-1} \boldsymbol{U}_{k'} \right) \\
&+ \left( y - \frac{1}{\sqrt{p}} \boldsymbol{y}^\top \left( \boldsymbol{U}_k \boldsymbol{U}_k^\top + \lambda_k \boldsymbol{I} \right)^{-1} \boldsymbol{U}_k \right) \left( \frac{1}{\sqrt{p}} \boldsymbol{\xi}_{k'}^\top \left( \boldsymbol{U}_{k'} \boldsymbol{U}_{k'}^\top + \lambda_{k'} \boldsymbol{I} \right)^{-1} \boldsymbol{U}_{k'} \right) \\
&+ \left( y - \frac{1}{\sqrt{p}} \boldsymbol{y}^\top \left( \boldsymbol{U}_{k'} \boldsymbol{U}_{k'}^\top + \lambda_{k'} \boldsymbol{I} \right)^{-1} \boldsymbol{U}_{k'} \right) \left( \frac{1}{\sqrt{p}} \boldsymbol{\xi}_k^\top \left( \boldsymbol{U}_k \boldsymbol{U}_k^\top + \lambda_k \boldsymbol{I} \right)^{-1} \boldsymbol{U}_k \right) \\
&+ \left( \frac{1}{\sqrt{p}} \boldsymbol{\xi}_k^\top \left( \boldsymbol{U}_k \boldsymbol{U}_k^\top + \lambda_k \boldsymbol{I} \right)^{-1} \boldsymbol{U}_k \right) \left( \frac{1}{\sqrt{p}} \boldsymbol{\xi}_{k'}^\top \left( \boldsymbol{U}_{k'} \boldsymbol{U}_{k'}^\top + \lambda_{k'} \boldsymbol{I} \right)^{-1} \boldsymbol{U}_{k'} \right)
\end{aligned} \tag{125}
$$

Taking the expectation value over the readout noise in the training set we get:

$$\mathbb{E}_{\{\boldsymbol{\xi}_1,\ldots,\boldsymbol{\xi}_K\}} E_{kk'} = \left( y - \frac{1}{\sqrt{p}} \boldsymbol{y}^\top \left( \boldsymbol{U}_k \boldsymbol{U}_k^\top + \lambda_k \boldsymbol{I} \right)^{-1} \boldsymbol{U}_k \right) \left( y - \frac{1}{\sqrt{p}} \boldsymbol{y}^\top \left( \boldsymbol{U}_{k'} \boldsymbol{U}_{k'}^\top + \lambda_{k'} \boldsymbol{I} \right)^{-1} \boldsymbol{U}_{k'} \right) \quad (k \neq k') \tag{126}$$

This is identical to $E_{kk'}$ when $\boldsymbol{\xi}_k = 0$ for all $k$. We may therefore calculate the off-diagonal error terms by setting $\boldsymbol{\xi}_k = 0$ in eq. 122, which gives a problem compatible with the theorem 4 of [31]. To calculate $E_{rr'}$, we appeal to theorem 4.1 and corollary 4.2 of [31]. The following objects defined in 121 are given the following definitions:

$$r(\{\hat{\boldsymbol{w}}_1, \ldots, \hat{\boldsymbol{w}}_K\}) = \frac{1}{2} \sum_{k=1}^K \nu_{kk} \lambda_k |\hat{\boldsymbol{w}}_k|^2 \tag{127}$$

$$\Delta(y, \hat{y}(\boldsymbol{x})) = (y - \hat{y}(\boldsymbol{x}))^2 \tag{128}$$

$$\hat{\ell}(y, \boldsymbol{\mu}) = \frac{1}{2} |\boldsymbol{\mu} - y\mathbf{1}|^2 \tag{129}$$

$$\mathcal{Z}^0(y, \mu, \sigma) := \int \frac{P_y^0(y \mid x) \mathrm{d}x}{\sqrt{2\pi\sigma}} \mathrm{e}^{-\frac{(x-\mu)^2}{2\sigma}} = \frac{1}{\sqrt{2\pi(\zeta^2 + \sigma)}} \exp\left( -\frac{(y-\mu)^2}{2(\zeta^2 + \sigma)} \right) \tag{130}$$

$$E_{kk'} = \mathbb{E}_{(y,\boldsymbol{x})} \left[ (y - \mu_k)(y - \mu_{k'}) \right] \to \mathbb{E}_{(\nu,\boldsymbol{x})} \left[ \int dy P_y^g(y|\nu)(y - \mu_k)(y - \mu_{k'}) \right] \tag{131}$$

$$= \mathbb{E}_{(\nu,\boldsymbol{x})} \left[ (\nu - \mu_k)(\nu - \mu_{k'}) \right] + \frac{1}{K^2} \sum_{k=1}^K \eta_k^2 \tag{132}$$

$$= \rho - [\boldsymbol{m}]_k - [\boldsymbol{m}]_{k'} + [\boldsymbol{Q}]_{kk'} + \frac{1}{K^2} \sum_{k=1}^K \eta_k^2 \tag{133}$$

Where $\rho$, $\boldsymbol{m}$ and $\boldsymbol{Q}$ are defined as in [31]:

$$(\nu, \boldsymbol{\mu}) \sim \mathcal{N}\left( \mathbf{0}_{1+K}, \begin{pmatrix} \rho & \boldsymbol{m}^\top \\ \boldsymbol{m} & \boldsymbol{Q} \end{pmatrix} \right) \tag{134}$$

What remains is to determine the values of the order parameters $\rho$, $\boldsymbol{m}$ and $\boldsymbol{Q}$. We next define the covariance matrices which characterize the feature maps. The feature-feature covariance is:

$$\boldsymbol{\Omega}_{kk'}^{ij} = \mathbb{E}_{\boldsymbol{x}}\left[(\boldsymbol{u}_k(\boldsymbol{x}))_i(\boldsymbol{u}_{k'}(\boldsymbol{x}))_j\right] = \mathbb{1}_{\{1\leq i\leq N_k, 1\leq j\leq N_{k'}\}}\left[\boldsymbol{A}_k\left(\boldsymbol{\Sigma}_s + \boldsymbol{\Sigma}_0\right)\boldsymbol{A}_{k'}^\top\right]_{ij} \tag{135}$$

$$= \sqrt{\nu_{kk}\nu_{k'k'}}\,\mathbb{1}_{\{1\leq i\leq N_k, 1\leq j\leq N_{k'}\}}\left[\tilde{\boldsymbol{\Sigma}}_{kk'}\right]_{ij} = \left[\bar{\boldsymbol{\Sigma}}_{kk'}\right]_{ij} \tag{136}$$

where we have defined $\nu_{kk} = \frac{N_k}{p}$, $\tilde{\boldsymbol{\Sigma}}_{kk'} = \frac{1}{\sqrt{\nu_{kk}\nu_{k'k'}}}\boldsymbol{A}_k\left(\boldsymbol{\Sigma}_s + \boldsymbol{\Sigma}_0\right)\boldsymbol{A}_{k'}^\top$, and $\bar{\boldsymbol{\Sigma}}_{kk'} = \sqrt{\nu_{kk}\nu_{k'k'}}\begin{bmatrix}\tilde{\boldsymbol{\Sigma}}_{kk'} & \mathbf{0} \\ \mathbf{0} & \mathbf{0}\end{bmatrix} \in \mathbb{R}^{p\times p}$. Note that while in [31], all $\boldsymbol{\Omega}_{kk}$ must have strictly positive eigenvalues, their result can be easily extended to cover the case where some eigenvalues are zero by a continuity argument.

The feature-label covariance is given by:

$$\left[\hat{\boldsymbol{\Phi}}\right]_k^i = \mathbb{E}_{\boldsymbol{x}}\left[\boldsymbol{u}_k(\boldsymbol{x})\boldsymbol{x}^\top\boldsymbol{\theta}\right]_i = \mathbb{1}_{\{1\leq i\leq N_k\}}\left[\boldsymbol{A}_k\boldsymbol{\Sigma}_s\boldsymbol{\theta}\right]_i = \left[\bar{\boldsymbol{A}}_k\boldsymbol{\Sigma}_s\boldsymbol{\theta}\right]_i \tag{137}$$

$$[\boldsymbol{\Theta}]_{kk'}^{ij} = \left[\hat{\boldsymbol{\Phi}}\right]_k^i\left[\hat{\boldsymbol{\Phi}}\right]_{k'}^j = \left[\bar{\boldsymbol{A}}_k\boldsymbol{\Sigma}_s\boldsymbol{\theta}\right]_i\left[\bar{\boldsymbol{A}}_{k'}\boldsymbol{\Sigma}_s\boldsymbol{\theta}\right]_j \tag{138}$$

Recalling $\boldsymbol{\omega} := \boldsymbol{Q}^{1/2}\boldsymbol{\xi}$, we can now obtain an explicit form for the proximal $\boldsymbol{h}$:

$$\boldsymbol{h} := \underset{\boldsymbol{u}}{\operatorname{argmin}}\left[\frac{(\boldsymbol{u}-\boldsymbol{\omega})\boldsymbol{V}^{-1}(\boldsymbol{u}-\boldsymbol{\omega})}{2} + \hat{\ell}(y,\boldsymbol{u})\right] = \boldsymbol{V}(\boldsymbol{I}+\boldsymbol{V})^{-1}(\boldsymbol{V}^{-1}\boldsymbol{Q}^{1/2}\boldsymbol{\xi} + y\mathbf{1}) \tag{139}$$

The proximal $\boldsymbol{G}$ should not arise in this special case.

Next, we will simplify the saddle-point equations. Simplifying where possible, we may write:

$$\boldsymbol{f} = \boldsymbol{V}^{-1}(\boldsymbol{h}-\boldsymbol{w}) = (\boldsymbol{I}+\boldsymbol{V})^{-1}(y\mathbf{1}-\boldsymbol{\omega}) \tag{140}$$

$$\partial_{\boldsymbol{\omega}}\boldsymbol{f} = -(\boldsymbol{I}+\boldsymbol{V})^{-1} \tag{141}$$

$$\rho = \mathbb{E}_{\boldsymbol{x}}\left[\left(\frac{1}{\sqrt{d}}\boldsymbol{\theta}^\top\boldsymbol{x}\right)^2\right] = \frac{1}{d}\boldsymbol{\theta}^\top\boldsymbol{\Sigma}_s\boldsymbol{\theta} \tag{142}$$

$$\omega_0 \equiv \boldsymbol{m}^\top\boldsymbol{Q}^{-1/2}\boldsymbol{\xi} \tag{143}$$

$$\sigma_0 = \rho - \boldsymbol{m}^\top\boldsymbol{Q}^{-1}\boldsymbol{m} = \frac{1}{d}\boldsymbol{\theta}^\top\boldsymbol{\Sigma}_s\boldsymbol{\theta} - \boldsymbol{m}^\top\boldsymbol{Q}^{-1}\boldsymbol{m} \tag{144}$$

$$\hat{\boldsymbol{V}} = -\alpha\int dy\,\mathbb{E}_{\boldsymbol{\xi}}\left[Z^0(y,\omega_0,\sigma_0)(-(\boldsymbol{I}+\boldsymbol{V})^{-1})\right] = \alpha(\boldsymbol{I}+\boldsymbol{V})^{-1}\mathbb{E}_{\boldsymbol{\xi}}\left[\underbrace{\int dy\,Z^0(y,\omega_0,\sigma_0)}_{1}\right] \tag{145}$$

$$\Rightarrow \hat{\boldsymbol{V}} = \alpha(\boldsymbol{I}+\boldsymbol{V})^{-1} \tag{146}$$

We can simplify the prior equation for $\hat{\boldsymbol{V}}$ to the following set of equations:

$$\boldsymbol{V}_{kk} = \frac{1}{p}\operatorname{tr}\left[\bar{\boldsymbol{\Sigma}}_{kk}\left[\nu_{kk}\lambda_k\boldsymbol{I}_p + \hat{\boldsymbol{V}}_{kk}\bar{\boldsymbol{\Sigma}}_{kk}\right]^{-1}\right] \qquad k=1,\ldots,K \tag{147}$$

$$\boldsymbol{V}_{kk'} = \frac{1}{p}\operatorname{tr}\left[\bar{\boldsymbol{\Sigma}}_{kk'}\left[\hat{\boldsymbol{V}}_{k'k}\bar{\boldsymbol{\Sigma}}_{k'k}\right]^{-1}\right] \qquad k'\neq k \tag{148}$$

Equations 146 and 148 are solved by setting $\hat{\boldsymbol{V}}_{kk'} = \boldsymbol{V}_{kk'} = 0$ for all $k \neq k'$ so that $\hat{\boldsymbol{V}}_{kk'} = \hat{V}_k \delta_{kk'}$ and $\boldsymbol{V}_{kk'} = V_k \delta_{kk'}$. The diagonal components then satisfy

$$\hat{V}_k = \frac{\alpha}{1 + V_k} \tag{149}$$

$$V_k = \frac{1}{p} \text{tr} \left[ \bar{\boldsymbol{\Sigma}}_{kk} \left[ \nu_{kk} \lambda_k \boldsymbol{I}_p + \hat{V}_k \bar{\boldsymbol{\Sigma}}_{kk} \right]^{-1} \right] \tag{150}$$

Which may be solved separately for each $k = 1, \ldots, K$. Simplifying the remaining channel equations we have:

$$\hat{\boldsymbol{Q}} = \alpha(\boldsymbol{I} + \boldsymbol{V})^{-1} \left[ (\zeta^2 + \rho) \, \boldsymbol{1}\boldsymbol{1}^\top - \boldsymbol{1}\boldsymbol{m}^\top - \boldsymbol{m}\boldsymbol{1}^\top + \boldsymbol{Q} \right] (\boldsymbol{I} + \boldsymbol{V})^{-1} \tag{151}$$

$$\Rightarrow \hat{Q}_{kk'} = \frac{1}{\alpha} \hat{V}_k \hat{V}_{k'} \left[ \zeta^2 + \rho - m_k - m_{k'} + Q_{kk'} \right] \tag{152}$$

$$\hat{\boldsymbol{m}} = \alpha(\boldsymbol{I} + \boldsymbol{V})^{-1} \boldsymbol{1} \quad \Rightarrow \quad \hat{m}_k = \frac{\alpha}{(1 + V_k)} = \hat{V}_k \tag{153}$$

Simplifying the prior equations we obtain (through some tedious but straightforward algebra):

$$Q_{kk'} = \hat{Q}_{kk'} J_{kk'} + \hat{V}_k \hat{V}_{k'} \Lambda_{kk'} \tag{154}$$

$$m_k = \hat{V}_k R_k \tag{155}$$

Where we have defined:

$$\boldsymbol{G}_k \equiv \nu_{kk} \lambda_k \boldsymbol{I}_p + \hat{V}_k \bar{\boldsymbol{\Sigma}}_{kk} \tag{156}$$

$$J_{kk'} \equiv \frac{1}{p} \text{tr} \left[ \bar{\boldsymbol{\Sigma}}_{kk'} \boldsymbol{G}_{k'}^{-1} \bar{\boldsymbol{\Sigma}}_{k'k} \boldsymbol{G}_k^{-1} \right] \tag{157}$$

$$\Lambda_{kk'} \equiv \frac{1}{p} \boldsymbol{\theta}^\top \boldsymbol{\Sigma}_s \bar{\boldsymbol{A}}_k^\top \boldsymbol{G}_k^{-1} \bar{\boldsymbol{\Sigma}}_{kk'} \boldsymbol{G}_{k'}^{-1} \bar{\boldsymbol{A}}_{k'} \boldsymbol{\Sigma}_s \boldsymbol{\theta} \tag{158}$$

$$R_k \equiv \boldsymbol{\theta}^\top \boldsymbol{\Sigma}_s \bar{\boldsymbol{A}}_k^\top \boldsymbol{G}_k^{-1} \bar{\boldsymbol{A}}_k \boldsymbol{\Sigma}_s \boldsymbol{\theta} \tag{159}$$

Solving eq's 154,152 for $\boldsymbol{Q}$, we obtain:

$$Q_{kk'} = \frac{\gamma_{kk'}}{1 - \gamma_{kk'}} \left( \zeta^2 + \rho - m_k - m_{k'} \right) + \frac{\hat{V}_k \hat{V}_{k'}}{1 - \gamma_{kk'}} \Lambda_{kk'} \qquad (k \neq k') \tag{160}$$

Combining these results, we arrive at a formula for $E_{kk'}$:

$$E_{kk'} = \frac{1}{1 - \gamma_{kk'}} \left( \gamma_{kk'} \zeta^2 + \rho - \hat{V}_k R_k - \hat{V}_{k'} R_{k'} + \hat{V}_k \hat{V}_{k'} \Lambda_{kk'} \right) + \frac{1}{K^2} \sum_k \eta_k^2 \qquad (k \neq k') \tag{161}$$

where $\gamma_{kk'} \equiv \frac{1}{\alpha} \hat{V}_k \hat{V}_{k'} J_{kk'}$ \tag{162}

Where the order parameters $\hat{V}_k$, $k = 1, \ldots, K$ satisfy the fixed-point equations given by eq's 149, 150.

### G.2.2 Diagonal Terms

To calculate $E_{rr}$, we cannot ignore the presence of readout noise in the loss function. However, as the loss function is separable over the readouts $k$, we may calculate $E_{rr}$ using the results of [31] in

the special case where $K = 1$, incorporating the readout noise into the label noise. Concretely, we may calculate $E_{rr}$ as the error of a single linear predictor under the same setup as the off-diagonal terms except that in the training set $y^\mu \sim P_y^0(y|\frac{1}{\sqrt{p}}\boldsymbol{\theta}^\top \boldsymbol{x}^\mu)$ where $P_y^0(y|x) = \mathcal{N}(x, \zeta^2 + \eta_k^2)$. This may be recovered from the calculation for the off-diagonal terms by setting $k = k'$ and re-scaling $\zeta^2 \to \zeta^2 + \eta_k^2$, giving:

$$E_{kk} = \frac{1}{1 - \gamma_{kk}}\left(\gamma_{kk}(\zeta^2 + \eta_r^2) + \rho - 2\hat{V}_k R_k + \hat{V}_k^2 \Lambda_{kk}\right) + \frac{1}{K^2}\sum_k \eta_k^2 \qquad (k \neq k') \quad (163)$$

where the definitions of $\gamma_{kk}$, $\rho$, $R_k$, and $\Lambda_{kk}$ can be inherited from the off-diagonal case, as well as the saddle-point equations 149, 150.

### G.2.3 Full Error

The results obtained here are equivalent to the results of our main theorem, up to a reshuffling of additive constants $\eta_k^2$ among the error terms, and a trivial re-scaling of the order parameters as follows: $V_k \to \frac{1}{\lambda_k}V_k$, $\hat{V}_k \to \lambda_k \hat{V}_k$

# H   Equicorrelated Data Model

To gain an intuition for the joint effects of correlated data, subsampling, ensembling, feature noise, and readout noise, we simplify the formulas for the generalization error in the following special case:

$$\boldsymbol{\Sigma}_s = s\left[(1 - c)\boldsymbol{I}_M + c\boldsymbol{1}_M\boldsymbol{1}_M^\top\right] \tag{164}$$

$$\boldsymbol{\Sigma}_0 = \omega^2 \boldsymbol{I}_M \tag{165}$$

Here $s$ is a parameter which sets the overall scale of the data and $c \in [0, 1]$ tunes the correlation structure in the data and $\omega^2$ sets the scale of an isotropic feature noise. We consider an ensemble of $k$ readouts, each of which sees a subset of the features. Due to the isotropic nature of the equicorrelated data model and the pairwise decomposition of the generalization error, we expect that the generalization error will depend on the partition of features among the readout neurons through only:

- The number of features sampled by each readout: $N_r \equiv \nu_{rr}M$, for $r = 1, \ldots, k$
- The number of features jointly sampled by each pair of readouts $n_{rr'} \equiv \nu_{rr'}M$ for $r, r' \in \{1, \ldots, k\}$

Here, we have introduced the subsampling fractions $\nu_{rr} = \frac{N_r}{M}$ and the overlap fractions $\nu_{rr'} = \frac{n_{rr'}}{M}$

We will average the generalization error over readout weights drawn randomly from the space perpendicular to $\boldsymbol{1}_M$, with an added spike along the direction of $\boldsymbol{1}_M$:

$$\boldsymbol{w}^* = \sqrt{1 - \rho^2}\mathbb{P}_\perp \boldsymbol{w}_0^* + \rho\boldsymbol{1}_M \tag{166}$$

$$\boldsymbol{w}_0^* \sim \mathcal{N}(0, \boldsymbol{I}_M) \tag{167}$$

The projection matrix may be written $\mathbb{P}_\perp = \boldsymbol{I}_M - \frac{1}{N}\boldsymbol{1}_M\boldsymbol{1}_M^\top$. The two components of the ground truth weights will yield independent contributions to the generalization error in the sense that

$$\langle E_{rr'}\rangle = (1 - \rho^2)E_{rr'}(\rho = 0) + \rho^2 E_{rr'}(\rho = 1) \tag{168}$$

Calculating $E_{rr}$ and $E_{rr'}$ is an exercise in linear algebra which is straightforward but tedious. To assist with the tedious algebra, we wrote a Mathematica package which can handle multiplication, addition, and inversion of matrices of symbolic dimension of the specific form encountered in this problem. This form consists of block matrices, where the blocks may be written as $a\delta_{MN}\boldsymbol{I}_M + b\boldsymbol{1}_M\boldsymbol{1}_N^\top$, where $a, b$ are scalars and $\delta_{MN}$ ensures that there is only a diagonal component for square blocks (when $M = N$). This package is included as supplemental material to this publication.

## H.1 Diagonal Terms and Saddle-Point Equations

Here, we solve for the dominant values of $q_r$ and $\hat{q}_r$ and simplify the expressions for $E_{rr}$ in the case of equicorrelated features described above. In this isotropic setting, $E_{rr}, q_r, \hat{q}_r$ will depend on the subsampling only through $N_r = \nu_{rr}M$. We may then write, without loss of generality $\boldsymbol{A}_r = (\boldsymbol{I}_{N_r} \quad \boldsymbol{0}) \in \mathbb{R}^{N_r \times M}$ where $\boldsymbol{0}$ denotes a matrix of all zeros, of the appropriate dimensionality.

We start by simplifying the saddle-point equations 53,54. Expanding $\frac{1}{M} \operatorname{tr}\left(\boldsymbol{G}_r^{-1}\tilde{\boldsymbol{\Sigma}}_{rr}\right)$ and keeping only leading order terms, the saddle-point equations for $q_r$ and $\hat{q}_r$ reduce to:

$$q_r = \frac{\nu_{rr}\left(s(1-c) + \omega^2\right)}{\hat{q}_r(s(1-c) + \omega^2) + \nu_r} \tag{169}$$

$$\hat{q}_r = \frac{\alpha}{\lambda + q_r} \tag{170}$$

Defining $a \equiv s(1-c) + \omega^2$ and solving this system of equations, we find:

$$q_r = \frac{\sqrt{a^2\alpha^2 + 2a\alpha(\lambda - a)\nu_r + (a+\lambda)^2\nu_r^2} - a\alpha + (a-\lambda)\nu_r}{2\nu_r} \tag{171}$$

$$\hat{q}_r = \frac{\sqrt{a^2\alpha^2 + 2a\alpha(\lambda - a)\nu_r + (a+\lambda)^2\nu_r^2} + a\alpha - (a+\lambda)\nu_r}{2a\lambda} \tag{172}$$

We have selected the solution with $q_r > 0$ because self-overlaps must be at least as large as overlaps between different replicas. This solution to the saddle-point equatios can be applied to each of the $k$ readouts.

Next, we calculate $E_{rr}$. Expanding $\gamma_{rr} \equiv \frac{\alpha}{M\kappa^2} \operatorname{tr}\left[(\boldsymbol{G}^{-1}\tilde{\boldsymbol{\Sigma}})^2\right]$ to leading order in $M$, we find:

$$\gamma_{rr} = \frac{a^2\alpha\nu_r}{(\lambda + q_r)^2 (a\hat{q}_r + \nu_r)^2} \tag{173}$$

$$\langle E_{rr}\rangle_{\mathcal{D},\boldsymbol{w}^*}(\rho = 0) = \frac{1}{1-\gamma_{rr}}\frac{1}{M}\operatorname{tr}\left[\mathbb{P}_\perp\left(\boldsymbol{\Sigma}_s - \frac{2}{\nu_{rr}}\hat{q}_r\boldsymbol{\Sigma}_s\boldsymbol{A}_r^\top\boldsymbol{G}_r^{-1}\boldsymbol{A}_r\boldsymbol{\Sigma}_s + \frac{1}{\nu_{rr}}\hat{q}_r^2\boldsymbol{\Sigma}_s\boldsymbol{A}_r^\top\boldsymbol{G}_r^{-1}\tilde{\boldsymbol{\Sigma}}\boldsymbol{G}_r^{-1}\boldsymbol{A}_r\boldsymbol{\Sigma}_s\right)\mathbb{P}_\perp\right]$$
$$+ \frac{\gamma_{rr}}{1-\gamma_{rr}}\zeta^2 + \eta_r^2, \tag{174}$$

$$\langle E_{rr}\rangle_{\mathcal{D},\boldsymbol{w}^*}(\rho = 1) = \frac{1}{1-\gamma_{rr}}\frac{1}{M}\boldsymbol{1}_M^\top\left[\boldsymbol{\Sigma}_s - \frac{2}{\nu_{rr}}\hat{q}_r\boldsymbol{\Sigma}_s\boldsymbol{A}_r^\top\boldsymbol{G}_r^{-1}\boldsymbol{A}_r\boldsymbol{\Sigma}_s + \frac{1}{\nu_{rr}}\hat{q}_r^2\boldsymbol{\Sigma}_s\boldsymbol{A}_r^\top\boldsymbol{G}_r^{-1}\tilde{\boldsymbol{\Sigma}}\boldsymbol{G}_r^{-1}\boldsymbol{A}_r\boldsymbol{\Sigma}_s\right]\boldsymbol{1}_m$$
$$+ \frac{\gamma_{rr}}{1-\gamma_{rr}}\zeta^2 + \eta_r^2, \tag{175}$$

With the aid of our custom Mathematica package, we calculate the traces and contractions in these expressions and expand them to leading order in $M$, finding:

$$\langle E_{rr}\rangle_{\mathcal{D},\boldsymbol{w}^*}(\rho = 0) = \frac{1}{1-\gamma_{rr}}\left(s(1-c)\left(1 - \frac{(1-c)s\hat{q}_r\nu_r\left(\hat{q}_r(s(1-c) + \omega^2) + 2\nu_r\right)}{(\hat{q}_r(s(1-c) + \omega^2) + \nu_r)^2}\right)\right) + \frac{\gamma_{rr}\zeta^2 + \eta_r^2}{1-\gamma_{rr}} \tag{176}$$

$$\langle E_{rr}\rangle_{\mathcal{D},\boldsymbol{w}^*}(\rho = 1) = \frac{1}{1-\gamma_{rr}}\left(\frac{s(1-c)(1-\nu_{rr}) + \omega^2}{\nu_{rr}}\right) + \frac{\gamma_{rr}\zeta^2 + \eta_r^2}{1-\gamma_{rr}} \tag{177}$$

In the "ridgeless" limit where $\lambda \to 0$, we obtain:

$$\gamma_{rr} = \frac{4\alpha\nu_{rr}}{(\alpha + \nu_{rr} + |\alpha - \nu_{rr}|)^2} \tag{178}$$

$$\langle E_{rr}(\rho = 0)\rangle_{\mathcal{D},\boldsymbol{w}^*} = \left\{ \begin{array}{ll} \frac{s(1-c)\nu_{rr}}{\nu_{rr}-\alpha}\left(1 + \frac{s\alpha(1-c)(\alpha-2\nu_{rr})}{\nu_{rr}[s(1-c)+\omega^2]}\right) + \frac{\alpha\zeta^2+\nu_{rr}\eta_r^2}{\nu_{rr}-\alpha}, & \text{if } \alpha < \nu_{rr} \\ \frac{s(1-c)\alpha}{\alpha-\nu_{rr}}\left(1 - \frac{s(1-c)\nu_{rr}}{s(1-c)+\omega^2}\right) + \frac{\nu_{rr}\zeta^2+\alpha\eta_r^2}{\alpha-\nu_{rr}}, & \text{if } \alpha > \nu_{rr} \end{array} \right\} \quad (\lambda \to 0) \tag{179}$$

$$\langle E_{rr}(\rho = 1)\rangle_{\mathcal{D},\boldsymbol{w}^*} = \left\{ \begin{array}{ll} \frac{\nu_{rr}}{\nu_{rr}-\alpha}\left(\frac{s(1-c)(1-\nu_{rr})+\omega^2}{\nu_{rr}}\right) + \frac{\alpha\zeta^2+\nu_{rr}\eta_r^2}{\nu_{rr}-\alpha}, & \text{if } \alpha < \nu_{rr} \\ \frac{\alpha}{\alpha-\nu_{rr}}\left(\frac{s(1-c)(1-\nu_{rr})+\omega^2}{\nu_{rr}}\right) + \frac{\nu_{rr}\zeta^2+\alpha\eta_r^2}{\alpha-\nu_{rr}}, & \text{if } \alpha > \nu_{rr} \end{array} \right\} \quad (\lambda \to 0) \tag{180}$$

## H.2 Off-Diagonal Terms

In this section, we calculate the off-diagonal error terms $E_{rr'}$ for $r \neq r'$, again making use of our custom Mathematica package to simplify contractions of block matrices of the prescribed form. By the isotropic nature of the equicorrelated data model, $E_{rr'}$ can only depend on the subsampling scheme through $\nu_{rr}$, $\nu_{r'r'}$, and $\nu_{rr'}$. We can thus, without loss of generality, write:

$$\boldsymbol{A}_r = \begin{pmatrix} \boldsymbol{I}_{n_r \times n_r} & \boldsymbol{0}_{n_r \times n_{r'}} & \boldsymbol{0}_{n_r \times n_s} & \boldsymbol{0}_{n_r \times l} \\ \boldsymbol{0}_{n_s \times n_r} & \boldsymbol{0}_{n_s \times n_{r'}} & \boldsymbol{I}_{n_s \times n_s} & \boldsymbol{0}_{n_s \times l} \end{pmatrix} \in \mathbb{R}^{N_r \times M} \tag{181}$$

$$\boldsymbol{A}_{r'} = \begin{pmatrix} \boldsymbol{0}_{n_{r'} \times n_r} & \boldsymbol{I}_{n_{r'} \times n_{r'}} & \boldsymbol{0}_{n_{r'} \times n_s} & \boldsymbol{0}_{n_{r'} \times l} \\ \boldsymbol{0}_{n_s \times n_r} & \boldsymbol{0}_{n_s \times n_{r'}} & \boldsymbol{I}_{n_s \times n_s} & \boldsymbol{0}_{n_s \times l} \end{pmatrix} \in \mathbb{R}^{N_{r'} \times M} \tag{182}$$

where we have defined $n_s$ to be the number of features shared between the readouts, $n_r = N_r - n_s$ and $n_{r'} = N_{r'} - n_s$ and the count of remaining features $l = M - n_r - n_{r'} - n_s$.

Then, to leading order in $M$, we find:

$$\gamma_{rr'} = \frac{\alpha\nu_{rr'}(s(1-c)+\omega^2)^2}{(\lambda + q_r)(\lambda + q_{r'})(\nu_{rr} + (s(1-c)+\omega^2)\hat{q}_r)(\nu_{r'r'} + (s(1-c)+\omega^2)\hat{q}_{r'})} \tag{183}$$

Averaging $E_{rr'}$ over $\boldsymbol{w}_0^* \sim \mathcal{N}(0, \boldsymbol{I}_M)$, we get:

$$\begin{aligned} \langle E_{rr'}(\mathcal{D})\rangle_{\mathcal{D},\boldsymbol{w}^*}(\rho = 0) = &\frac{\gamma_{rr'}}{1-\gamma_{rr'}}\zeta^2 + \frac{1}{1-\gamma_{rr'}}\left(\frac{1}{M}\operatorname{tr}\left[\mathbb{P}_\perp \boldsymbol{\Sigma}_s \mathbb{P}_\perp\right]\right) \\ &- \frac{1}{M(1-\gamma_{rr'})}\operatorname{tr}\left[\mathbb{P}_\perp \boldsymbol{\Sigma}_s \left(\frac{1}{\nu_{rr}}\hat{q}_r \boldsymbol{A}_r^\top \boldsymbol{G}_r^{-1}\boldsymbol{A}_r + \frac{1}{\nu_{r'r'}}\hat{q}_{r'}\boldsymbol{A}_{r'}^\top \boldsymbol{G}_{r'}^{-1}\boldsymbol{A}_{r'}\right)\boldsymbol{\Sigma}_s \mathbb{P}_\perp\right] \\ &+ \frac{\hat{q}_r \hat{q}_{r'}}{M(1-\gamma_{rr'})}\frac{1}{\sqrt{\nu_{rr}\nu_{r'r'}}}\operatorname{tr}\left[\mathbb{P}_\perp \boldsymbol{\Sigma}_s \boldsymbol{A}_r^\top \boldsymbol{G}_r^{-1}\tilde{\boldsymbol{\Sigma}}_{rr'}\boldsymbol{G}_{r'}^{-1}\boldsymbol{A}_{r'}\boldsymbol{\Sigma}_s \mathbb{P}_\perp\right], \end{aligned} \tag{184}$$

$$\begin{aligned} \langle E_{rr'}(\mathcal{D})\rangle_{\mathcal{D},\boldsymbol{w}^*}(\rho = 1) = &\frac{\gamma_{rr'}}{1-\gamma_{rr'}}\zeta^2 + \frac{1}{M(1-\gamma_{rr'})}\left(\boldsymbol{1}_M^\top \boldsymbol{\Sigma}_s \boldsymbol{1}_M\right) \\ &- \frac{1}{M(1-\gamma_{rr'})}\boldsymbol{1}_M^\top \boldsymbol{\Sigma}_s \left(\frac{1}{\nu_{rr}}\hat{q}_r \boldsymbol{A}_r^\top \boldsymbol{G}_r^{-1}\boldsymbol{A}_r + \frac{1}{\nu_{r'r'}}\hat{q}_{r'}\boldsymbol{A}_{r'}^\top \boldsymbol{G}_{r'}^{-1}\boldsymbol{A}_{r'}\right)\boldsymbol{\Sigma}_s \boldsymbol{1}_M^\top \\ &+ \frac{\hat{q}_r \hat{q}_{r'}}{M(1-\gamma_{rr'})}\frac{1}{\sqrt{\nu_{rr}\nu_{r'r'}}}\boldsymbol{1}_M^\top \boldsymbol{\Sigma}_s \boldsymbol{A}_r^\top \boldsymbol{G}_r^{-1}\tilde{\boldsymbol{\Sigma}}_{rr'}\boldsymbol{G}_{r'}^{-1}\boldsymbol{A}_{r'}\boldsymbol{\Sigma}_s \boldsymbol{1}_M \end{aligned} \tag{185}$$

Calculating these contractions and traces and expanding to leading order in $M$, we get:

$$\langle E_{rr'}(\mathcal{D})\rangle_{\mathcal{D},\boldsymbol{w}^*}(\rho=0) = \frac{s(1-c)}{1-\gamma_{rr'}}\left(1 - \frac{s(1-c)\nu_{rr}\hat{q}_r}{\nu_{rr}+(s(1-c)+\omega^2)\hat{q}_r} - \frac{s(1-c)\nu_{r'r'}\hat{q}_{r'}}{\nu_{r'r'}+(s(1-c)+\omega^2)\hat{q}_{r'}}\right.$$
$$\left.+ \frac{s(1-c)(s(1-c)+\omega^2)\nu_{rr'}\hat{q}_r\hat{q}_{r'}}{(\nu_{rr}+(s(1-c)+\omega^2)\hat{q}_r)(\nu_{r'r'}+(s(1-c)+\omega^2)\hat{q}_{r'})}\right)$$
$$+ \frac{\gamma_{rr'}}{1-\gamma_{rr'}}\zeta^2 \tag{186}$$

$$\langle E_{rr'}(\mathcal{D})\rangle_{\mathcal{D},\boldsymbol{w}^*}(\rho=1) = \frac{1}{1-\gamma_{rr'}}\left(\frac{s(1-c)(\nu_{rr'}-\nu_{rr}\nu_{r'r'})+\omega^2\nu_{rr'}}{\nu_{rr}\nu_{r'r'}}\right) + \frac{\gamma_{rr'}}{1-\gamma_{rr'}}\zeta^2 \tag{187}$$

Taking $\lambda \to 0$ we get the ridgeless limit:

$$\gamma_{rr'} \to \frac{4\alpha\nu_{rr'}}{(\alpha+\nu_{rr}+|\alpha-\nu_{rr}|)(\alpha+\nu_{r'r'}+|\alpha-\nu_{r'r'}|)} \qquad (\lambda\to 0) \tag{188}$$

$$\langle E_{rr'}(\mathcal{D})\rangle_{\mathcal{D},\boldsymbol{w}^*}(\rho=0) = \frac{1}{1-\gamma_{rr'}}F_0(\alpha) + \frac{\gamma_{rr'}}{1-\gamma_{rr'}}\zeta^2 \quad (r\neq r') \tag{189}$$

where

$$F_0(\alpha) \equiv \begin{cases} \frac{(c-1)s\left(\nu_r\nu_{r'}((2\alpha-1)(c-1)s+\omega^2)-\alpha^2(c-1)s\nu_{rr'}\right)}{\nu_r((c-1)s-\omega^2)\nu_{r'}}, & \text{if } \alpha\leq\nu_{rr}\leq\nu_{r'r'} \\ \frac{(c-1)s\left(\nu_{r'}\left((c-1)s\nu_r+(\alpha-1)(c-1)s+\omega^2\right)-\alpha(c-1)s\nu_{rr'}\right)}{((c-1)s-\omega^2)\nu_{r'}}, & \text{if } \nu_{rr}\leq\alpha\leq\nu_{r'r'} \\ \frac{(c-1)s\left((c-1)s\nu_{r'}-cs\nu_{rr'}+(c-1)s\nu_r-cs+s\nu_{rr'}+s+\omega^2\right)}{(c-1)s-\omega^2}, & \text{if } \nu_{rr}\leq\nu_{r'r'}\leq\alpha \end{cases} \tag{190}$$

$$\langle E_{rr'}(\mathcal{D})\rangle_{\mathcal{D},\boldsymbol{w}^*}(\rho=1) = \frac{1}{1-\gamma_{rr'}}\left(\frac{s(1-c)(\nu_{rr'}-\nu_{rr}\nu_{r'r'})+\omega^2\nu_{rr'}}{\nu_{rr}\nu_{r'r'}}\right) + \frac{\gamma_{rr'}}{1-\gamma_{rr'}}\zeta^2 \qquad (\lambda\to 0) \tag{191}$$

## H.3    Optimal Regularization

Here, we derive the "locally" optimal regularization which minimizes the prediction error of the ensemble members independently. Equivalently we derive the optimal regularization for a single readout ($k=1$) or the regularization which minimizes the diagonal terms $E_{rr}$ of the generalization errror. By differentiating $E_{rr}$ with respect to the regularization $\lambda$, one can show that for the optimal regularization, we will have

$$\frac{1}{(1-\gamma_{rr})}\frac{dS_r}{d\lambda}\left[\frac{a^2\nu_{rr}}{\alpha(1-\gamma_{rr})}S_r\left((1-\rho^2)I_{rr}^0+\rho^2 I_{rr}^1+\zeta^2+\eta_r^2\right)+(1-\rho^2)s^2(1-c^2)\nu_{rr}(aS_r-1)\right] = 0 \tag{192}$$

It is easy to show that $\frac{dS_r}{d\lambda}$ cannot be equal to $0$. Setting the term in brackets to zero and solving for $\lambda$, we find with the aid of the computer algebra system Mathematica [54]:

$$\lambda^\star = a\left(\frac{a\left(\nu_{rr}(\zeta^2+\eta^2)+a\rho^2+s(1-c)\nu_{rr}(1-2\rho^2)\right)}{(1-c)^2\nu_{rr}^2(1-\rho^2)s^2}-1\right), \qquad (0<c\leq 1) \tag{193}$$

In the limiting case of isotropic data ($c=0$), the optimal regularization reduces to:

$$\lambda^\star = \frac{1}{\nu_{rr}}(\zeta^2 + \eta^2) + s\left(\frac{1}{\nu_{rr}} - 1\right), \qquad (c = 0) \tag{194}$$

We note that while generalization error is discontinuous at $c = 0$, this result can be quickly obtained from eq. 193 by noticing that the formula for the generalization error with $0 < c \leq 1$ reduces to the formula for generalization error with $c = 0$ when we set $\rho = c = 0$ (when $c = 0$, the generalization error does not depend on $\rho$ because there is no special direction in the data).

## H.4 Homogeneous Ensembling with Resource Constraints

We make further simplifications in the special case where $\lambda_r = \lambda$, $\eta_r = \eta$, $\nu_{rr} = \frac{1}{k}$ for all $r = 1, \ldots k$. For simplicity, we also set $\nu_{rr'} = 0$ for all $r \neq r'$ so that ensemble members sample mutually exclusive subsets of the features. We will consider both the ridgeless limit $\lambda \to 0$ and the case of "locally optimal" regularization $\lambda = \lambda^*$ (see eq. 193). Concretely, we show here that under these special cases, generalization error has the forms given in eqs. 25, 26.

We start by analyzing the saddle-point equations 111, 112. Note that in this special case the saddle point equations will be identical for all $r = 1, \ldots, k$. We will therefore suppress the $r$ index. Solving this quadratic system of equations explicitly, we encounter the following radical, which we assign variable name $x$:

$$x \equiv \sqrt{(a\alpha - a\nu + \lambda\nu)^2 + 4a\lambda\nu^2} \tag{195}$$

In order to simplify this radical to begin extracting the factor of $s(1 - c)$ that appears in eqs. 25, 26, we define a reduced regularization:

$$\Lambda \equiv \frac{\lambda}{s(1 - c)} \tag{196}$$

And substitute $\Lambda$, $H$, $W$, and $Z$ (recall eq. 24) for $\lambda$, $\eta$, $\omega$, and $\zeta$, giving

$$x = s(1 - c)\mathcal{X}(\alpha, \nu, W, \Lambda) \tag{197}$$

$$\text{where } \mathcal{X}(\alpha, \nu, W, \Lambda) \equiv \sqrt{(\alpha(W + 1) + \nu(\Lambda - W - 1))^2 + 4\Lambda\nu^2(W + 1)} \tag{198}$$

making the same substitutions, we can then write

$$\hat{q} = [s(1 - c)]^{-1}\mathcal{Q} \tag{199}$$

$$\text{where } \quad \mathcal{Q} \equiv \frac{2\alpha\nu}{(-\alpha(W + 1) + \nu(\Lambda + W + 1) + X)} \tag{200}$$

Continuing to make substitutions in this manner, we arrve at:

$$S = [s(1 - c)]^{-1}\mathcal{S} \tag{201}$$

$$\text{where } \quad \mathcal{S} \equiv \frac{\mathcal{Q}}{\nu + \mathcal{Q}(1 + W)} \tag{202}$$

Finally, we may express the generalization error in a reasonably compact form. In the special case at hand, the generalization error is written:

$$E_k = \frac{1}{k}E_{rr}(\nu_{rr} = \frac{1}{k}) + \frac{k - 1}{k}E_{rr'}(\nu_{rr'} = \frac{1}{k}\delta_{rr'}) \tag{203}$$

Substituting 201 for S in eq. 16, and making further substitutions as necessary, we arrive at:

$$E_{rr}(\nu_{rr} = \frac{1}{k}) = s(1 - c)\mathcal{E}_{rr}(k, \alpha, \rho, \Lambda, H, W, Z) \tag{204}$$

where

$$\mathcal{E}_{rr} = \frac{N_1 + N_2}{D} \tag{205}$$

$$N_1 = \alpha(H+1)k + \mathcal{S}(\mathcal{S}(W+1)(\alpha + WZ + Z) - 2\alpha) \tag{206}$$

$$N_2 = \alpha\rho^2(k - \mathcal{S})(W(k + \mathcal{S}) + k + \mathcal{S} - 2) \tag{207}$$

$$D = \alpha k - \mathcal{S}^2(W+1)^2 \tag{208}$$

We also obtain

$$E_{rr'}(\nu_{rr'} = \frac{1}{k}\delta_{rr'}) = s(1-c)\mathcal{E}_{rr'}(k, \alpha, \rho, \Lambda, H, W, Z) \tag{209}$$

where

$$\mathcal{E}_{rr'} = \frac{2(\rho^2 - 1)\mathcal{S}}{k} - 2\rho^2 + 1 \tag{210}$$

Combining these, we obtain the error of the ensemble as:

$$E_k = s(1-c)\mathcal{E}(k, \alpha, \rho, \Lambda, H, W, Z) \tag{211}$$

where:

$$\mathcal{E} = \frac{1}{k}\mathcal{E}_{rr} + \frac{k-1}{k}\mathcal{E}_{rr'} \tag{212}$$

These result are derived in the Mathematica notebook titled "EquiCorrParameterReduction.nb" included with the available code. It follows from eq. 211 that when $\lambda = 0$ error can be written in the form of eq. 22. It follows from equations 211, 209 that at "locally optimal" regularization $\lambda^*$, error can be written in the form of eq. 23. To see this, note that the reduced regularization $\Lambda^*$ which minimizes $E_{rr}$ will only depend on the other arguments of $\mathcal{E}_{rr}$. Full expresions for the generalization error in the case $\lambda \to 0$ and $\lambda = \lambda^*$ can be found in the mathematica notebooks "EquiCorrPhaseDiagram_ZeroReg.nb" and "EquiCorrPhaseDiagram_LocalReg.nb" included with the available code. These equations are long and difficult to interperet – nor are they directly used in our code – and so are omitted here.

### H.4.1 The Intermediate to Noise-Dominated Transition

The transition between the intermediate regime where $1 < k^* < \infty$ and the noise-dominated regime where $k^* = \infty$ can be studied analytically relatively painlessly in the ridgeless limit $\lambda \to 0$. The strategy we employ to determine this phase boundary is to examine the large-$k$ asymptotic expansion of $E_k$ to determine whether the error approaches its asymptotic value from below or above. If $E_k$ approaches $E_\infty$ from below, then $k^*$ must be finite. If $E_k$ approaches $E_\infty$ from above, then $k^*$ may be infinite – however, there is still the possibility of $k^* < \infty$ if $E_k$ is non-monotonic in $k$. In practice, we check the values of $E_k$ for $k = 1, 2, \dots, 100$ and $k = \infty$.

Setting $\Lambda = 0$ and expanding $E_k$ around $k = \infty$, we find:

$$\frac{E_k}{s(1-c)} = 1 - \rho^2 + \rho^2 W + \frac{((1+W)^2\rho^2 + \alpha(-2 + H + HW + 2\rho^2))}{(1+W)\alpha k} + \mathcal{O}\left(\frac{1}{k^2}\right) \tag{213}$$

Setting the coefficient of $k^{-1}$ to zero, we find the phase boundary as:

$$\alpha = \frac{(1+W)^2\rho^2}{2(1-\rho^2) - H(1+W)} \tag{214}$$

This equation explains the shapes of the boundaries between the intermediate and noise-dominated regions in the phase diagrams of fig. 4 with $\lambda = 0$ (see black dotted lines in panels b, c, d).

An analytical formula for the boundary between the intermediate and noise-dominated regimes at locally optimal regularization $\lambda = \lambda^*$ cannot be easily obtained. To understand why, we can asses the large-$k$ asymptotic expansion of the generalization error at locally optimal regularization:

$$E_k(\lambda = \lambda^*) = s(1 - c)\left[1 - \rho^2 + \rho^2 W + \frac{H}{k}\right] + \mathcal{O}\left(\frac{1}{k^2}\right) \tag{215}$$

This shows that at large $k$, when $H > 0$, error always approaches its asymptotic value from above (recall that when $H = 0$ we always have $k^* = 1$, so that there is no phase boundary unless $H > 0$). Thus, determining the phase boundary requires determining the value of $k$ which minimizes $E_k$, which is not analytically tractable.

## H.5  Infinite Data Limit

In this section we consider the behavior of generalization error in the equicorrelated data model as $\alpha \to \infty$ while keeping the $\lambda \sim \mathcal{O}(1)$. For simplicity, we assume $\nu_{rr'} = 0$ for $r \neq r'$, isotropic features ($c = 0$), no feature noise ($\omega = 0$) and uniform readout noise $\eta_r = \eta$ as in main text Fig. 3. This limit corresponds to data-rich learning, where the number of training examples is large relative to the number of model parameters. In this case, the saddle point equations reduce to:

$$\hat{q}_r \to \frac{\alpha}{\lambda} \tag{216}$$

$$q_r \to \frac{\nu_{rr}\lambda}{\alpha} \tag{217}$$

In this limit, we find that $\gamma_{rr'} \to 0$. Using this, we can simplify the generalization error as follows:

$$E_g = \frac{1}{k^2}\sum_{rr'=1}^{k} E_{rr'} = s\left[1 - \left(2 - \frac{1}{k}\right)\left(\frac{1}{k}\sum_{r=1}^{k}\nu_{rr}\right)\right] + \frac{\eta^2}{k} \tag{218}$$

Interestingly, we find that the readout error in this case depends on the subsampling fractions $\nu_{rr}$ only through their mean. Therefore, with infinite data, there will be no distinction between homogeneous and heterogeneous subsampling.

# I  Theoretical Learning Curves and Optimal Subsampling Phase Diagrams

Here, we provide additional learning curves and phase diagrams of $k^*$ such as those in Fig. 4a,b,c,d, exploring more parameter values for the task-model alignment $\rho$ and the Reduced noises $H$, $W$, and $Z$. Generalization errors are calculated for a homogeneous ensemble of $k$ linear regressors, as described in sections 5, H.4. We also show diagrams of generalization error $E_{k^*}$.

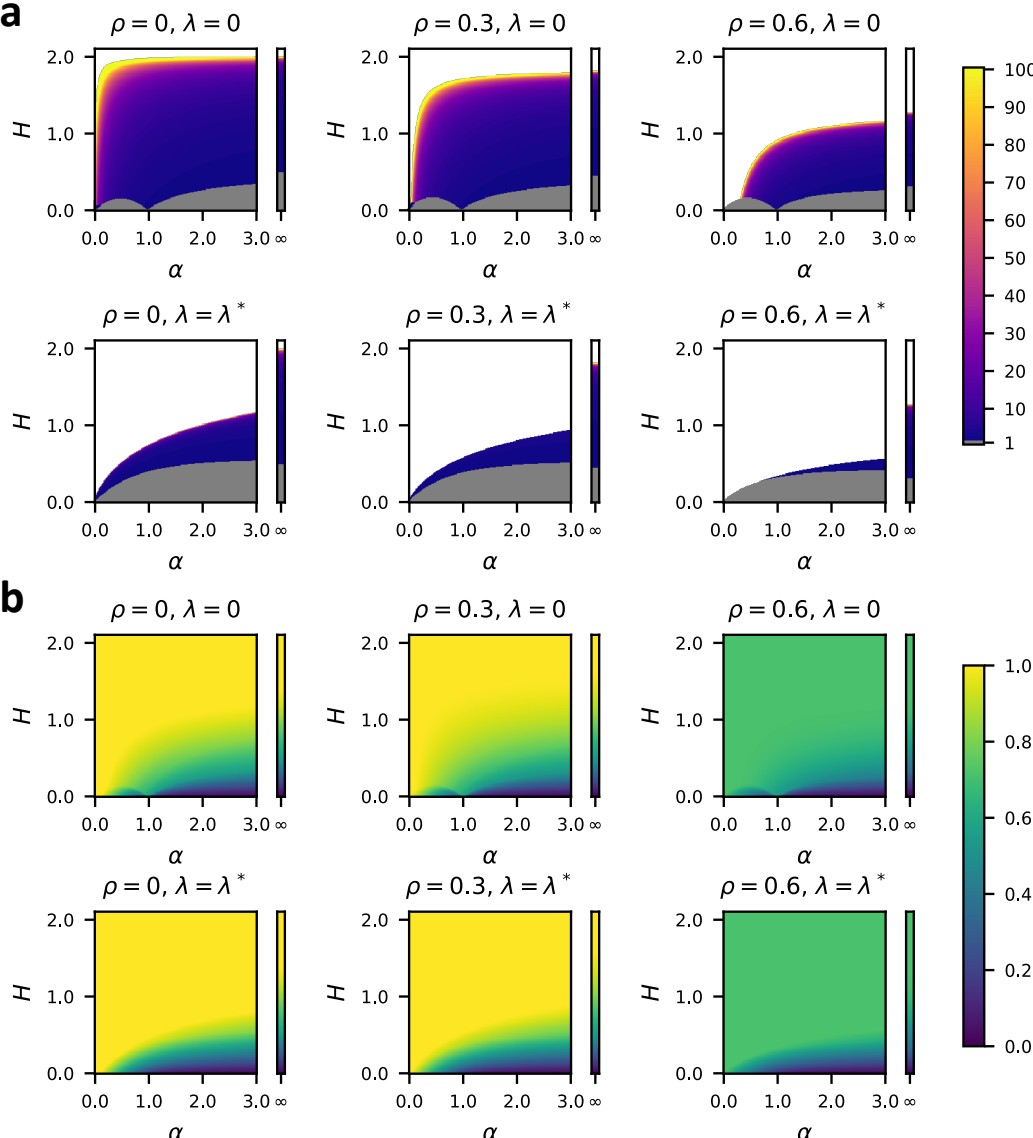

Figure S6: (a) Optimal ensemble size $k^*$ (eqs. 25, 26) in the parameter space of sample size $\alpha$ and reduced readout noise scale $H$ setting $W = Z = 0$. Grey indicates $k^* = 1$ and white indicates $k^* \to \infty$, with intermediate values given by the colorbar. Appended vertical bars show $\alpha \to \infty$. $\rho$ and $\lambda$ indicated in panel titles. $\lambda = \lambda^*$ denotes the 'locally optimal" regularization (see section H.3) (b) Optimal generalization error $E_{k^*}$ for the same parameter values in (a).

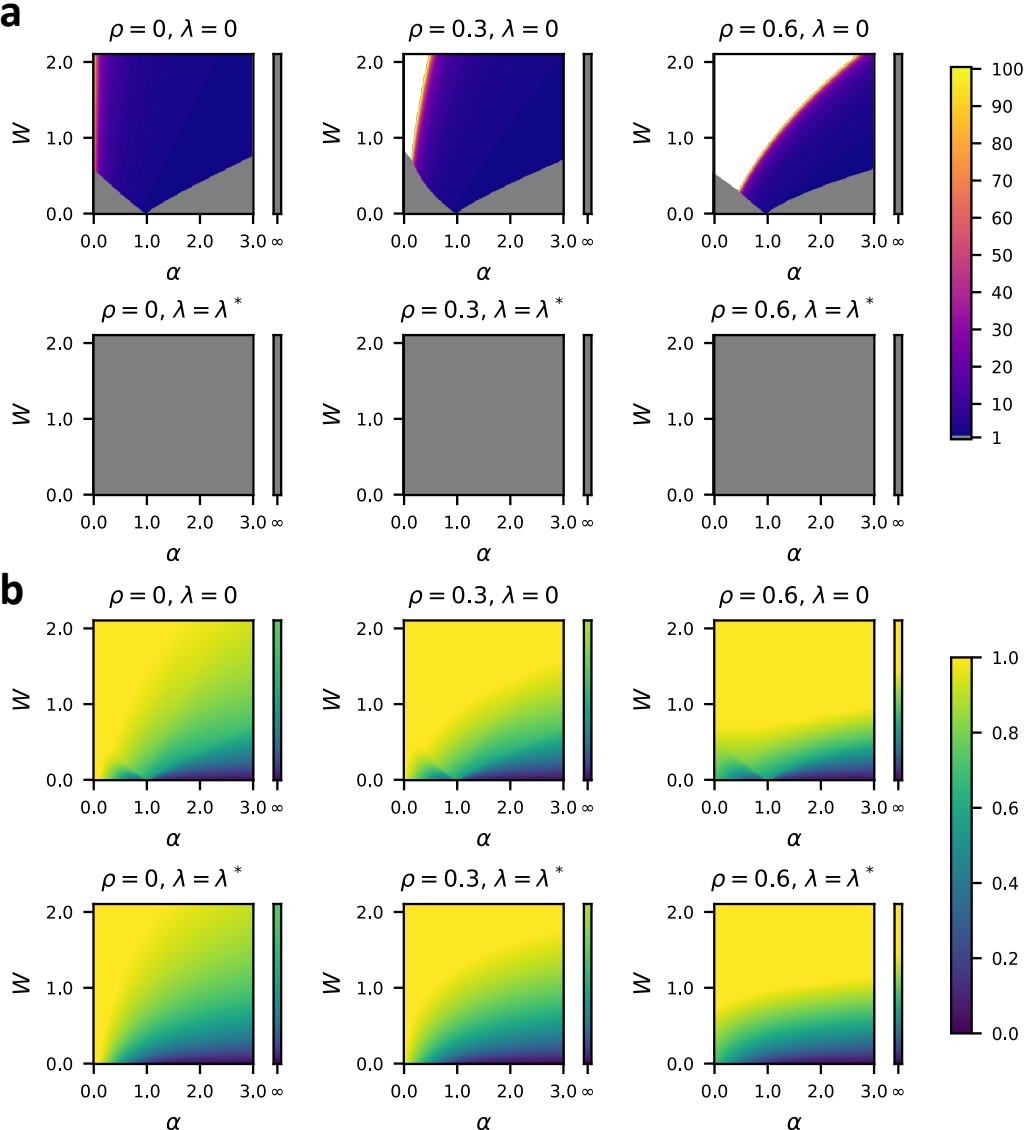

Figure S7: (a) Optimal ensemble size $k^*$ (eqs. 25, 26) in the parameter space of sample size $\alpha$ and reduced feature noise scale $W$ setting $H = Z = 0$. Grey indicates $k^* = 1$ and white indicates $k^* \to \infty$, with intermediate values given by the colorbar. Appended vertical bars show $\alpha \to \infty$. $\rho$ and $\lambda$ indicated in panel titles. $\lambda = \lambda^*$ denotes the 'locally optimal' regularization (see section H.3) (b) Optimal generalization error $E_{k^*}$ for the same parameter values in (a).

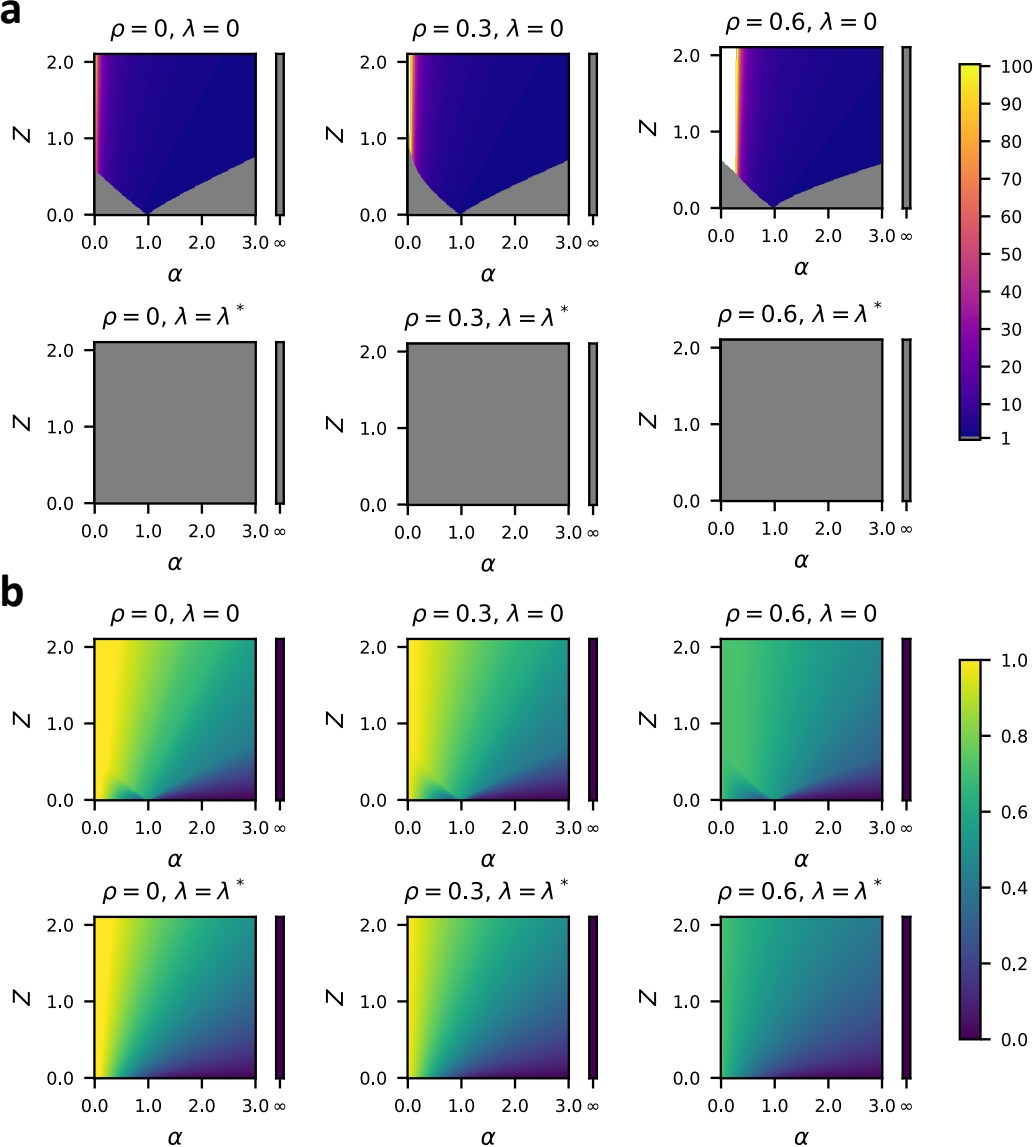

Figure S8: (a) Optimal ensemble size $k^*$ (eqs. 25, 26) in the parameter space of sample size $\alpha$ and reduced label noise scale $Z$ setting $H = W = 0$. Grey indicates $k^* = 1$ and white indicates $k^* \to \infty$, with intermediate values given by the colorbar. Appended vertical bars show $\alpha \to \infty$. $\rho$ and $\lambda$ indicated in panel titles. $\lambda = \lambda^*$ denotes the 'locally optimal" regularization (see section H.3) (b) Optimal generalization error $E_{k^*}$ for the same parameter values in (a).

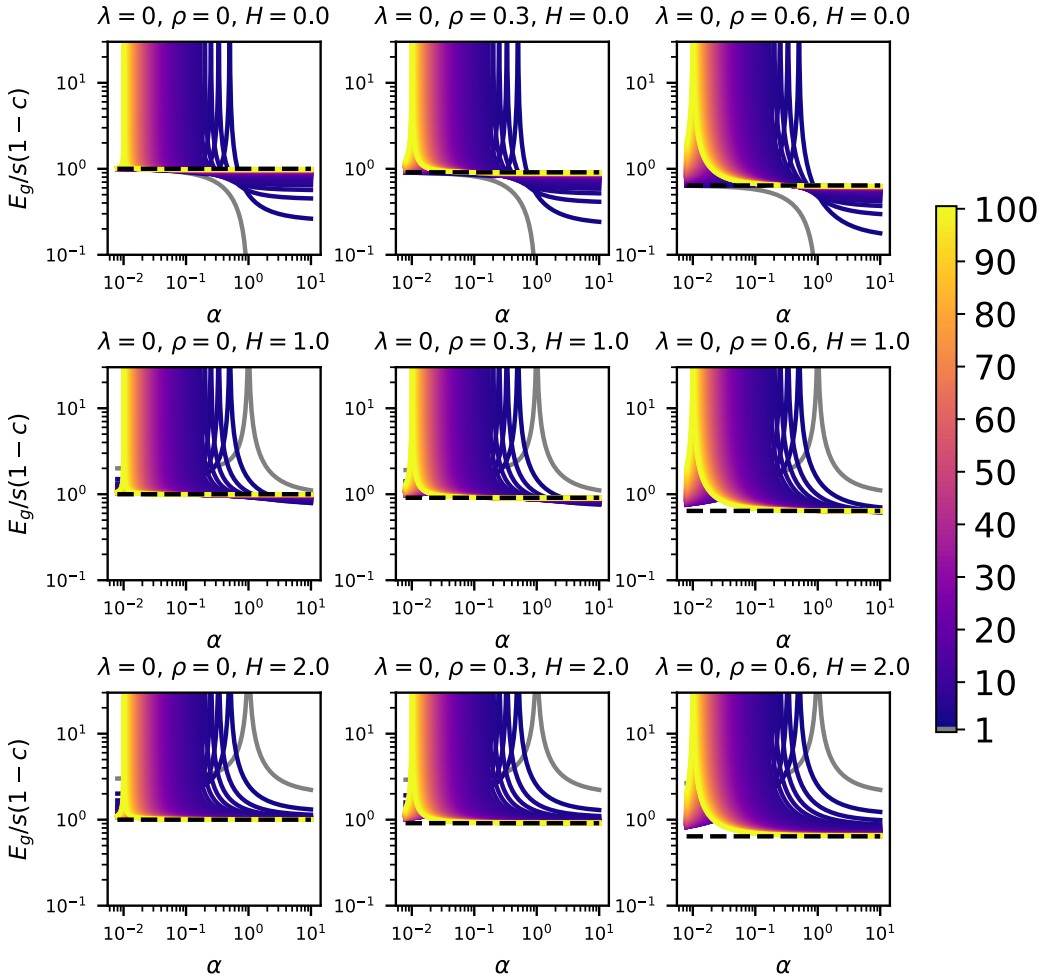

Figure S9: Reduced generalization errors $E_g/s(1-c)$ with $\lambda = 0$ and $W = Z = 0$ (given by eq. 22) for linear ridge ensembles of varying size $k$. $\rho$ and $H$ values indicated above plots. Grey lines indicate $k = 1$, dashed black lines $k \to \infty$, and intermediate $k$ values by the colorbar.

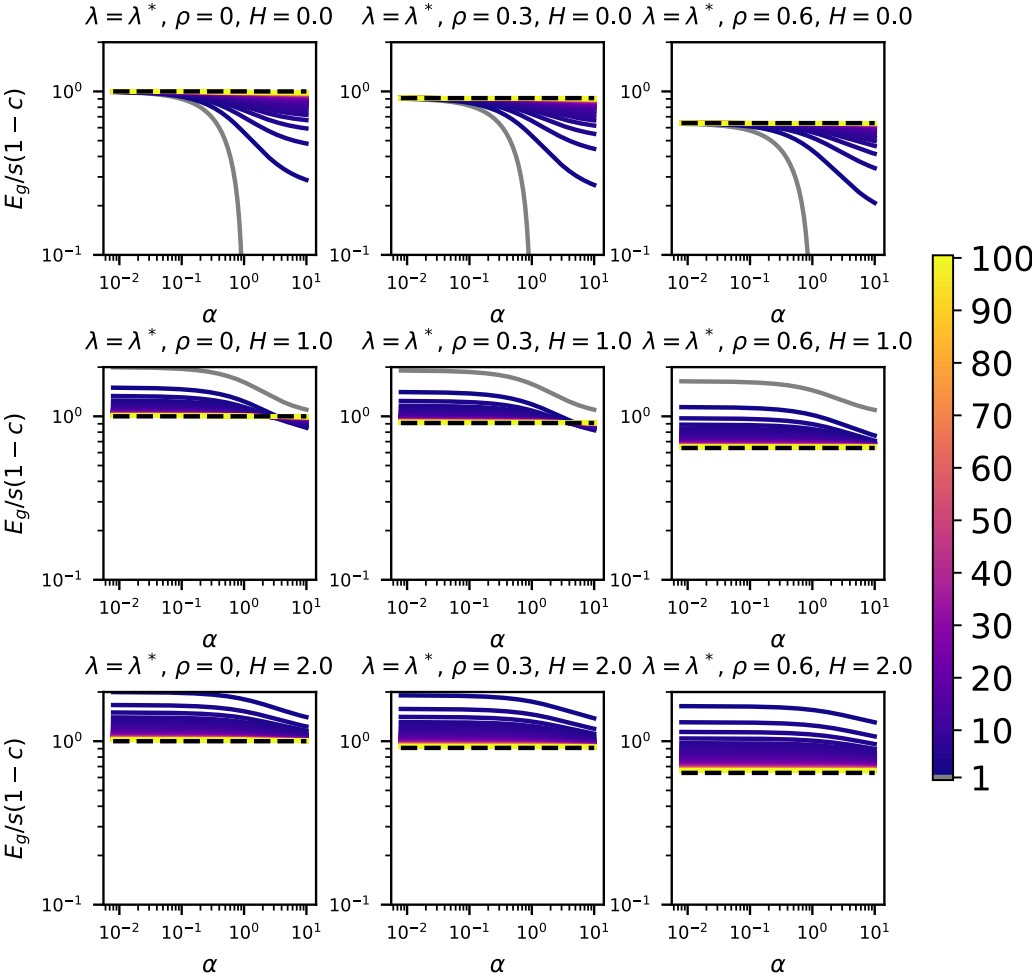

Figure S10: Reduced generalization errors $E_g/s(1-c)$ with $\lambda = \lambda^*$ and $W = Z = 0$ (given by eq. 23) for linear ridge ensembles of varying size $k$. $\rho$ and $H$ values indicated above plots. Grey lines indicate $k = 1$, dashed black lines $k \to \infty$, and intermediate $k$ values by the colorbar.

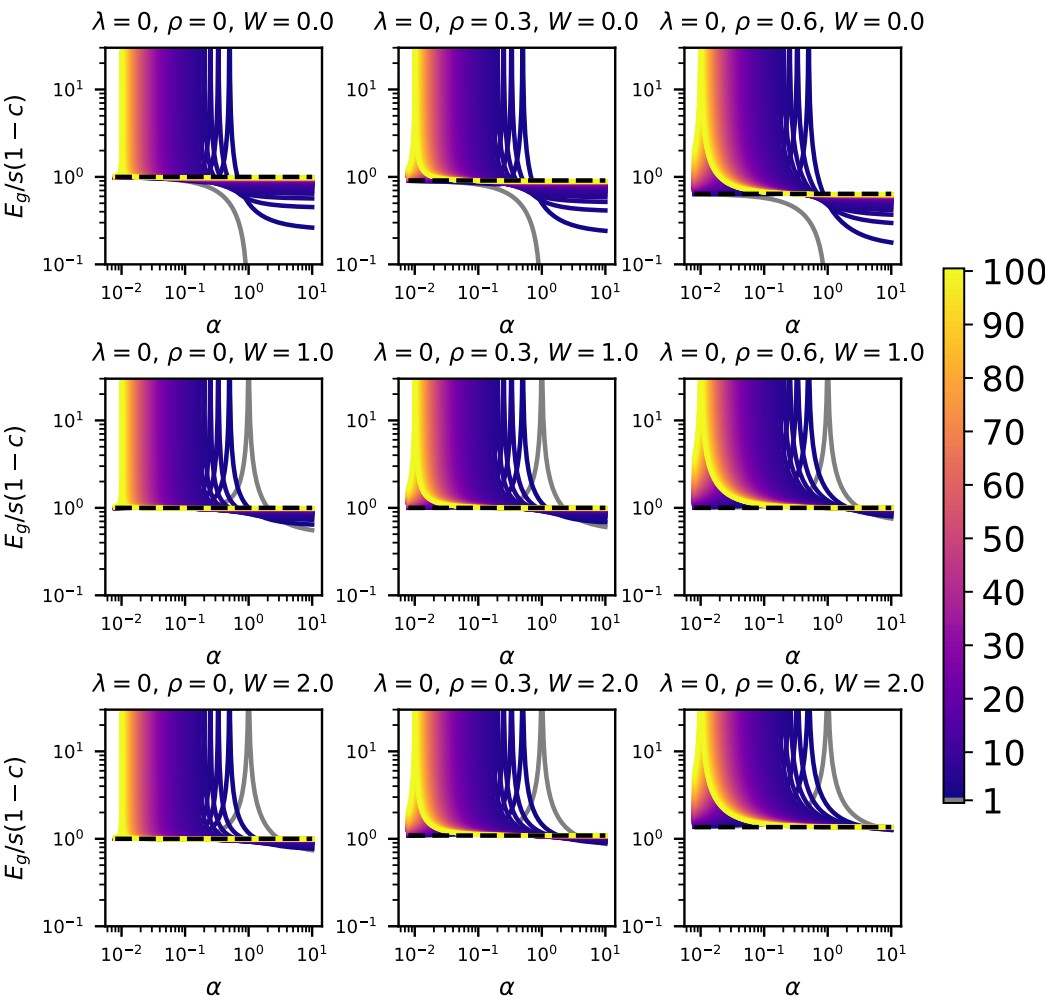

Figure S11: Reduced generalization errors $E_g/s(1-c)$ with $\lambda = 0$ and $H = Z = 0$ (given by eq. 22) for linear ridge ensembles of varying size $k$. $\rho$ and $W$ values indicated above plots. Grey lines indicate $k = 1$, dashed black lines $k \to \infty$, and intermediate $k$ values by the colorbar.

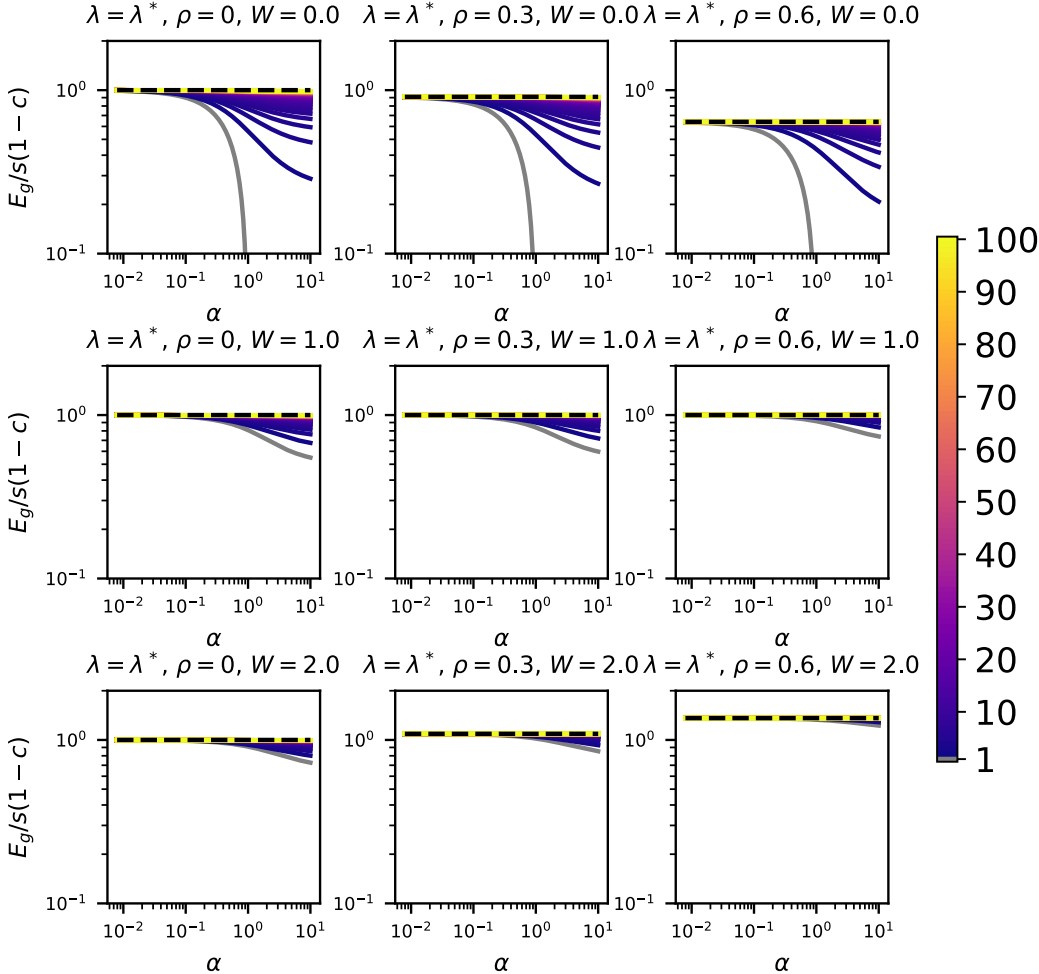

Figure S12: Reduced generalization errors $E_g/s(1-c)$ with $\lambda = \lambda^*$ and $H = Z = 0$ (given by eq. 23) for linear ridge ensembles of varying size $k$. $\rho$ and $W$ values indicated above plots. Grey lines indicate $k = 1$, dashed black lines $k \to \infty$, and intermediate $k$ values by the colorbar.

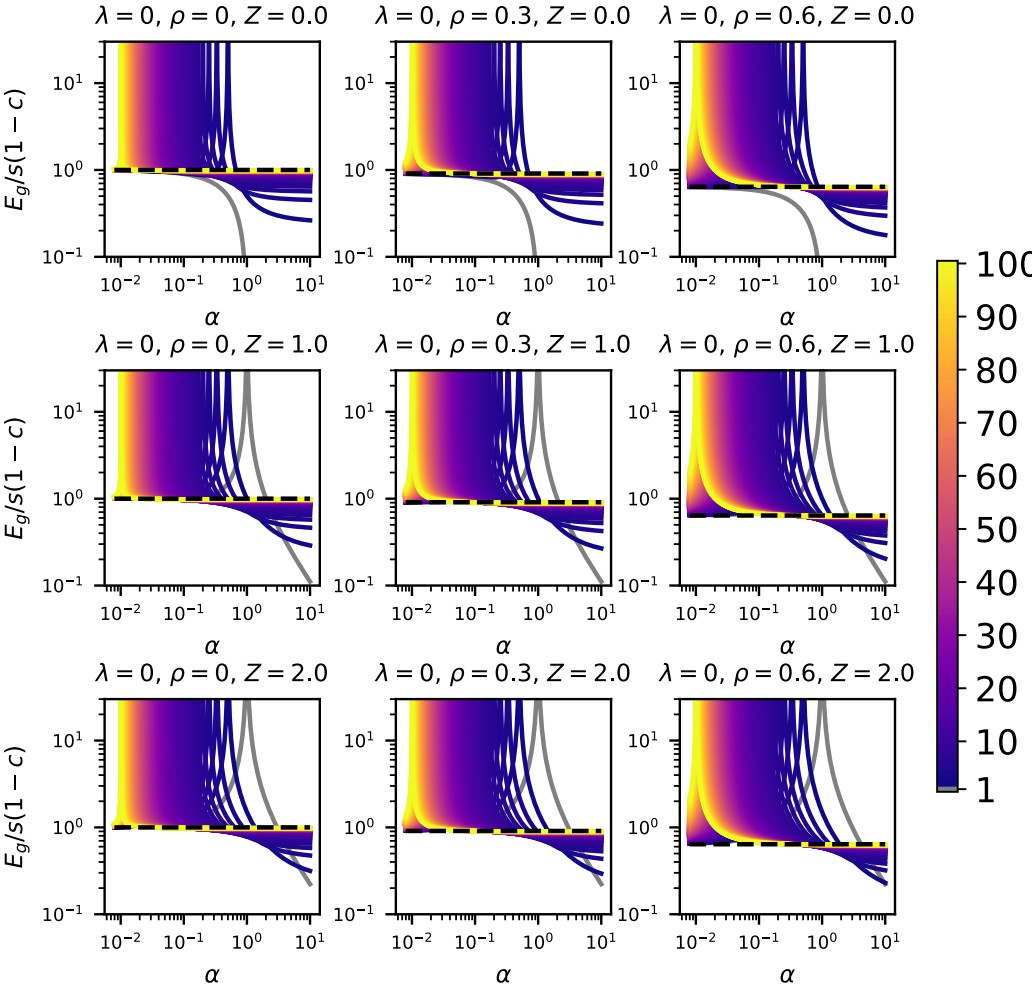

Figure S13: Reduced generalization errors $E_g/s(1-c)$ with $\lambda = 0$ and $H = W = 0$ (given by eq. 22) for linear ridge ensembles of varying size $k$. $\rho$ and $Z$ values indicated above plots. Grey lines indicate $k = 1$, dashed black lines $k \to \infty$, and intermediate $k$ values by the colorbar

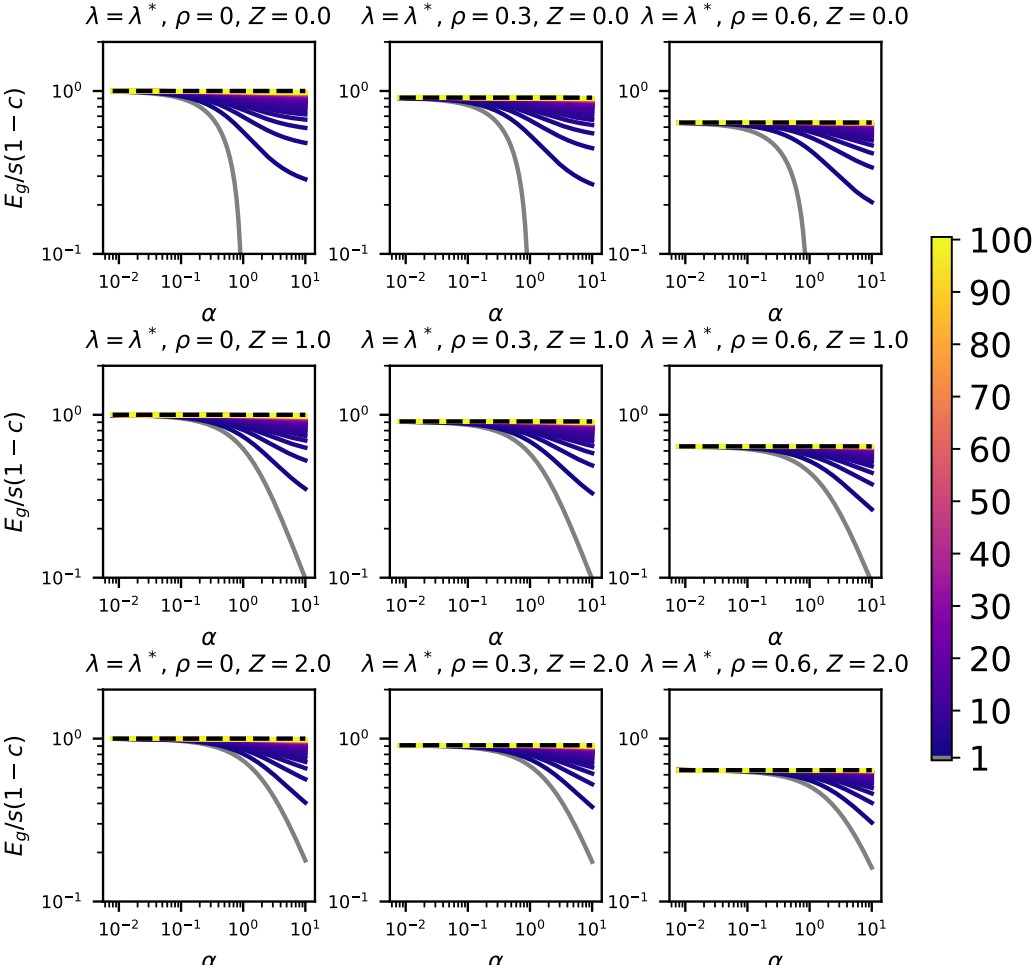

Figure S14: Reduced generalization errors $E_g/s(1-c)$ with $\lambda = \lambda^*$ and $H = W = 0$ (given by eq. 23) for linear ridge ensembles of varying size $k$. $\rho$ and $Z$ values indicated above plots. Grey lines indicate $k = 1$, dashed black lines $k \to \infty$, and intermediate $k$ values by the colorbar.

