# OpenReview forum: "Learning Curves for Noisy Heterogeneous Feature-Subsampled Ridge Ensembles"
_NeurIPS.cc/2023/Conference — NeurIPS 2023 poster_

### Official Review · Reviewer_uQUq · 2023-06-29

**Soundness:** 3 good
**Presentation:** 3 good
**Contribution:** 3 good
**Rating:** 7
**Confidence:** 2

**Summary:**

The authors provide a theoretical analysis of the case of ensemble learning with linear ridge regression for the case where heterogeneous feature subsampling is used. The authors make a number simplifying assumptions about the distribution of the data (Gaussian distribution and noise, linear function), and using the replica trick from statistical physics derive generalization error for the L2-regularized least-squares solution. In simulations, the authors demonstrate that the derived solution coincides with numerically calculated generalization error, and based on analyzing learning curves provide a number of insights about the behavior of ensembled ridge regression, suggesting a novel way for mitigating the double descent phenomenon through the use of heterogenous feature subsampling in ensembling.

**Strengths:**

The work provides novel insights about how double-descent behaves based on noise, regularization, and subsampling and provides alternative strategies to regularization for avoiding the peak. These are quite central questions in modern machine learning theory and new ideas here can prove to be significant. The mathematical analysis seems rigorous and the replica trick used for the analysis appears to be a new type of tool for studying this phenomenon, though I do not have suitable mathematical background to follow all the proofs in detail. The basic setting and assumptions are clearly defined, related work seems to be covered well, and novelty of the contribution compared to related similar works is clearly established.

**Weaknesses:**

The work makes many simplifying assumptions that are unlikely to match most real-world learning problems (e.g. Gaussian data, linear teacher function), and analyzes may not be in practice tractable beyond the simpler cases such as analyzed in Section 2.3. with globally correlated data. It is not so clear whether the results would offer practical tools for an analyst wishing to apply heterogeneous feature-subsampled ridge ensembles in an optimal way, though results in the supplementary materials about CIFAR10 Classification Task suggest the results can have also practical relevance.


**Questions:**

How computationally demanding is the computation of the generalization error curves?

---

> ### Author Rebuttal · Authors · 2023-08-08
>
> *Weaknesses:
> The work makes many simplifying assumptions that are unlikely to match most real-world learning problems (e.g. Gaussian data, linear teacher function), and analyzes may not be in practice tractable beyond the simpler cases such as analyzed in Section 2.3. with globally correlated data.
> It is not so clear whether the results would offer practical tools for an analyst wishing to apply heterogeneous feature-subsampled ridge ensembles in an optimal way, though results in the supplementary materials about CIFAR10 Classification Task suggest the results can have also practical relevance.*
>
> We thank the reader for raising this point.  To address this weakness, we will add the following sentences to the discussion section:
> “A large  line of work has shown that Gaussian data approximations accurately predict the learning curves in many real world problems, particularly when using a least-squares loss function (see, for example, Pesce [2023], Canatar [2021], Gerace [2022], Hu [2022]).”
> Referring to the following papers:
> Pesce [2023]: https://doi.org/10.48550/arXiv.2302.08923
> Canatar [2021]: https://www.nature.com/articles/s41467-021-23103-1
> Gerace [2022]: https://arxiv.org/abs/2205.13303
>  Hu [2022]: https://arxiv.org/abs/2009.07669
>
> We have also performed our own realistic simulations which are presented in the supplemental material, demonstrating that our main insights carry over  to the CIFAR10 classification task.  We will add the following sentences  to section 2.5 of the main text to highlight these findings:
> “In figure S1, we train linear models to predict the labels of images from the CIFAR10 dataset from subsamples of the features in the top hidden layer of a pre-trained deep neural network.   We find that heterogeneous ensembling (over linear models with varying input dimensionality) prevents catastrophic over-fitting, yielding monotonic learning curves without the need for regularization.  (see SI for details)”
>
> *Questions:
> How computationally demanding is the computation of the generalization error curves?*
>
> All of the computations necessary for this paper can be performed in a matter of hours on a single GPU.  Computing the theoretical generalization error curves can be expensive (taking many minutes) in the general case where the result is given as a contraction of matrices, and where solving the fixed point equations for the order parameters requires inverting a large matrix at each step.  However, in the special case of globally correlated (equicorrelated) features, we obtain accurate, fully analytical expressions which can be plotted instantly. If accepted, we will add a note about the computational complexity of the theoretical and empirical calculations of the error curves to the paper.

---

> > ### Comment · Reviewer_uQUq · 2023-08-14
> >
> > Thank you for the clarifications provided.

---

### Official Review · Reviewer_QQ7H · 2023-07-02

**Soundness:** 4 excellent
**Presentation:** 2 fair
**Contribution:** 2 fair
**Rating:** 5
**Confidence:** 4

**Summary:**

The authors provide theoretical results on generalization for ridge regression for the case of an ensemble of regression models trained on feature subsets, with noise, and correlation.

They relate this to the previously observed double-descent phenomena and characterize how the fraction of subsampled features, the regularization strength, and other properties affect the double descent phenomenon and where it occurs (i.e., how these impact generalization error).

Finding that double descent occurs at different places for different feature subset amounts, they also propose and analyze using a heterogenous ensemble of different feature subsampling sizes as a way to mitigate double descent without having to tune regularization or model architectures.


****Update****
Revised my score after the responses - see my response comment for rationale.


**Strengths:**

The generalization bounds for the particular setting seem novel.

The problem is a highly relevant one and the idea of using feature subset ensembles is interesting and potentially useful- as alluded to in the conclusion - as searching over many architecture and regularization settings for large neural nets may be prohibitive, so this kind of approach could provide an alternative for such cases with a fixed architecture, by creating a feature subsample ensemble instead - however this possibility is just alluded to and not tested, as only theoretical results for simple ridge regression problems are analyzed.

The authors provide thorough analyses of the results of the theorem for generalization error, to try to elucidate what different aspects of the ridge regression problem and choices for the modeling mean in terms of generalization error.

**Weaknesses:**

1) The paper is hard to follow and extract any useful meaning or conclusions from.

a) Some terms are not defined or given adequate explanation.  As particular example - it would be helpful to explain what "readout noise" and readout dimensionality mean in this paper when first introduced - this was not clear to me and I found its use to be confusing in the theoretical setup section and even the related work.  It's just mentioned without defining what it is or how it related to the model, and it would help to give an example of what it would correspond to.

b) A lot of different parameters are introduced and hard to remember when trying to connect to the various results figures and analyses -  it may help to remind the reader and also include simple examples of models describing what the different parameters and values would correspond to.

c) I feel the main results and conclusion need a simpler explanation and deciphering for the reader - as the various plots and analyses are still based on a large number of symbols / parameters that it's hard to remember, and the reader would have to dig through the text each time to recall what they are.  Some simplified example and conclusion, as well as a table may help.  It's hard to come away with any kind of recommendation or firm conclusion of what we could expect to work better in practice as well.

d)

2) It's not clear how useful and impactful the results from the paper and the derived theorems are.
It seems the complicated expression derived is too complex to derive any useful properties, and requires knowing ground truth properties - so the benefit is not clear.

Even in the simplified tractable case - it's not clear what interpretation comes out of it.

In generally it's not clear if any novel and useful findings have come out of this.

I think a key part of the issue might be - after presenting each set of results / figures, some interpretations and conclusions and take-ways for the reader are needed and currently lacking - results are just presented with little explanation or analysis.


3) Related to the previous points, I feel having real data and model experiments and results could really strengthen this work and also more fully explore the idea and benefit proposed of using heterogenous feature sub-sampling ensembles.


4) Novelty is also somewhat unclear.
Obviously, heterogenous feature subsampling ensembles have been studied and applied in the past, and for some specific models there are theoretical results on generalization error as well, such as tree ensembles or even linear model ensembles with different kinds of approaches such as random subspace models.
Additionally, the authors point to some specific related work as well on theoretical study of generalization error as well.  However, clearly spelling out the differences in this work seems lacking.  For example, in related work - they mention other work also studying feature subsampling for ensembles and finding it provides an implicit regularization approach, in particular [20].  However, it's not stated what differentiates this work from this prior work - what novel aspect does this provide?  It's necessary to spell out what limitations of previous work this addresses and overcomes.

I.e., overall it would help to clearly spell out how their work is different and novel with respect to prior work on feature-subsampling ensembles and what new conclusions and approaches result from it.


**Questions:**


"...correlated data by ensembles with heterogeneous readout dimensionality."
This is not very clear - what is meant by "readout dimensionality"?  I'm assuming
it means heterogeneous input data dimensionality - so why not just say that?  "Readout dimensionality" is a term I've almost never heard and sounds like it mean the dimension of the model output, which doesn't make sense in this case.


It would be helpful to explain what "readout noise" means in this paper when first introduced - this was not clear to me and I found its use to be confusing in the theoretical setup section.  It's just mentioned without defining what it is or how it related to the model, and it would help to give an example of what it would correspond to.  After several re-reads I concluded it probably corresponds to noisy model outputs during training when using something like dropout.  However, for the most cases in practice for regression, even if noise is used in training, when applying the models in practice deterministic outputs are used (no longer any "readout noise").


2.6 is hard to follow - again what is mean by each type of "readout"?  What is a "single fully-connected readout" and what does it mean by "multiple sparsely connected readouts"?  I'm assuming the latter is the feature subsampling ensemble, and the former a single model with no feature-subsampling, but this is described in a confusing way so it's not clear and took some though to try to decipher this.
What is mean by correlation code and correlation strength?  How are these defined, and how are they used to generate the models / how are they used in these analyses and results?

Also how do these conclusions differ from past studies of ensembling approaches?




Why in all the examples is P <= M?  It would be nice to explain this as well, and also what happens when P > M.  I.e., after some thought I assume this is to focus on the over-parameterized regime, but this is never really stated and explained, and there are still many practical cases where P >> M.

What impact does k have on generalization error?

Also wondering how the results compare to the non-ensemble case?  I.e., a single model with no feature subset sampling.
I.e., this shifts the problem to tuning the subset sample fraction sizes


Also it's not clear to me from the results and figures if tuning lambda (regularization) can always result in better performance then using the heterogenous ensemble?





**Limitations:**

Yes - doesn't seem needed in this case.

---

> ### Author Rebuttal · Authors · 2023-08-09
>
> Thank you for writing a thorough review of our submission.  Please see the response to all reviewers for a discussion of the meaning of “readout noise”, a proposed table clarifying the meaning of the parameters referenced in proposition 2 and figures 2, 3, and 4, discussion of a comparison between ensembling and L2 regularization, and a discussion of applications to real data.
>
> Thank you also for pointing out that “readout dimensionality” is confusing terminology.  This confusion may be due to the fact that in linear regression the number of input features is equal to the number of model parameters, which we referred to as the “readout dimensionality.”  If accepted, we will remove this term from the paper, and instead refer to the “input data dimensionality” as suggested.  We clarify that the model output is one-dimensional for the setup described in propositions 1 and 2 and all figures of the main text.  Our qualitative finding that heterogeneous input data dimensionality can mitigate double-descent carries over to simulations of the CIFAR10 classification task where the output dimensionality is 10, as shown in the SI.
>
> On terminology – You have assumed correctly that a “single fully-connected readout”  is a single model with no feature-subsampling, and “multiple sparsely connected readouts” is the feature subsampling ensemble.  We will update this sentence to the following: “We now ask whether ensembling is a fruitful strategy – i.e. whether a feature subsampling ensemble outperforms a single model without feature subsampling.”
>
> Thank you for prompting us to clarify the novelty of our work. We introduce and study a new form of heterogeneous subsampling, and demonstrate its benefits in an analytically tractable setting.  Specifically, we consider ensembling over models with heterogeneity in the number of features connected by each ensemble member.  All prior research that we are aware of, including [20], considers only the setting in which all ensemble members have the same size.  We emphasize this distinction in the “Related works” section, writing: “However, [18] and [19] focus their analysis on the case of isotropic data and Gaussian random masks of homogeneous dimensionality.  In contrast, we explicitly consider learning from correlated data by ensembles with heterogeneous [input data dimensionality].” More work is required to establish and study the practical benefits of this form of heterogeneity.  If accepted, we will acknowledge this more clearly in the final version of the paper.  This is also the first work we are aware of which presents a detailed study of the interplay between ensembling and readout noise, or to produce phase diagrams of the type shown in figure 4.
>
> Thank you for pointing out that Section 2.6 is hard to follow.  This will be re-written to reflect an updated version of the special case in proposition 2.  The parameters of the phase diagrams in figure 4 will be re-interpreted in terms of an effective signal-to-noise ratio and data-task alignment.  Please see the response to all reviewers for an overview of these developments and the attached updated version of figure 4.
>
> Also how do these conclusions differ from past studies of ensembling approaches?
> Past studies of ensembling approaches have typically aimed to establish formal equivalences between optimal ensembling and optimal regularization.  The optimal subsampling fraction, though, depends on the size (number of samples) of the training set.  In this work, we present heterogeneous ensembling as a strategy to mitigate double-descent in settings where the size of the training set is unknown.  We are the first study to explicitly analyze the behavior of ensembles with heterogeneous input data dimensionality and to recognize the benefits of this type of heterogeneity to mitigate double-descent.
>
> To clarify why P<=M in all the figures, we will add the following sentence to the discussion:
> “As ensemble members are trained independently, the learning curve cannot be divided into an under-parameterized and over-parameterized regime.  Rather, at any sample size $P$, each ensemble member with $N_r<P$ is in its over-parameterized regime and each ensemble member with $N_r>P$ is in its under-parameterized regime.”
> Because of this, the plots shown cover both the under-parameterized and over-parameterized regimes of each member of the ensemble (since $N_r \leq M$ for each ensemble member)
>
> To clarify the impact of k on generalization error, we will add the following sentences to the discussion of figure 4 at the end of section 2.6:
> “These plots show the effect of the ensemble size $k$ on the generalization error curves for ‘homogeneous ensembles’, where each ensemble member sees the same number of features, and in the absence of regularization.  We find that increasing $k$ may either help or hurt generalization performance, depending on the parameters of the task, and the size of the training set.  The shape of the learning curve changes with $k$ to shift the double-descent peak to the value $\alpha = \frac{1}{k}$.”
> However, readers may still wonder how the value of $k$ affects generalization at finite ridge, and in heterogeneous ensembles.  If accepted, we will clarify this by adding panels to figure 3 which plot learning curves for ensembled linear regression at finite ridge $\lambda$ for ensembles of a handful of $k$ values, as we do for the CIFAR10 classification problem in figure S1.
> To the question:
> *Also wondering how the results compare to the non-ensemble case? I.e., a single model with no feature subset sampling. I.e., this shifts the problem to tuning the subset sample fraction sizes*
> Calculating the optimal subset sample fraction sizes is precisely what we do in the phase diagrams in figure 4.  Please see the response to all reviewers for an updated version of figure 4 and subsequent analysis.

---

> > ### Comment · Reviewer_QQ7H · 2023-08-18
> >
> > Thank you for the responses - I think the proposed changes are very helpful.  The table of key parameters / terms is very useful to quickly look things up.
> >
> > On further thought - I feel one other possible thing that also may help a bit is using the commonly used symbols in ML that people are familiar with - as they may more easily have an idea of what is meant.  For instance - sample size is most commonly represented with "N" or "n" (and more rarely "m"), but here "P" is used.  Similarly feature dimension is most commonly "p" (though  sometimes "m" as well as is used here).  But all these symbols are used to mean different things here.
> >
> > Also it would be best to mention the additional results in the supplementary materials in the main paper, even if they can't be fit in the main paper - in particular mentioning experiments were also performed on Cifar 10 and briefly summarizing the results.  I think these set of experiments are very helpful in illustrating the point and validating the idea on real data.  It still would be nice to have results on additional datasets - to show how this behavior holds more generally.
> >
> > Overall I am more inclined to lean on the accept side after the responses and other discussions.  I think it is an interesting work that is useful for researchers to see.  I still feel the presentation could be improved and hopefully the promised changes will address this.  I.e., it just reads very busy and hard to follow and parse in its current form - even some of the experiment results which are nice I expect could be quite difficult to parse for readers.  Part of this is due to the large amount of parameters / variables included, but I can understand its difficult to get around these in this kind of theoretical work.

---

### Official Review · Reviewer_gGcm · 2023-07-05

**Soundness:** 3 good
**Presentation:** 3 good
**Contribution:** 2 fair
**Rating:** 5
**Confidence:** 4

**Summary:**

This paper introduces a theoretical investigation into ensembling methods applied to linear ridge regression with feature subsampling. It builds upon previous research in this area by extending the analysis to include scenarios with varying readout dimensionality.

By employing the replica method, the authors derive expressions for the average-case generalization error. These expressions capture the influence of both the data structure and the hyperparameters of the ensembling method employed.

Furthermore, the paper focuses on a simplified scenario to examine how the degree of subsampling and heterogeneity affect the occurrence of the double descent phenomena in learning curves for generalization error. This analysis provides potential insights into the understanding of this well-known and recurring phenomenon in machine learning.

**Strengths:**

- The paper is clearly written, and the experiments presented are of high quality.

**Weaknesses:**

- The authors use as proof technique for the main theorem the replica method, a heuristic method coming from statistical physics. While this method has been proven to be rigorous in certain contexts, its general correctness has not yet been formally established.  Unfortunately, the paper fails to acknowledge this limitation, which undermines the overall credibility of the research findings.
- In terms of technical results, the computation leading to preposition 1 appears to be a minor modification of the calculation presented in a previous work by Loureiro [2022]. However, unlike the present study, Loureiro's work accompanies the calculation with a rigorous proof. This discrepancy raises concerns about the rigour and robustness of the current paper's technical contributions.

(Minor) Typos:
- Line 4: the the
- Line 53: studied studied
- Line 212: eta

**Questions:**

- Would the authors consider revising the presentation of their results from Proposition 1 to reflect the heuristic nature of the findings? This would involve explicitly stating that the results are derived using a heuristic method, thereby providing clarity to readers about the level of mathematical rigour employed in the study. Such a revision would enhance transparency and enable readers to appropriately interpret the results.

**Limitations:**

While the authors do acknowledge certain limitations in their work and propose potential new directions for applying the techniques employed, there are a couple of crucial aspects that remain unaddressed. Firstly, the paper fails to explicitly mention that the replica method utilized lacks mathematical rigour, which is an important consideration for interpreting the validity and reliability of the results. Additionally, the authors do not provide any indications or suggestions on how to potentially establish a more rigorous foundation for their findings.

---

> ### Author Rebuttal · Authors · 2023-08-09
>
> Thank you for pointing out typos, we will correct them.  We thank the reader for their comments and suggestions.  The weaknesses identified in this review revolve mainly around the fact that we have derived our results using the replica method from statistical physics. We do in the remarks made directly after the statement of our main result in the main text plainly state that the replica trick is a “non-rigorous but standard” approach.  We make no claim that our results are fully rigorous, but because the results obtained from the replica method have been demonstrated to coincide with rigorous results for many similar problems and our results show excellent agreement with numerical experiments,, we have good reason to expect that our result is correct. However, in light of the concerns raised here we see it fit to change the statement of our main result to further emphasize that the replica trick is not rigorous, and to establish a rigorous basis for our main result.
>
> In the case where the data covariance matrix has a bounded spectrum, we believe  that our results may be obtained through a clever special case of the rigorous result of Loureiro [2022].  While their derivation does not explicitly consider ensembles with variation in the number of features viewed by each ensemble member, this type of heterogeneity may be added in post-hoc by choosing data covariance matrices which “zero out” a number of neurons which varies over the ensemble.  If accepted, we will add a supplemental section discussing this correspondence and including a detailed derivation of our result from the general result of Loureiro [2022].  We will also update the “proof” of the main theorem to the following:
>
> “We calculate the terms in the generalization error using the replica method,  a standard but non-rigorous method from the statistical physics of disordered systems.  The full derivation may be found in the SI.  In the special case where the covariance matrices $\Sigma_s, \Sigma_0$ have bounded spectrum, this result may be obtained as a clever special case of the results of Loureiro [2022] (see SI for derivation).”
>
> We will also add the following remark after the proof of proposition 2:
> “Note that, as in this case $\Sigma_s$ does not have a bounded spectrum, this result does not follow from the rigorous results of Loureiro [2022].  However, we find excellent agreement between theory and experiment when data dimension is sufficiently large.”
>
> We emphasize that, even as our general result may be recovered as a special case of the results of Loureiro [2022], simplifying this general result in the special case of subsampling from globally correlated features is a very tedious calculation which requires significant work.  Our investigation of this general result also differs entirely from Loureiro [2022], which considered only ensembles with the same number of readout weights across ensemble members, and included no study of feature noise or readout noise.

---

> > ### Comment · Reviewer_gGcm · 2023-08-11
> >
> > Thank you for taking the time to address my criticisms.
> >
> > I have raised my grade to "5: Borderline accept"

---

> > > ### Author Response · Authors · 2023-08-14
> > >
> > > Thank you for taking the time to read our rebuttal and modify your review.

---

### Official Review · Reviewer_JR3L · 2023-07-10

**Soundness:** 3 good
**Presentation:** 3 good
**Contribution:** 2 fair
**Rating:** 5
**Confidence:** 4

**Summary:**

This article characterizes the asymptotic performance curve of a heterogeneous feature ensembling framework for ridge linear regression, in the limit of comparably large numbers of data samples and variables. For having different error peaks in a double-decent performance curve as a function of the data sample size, predictors built of heterogeneous feature sets give rise to an ensemble that is robust to the number of data samples, without a carefully tuned regularization parameter.

**Strengths:**

* Clarity of the mathematical setup.
* Numerical validation of asymptotic results on finite data sets.
* Discussion of the theoretical implications of the analysis.

**Weaknesses:**

* The analysis applies only to the square loss while recent related work addressed already a large family of convex losses.
* Even though the interest of heterogeneous feature ensembling is explained as mitigating the double-descent phenomenon, this problem can be in fact settled by optimizing the regularization parameter on a cross-validation set.
* There is no experiment on real data sets to see how well the observed consequences such as the mitigated double-descent curve apply to real data.

**Questions:**

I wonder if the optimal learning performance is achieved at a uniform subsampling rate (i.e., homogeneous ensembles); and whether the feature ensembling can indeed improve the performance without readout noise, as is the case with readout noise (and under certain conditions on the data model parameters) according to the discussion of Section 2.6.

**Limitations:**

The limitations are discussed. There does not seem to be any negative societal impact.

---

> ### Author Rebuttal · Authors · 2023-08-09
>
> Thank you for writing a helpful review.  Your questions about cross-validated regularization and real-world experiments are addressed in the global rebuttal and attached PDF.

---

### Official Review · Reviewer_tCNd · 2023-07-13

**Soundness:** 3 good
**Presentation:** 2 fair
**Contribution:** 3 good
**Rating:** 6
**Confidence:** 4

**Summary:**

This paper provides an asymptotic analysis of ensembles of ridge regressors using varying numbers of subsampled features. The authors consider a Gaussian data model with feature noise and readout noise in addition to label noise. Using the replica method from statistical physics, they obtain precise asymptotics of the limiting ensemble risk for any collection of subsampling operators. They specialize this result in the case of globally correlated isotropic data and numerically plot double descent curves for various non-overlapping ensemble strategies. They find that in poorly regularized regimes, increased heterogeneity of feature subset sizes leads to a significant decrease of worst-case generalization risk, in other words dampening the double descent peak, albeit typically at the expense of increased risk in other data aspect ratios. Departing from heterogeneity to consider instead homogeneous non-overlapping ensembles, the authors further demonstrate that in the presence of readout noise, there are regimes in the phase space of readout noise level and global feature correlation in which it is optimal to use an ensemble rather than a single learner, although without readout noise it is never optimal to ensemble.

**Strengths:**

The biggest strength of this paper in comparison to existing work on linear ensembles of random projections of features is that this analysis applies to arbitrary projection operators that need not be independent. In fact, for the specific cases the authors consider of non-overlapping feature subsets, which would correspond closely with distributed optimization practice (I recommend the authors emphasize this in their introduction), the projection operators are clearly very much not independent. Furthermore, the most general form of their results in Proposition 1 also makes no assumption on the relationship between the projection operators and the covariances, in contrast with most existing work which assumes them to also be independent.

I am also not aware of any other work which explicitly considers heterogeneous linear ensembles, which is a novel contribution.

**Weaknesses:**

I think that the biggest weakness of this paper is the justification of the data model and of the advantage of using a (heterogenous) ensemble. I think that these could be improved by adding justifying and motivating comments throughout the paper, and if the authors would indicate how they would make such changes for the camera ready version, I would increase my score of the paper.

Data model:
- The features are corrupted with additional noise that is independent of the label, which is an interesting but highly non-standard assumption. Even more non-standard is the assumption that this noise is drawn independently in every forward pass. This assumption really should be motivated---should the reader be imagining this as something like noise due to dropout in a neural network?
- The model outputs are corrupted with readout noise independent across each member of the ensemble, again drawn independently each forward pass. This seems like an even stranger assumption to me; I could imagine some negligible computational noise, but I struggle to come up with a setting in which we would see the noise levels that are modeled in this paper. It is very important to motivate this, as the experiment in Figure 4 only justifies ensembles in settings where readout noise is high. Without good motivation, it seems like a strawman.

Ensemble advantages:
- This is a complaint I have for many other papers that deal with double descent, but applies equally to this paper. The "correct" solution to double descent is to properly regularize, and other strategies such as the ensembling proposed here can be seen as "hacks" that can mitigate some of the worst double descent effects but are typically suboptimal. We see this in this case as well. E.g., consider Figure 3, where the heterogenous ensembles only improve risk in the under-regularized setting. In the setting with more appropriate regularization (c.iv-vi), where the risk is lower than in (c.i-iii) for most values of $\alpha$, the heterogeneous ensemble is strictly worse than the homogeneous ensemble, which I am sure itself is strictly worse than the optimally tuned full ridge regressor.
- No comparisons are made to the full ridge regressor. Obviously, the ensemble would have to be worse, as ridge regression is typically the minimum mean squared error estimator in additive noise settings. However, this is a lost opportunity to demonstrate perhaps that bad double descent effects could be mitigated at a fraction of the computational cost (since the ensembles consider small subsets).
- Essentially, the risks are a linear combination of the individual risks of each ensemble member, plus some cross-terms. Thus in the limit of zero regularization, these ensembles would still have infinte double descent peaks, one for each different subsampling ratio. So to an extent, the authors already rely on regularization to help make the ensemble robust to double descent, although the benefits are realized for fairly small regularization already. Still, the authors should justify the need to regularize even with ensembles and discuss tuning.
- Related to the previous points, it's not clear why worst case risk is an important metric, since tuning of regularization in some way or another seems necessary.
- Part of the issues of statistical performance may be due to the non-overlapping partition of features. In the related works, ensembles with optimality guarantees typically require many members with overlapping subsets of features, which the authors do not consider here.

Others:
Overall, I found the plots to be very busy and difficult to decipher. I give one suggestion on how to improve Figure 4 in Questions.

**Questions:**

Instead of the replica method, do the authors think that the same result could be obtained by the more rigorous "deterministic equivalences" from random matrix theory [DS21]? Related work has recently successfully applied this technique to random projections, which should be directly applicable to ensembles [LPJBT22], which seems closely related to the future direction raised by the authors at the end of the paper.

In Figure 4.a.i, I cannot understand the meaning of the white dashed line. I thought it should indicate the boundary between when $k^* = 1$ and $k^* = 2$, but I cannot figure out why there is a purple region above the line, and why the caption says "there is no region of the phase
space where $k^* = 2$." Another suggestion is to add markers in the (a) plots to indicate the points in phase space that correspond to the (b) plots.

[DS21] E. Dobriban and Y. Sheng, Distributed linear regression by averaging, The Annals of Statistics, 49
(2021), pp. 918 – 943, https://doi.org/10.1214/20-AOS1984.

[LPJBT22] D. LeJeune, P. Patil, H. Javadi, R. G. Baraniuk, R. J. Tibshirani, “Asymptotics of the sketched pseudoinverse,” 2022, https://arxiv.org/abs/2211.03751

Other remarks:
- 1: "Bagging" is conventionally short for "bootstrap aggregating", but you only subsample, which is often called "subbagging" in the literature.
- 2: Ensembles do reduce variance, but subsampling feature adds bias.
- 4: "the the"
- 212: "eta" should be $\eta$

**Limitations:**

While I understand the usefulness of the replica method for obtaining quick and often accurate results, it is unfortunate that it is a non-rigorous approach. The authors have mentioned this in the supplemental material, but I think it needs mention somewhere in the main paper. This is important, as it is not clear whether some of the things done here are "legal"---for example, many random matrix theory results which could be applied here to perform similar analysis require covariance matrices with uniformly bounded operator norm, yet the globally correlated model violates this, which would typically require additional care to ensure that the result should hold, rather than simply plugging in the covariance. The reader deserves to know that while the general intuitions are likely true to hold, they should be careful in blindly applying the results.

---

> ### Author Rebuttal · Authors · 2023-08-09
>
> Thank you for writing a thorough review of our submission. Please see the global response as well as the below.
>
> We had two interpretations of the feature noise in mind when creating the problem setup.  The first is as an inherent noise due to stochasticity in a physical neural network, such as an analog neural network or a biological neural circuit.  The second is as a corruption of the input data. In both cases, the noise is present in both training and test data. Whether the noise is introduced through noise in the data itself or introduced through noise in the neural network computation is a matter of interpretation—but leads to an identical mathematical model.  If accepted, we will clarify this in the statement of and discussion following proposition
>
> Thank you for your comment that regularization is the “‘correct’ solution to double descent,” which we agree with. We do not argue that heterogeneous ensembling is the best regularizer, but to introduce heterogeneous ensembling as a regularization method and study its properties. We will add to figure 3c comparisons with the learning curves at optimal L2 regularization.  We will also add the following sentences: “in the feature subsampling ensemble, we find that heterogeneity of this type smooths out the double-descent peak of the learning curve. However, when computational resources permit it, a single fully-connected readout layer with optimal l2 regularization outperforms a feature subsampling ensemble, unless additional sources of variance over the ensemble are introduced (see section 2.6).
> Thank you for raising the interesting point that error diverges at zero regularization.  Note that the  genuine divergence of the generalization error is an artifact of the infinite-dimensional limit.  We also show in the supplemental material that heterogeneous subsampling can mitigate double-descent without any regularization in a classification setting.  The linear regression model with small positive regularization serves as a fruitful demonstration of the benefits of heterogeneous ensembling.  We will add that “in the infinite-dimensional regression setting, a small regularization  is always necessary to prevent divergence of the generalization error at the interpolation threshold of each ensemble member.”  We will also add to figure 3c comparisons with the learning curves at optimal regularization..
>
> In a regression setting, where regularization is necessary to prevent divergence of the generalization error, we propose the following procedure to quickly obtain a robust predictor:
> Pick a reasonable but small regularization parameter.
> Train an ensemble of linear predictors on varying numbers of data features.
> Through this procedure, one can avoid the sharp edges of double-descent without a computationally expensive parameter sweep.  We will update our discussion to make this clearer.  In the CIFAR10 classification setting, we demonstrate that heterogeneous ensembling smooths out the double-descent curve at zero regularization, as in a classification task percentage error is naturally bounded.
>
> We have since updated our calculations for proposition 2 to consider overlapping partitions of features.
>
> ## On Using the Replica Trick
>
>  We do in the remarks made directly after the statement of our main result in the main text state that the replica trick is a “non-rigorous but standard” approach.  We make no claim that our results are fully rigorous, but because the results obtained from the replica trick have been demonstrated to coincide with rigorous results for many similar problems and our analytical results show excellent agreement with numerical simulations, we have good reason to expect that our result is correct. However, In light of the concerns raised here we see it fit to change the statement of our main result to further emphasize that the replica trick is not rigorous, and to establish a rigorous basis for our main result.
>
> In the case where the data covariance matrix has a bounded spectrum, we believe  that our results may be obtained through a clever special case of the rigorous result of Loureiro [2022].  While their derivation does not explicitly consider ensembles with variation in the number of features viewed by each ensemble member, this type of heterogeneity may be added in post-hoc by choosing data covariance matrices which “zero out” a number of features which varies over the ensemble.  If accepted, we will add a supplemental section discussing this correspondence and including a detailed derivation of our result from the general result of Loureiro [2022].  We will also update the “proof” of the main theorem to the following:
>
> “We calculate the terms in the generalization error using the replica trick,  a standard but non-rigorous method from the statistical physics of disordered systems.  The full derivation may be found in the SI.  In the special case where the covariance matrices $\Sigma_s, \Sigma_0$” have bounded spectrum, this result may be obtained as a clever special case of the results of Loureiro [2022] (see SI for derivation).”
>
> We will also add the following remark after the proof of proposition 2:
> “Note that, as in this case $\Sigma_s$ does not have a bounded spectrum, this result does not follow from the rigorous results of Loureiro [2022].  However, we find excellent agreement between theory and experiment when data dimension is sufficiently large.”
>
> We emphasize that, even as our general result may be recovered as a special case of the results of Loureiro [2022], simplifying this general result in the special case of subsampling from globally correlated features is a very tedious calculation which requires significant work.  Our investigation of this general result also differs entirely from Loureiro [2022], which considered only ensembles with the same number of readout weights across ensemble members, and included no study of feature noise or readout noise.

---

> > ### Comment · Reviewer_tCNd · 2023-08-15
> > **Requesting further justification of readout error**
> >
> > I appreciate the authors' detailed response and revision proposals. In general, I like the revisions that you have suggested, and am inclined to increase my score. In particular, I think it would be very good to add the proof via special case of Loureiro [2022] for bounded operators. However, I am still not satisfied with the justification of readout noise, which seems to be a significant part of the settings in which these ensembles are worth using (e.g., Figure 4).
> >
> > The authors have given justification of readout noise from dropout or biological neural networks, which is somewhat reasonable in my opinion if there is no feature subsampling (i.e., all members of the ensemble use the same features). However, I struggle to imagine a concrete setting in which feature subsampling and readout noise would occur together. Do the authors have a better example of where I could expect to see both together, where a bunch of models with readout noise would be trained on different features? The artificial and biological neural network justification seems weak here, since such models are rarely trained on subsets of features, not to mention they are not linear.

---

> > > ### Author Response · Authors · 2023-08-20
> > >
> > > Thank you for taking the time to read and consider our rebuttal. One concrete setting in which feature subsampling and readout noise necessarily co-exist is in biological neural circuits for recognition of visual stimuli. We clarify that in this setting, we conceptualize the features as representations in the visual cortex (not as the features of the raw data) from which noisy downstream neurons can access only a subset of the millions of relevant “feature neurons.”  In this case, each member of the ensemble of readouts would sample from a different subset of the available features.
> > >
> > > Similarly, in artificial neural networks, we may conceptualize features not as pixels of the raw inputs, but as the dimensions of a feature map, from which readout weights are learned during fine-tuning. This is the setting that we explore in figure S1, where feature subsampling ensembles are trained on the neurons of the top hidden layer of a pre-trained deep network. Here there is no readout noise, but we still see a benefit to the feature-subsampling ensemble with heterogeneous input dimensionality in the absence of regularization, though we agree that this approach isn’t necessarily worth using when computational resources permit a hyper-parameter search for the optimal regularization.  We will make this clear by adding a direct comparison with predictions from readouts trained with “optimal regularization” in figure 3c and figure S1.
> > >
> > > Readout noise and feature subsampling can also coexist in any physical neural network with sparse connectivity. While feature subsampling may not currently be widely employed in this context, our results suggest that it should be used for any neural network which is “noise-dominated”— a notion we have made precise the special case of equicorrelated features (see global rebuttal).

---

> > > > ### Comment · Reviewer_tCNd · 2023-08-21
> > > >
> > > > I appreciate the authors' response and example of physical neural networks as a justification for readout noise---to me, this is the best justification. I do think that since the authors rely on this example to motivate their theoretical model, it would be ideal to have a citation to the biological literature that indicates that we know that such structures actually exist that correspond to this model, otherwise the model is purely hypothetical with no application. I will still raise my score by 1 point since I think the analysis here is novel and the authors have addressed many concerns.

---

### Author Rebuttal · Authors · 2023-08-02

Thank you for your insightful reviews.  Please find below a description of updates to the paper and responses to comments which were raised in multiple reviews.

-We have changed the definition of the “readout noise” so that it is present both during training and evaluation of the model.  This leads to a minor change in the form of the generalization error. Now, the contribution due to the readout noise goes as $\langle E_{rr’} \rangle \sim \frac{\delta_{rr’} \eta_r^2}{1-\gamma_{rr’}}$

-We have changed “globally correlated” to “equicorrelated” to agree with previous literature.

-We have updated proposition 2 to be more general, including an overall scale of the data, isotropic feature noise, and allowing for overlap in the features sampled by different readouts in the ensemble.  In the updated statement, we simplify the general expression for the generalization error from proposition 1 under the following special case:
>$$ \mathbf{w}^* = \sqrt{1-\rho^2} \mathbb{P}_{\perp} \mathbf{w}^*_0 + \rho\mathbf{1}_M $$
$$ \mathbf{w}^*_0 \sim \mathcal{N}(0, \mathbf{I}_M) $$
$$   \mathbf{\Sigma}_s = s \left[(1-c) \mathbf{I}_M + c \mathbf{1}_M \mathbf{1}_M^\top \right] $$
$$ \mathbf{\Sigma}_0 = \omega \mathbf{I}_M $$

-To enhance readability, we will use the extra space to include a table of parameters relevant to proposition 2 and figures 2, 3, and 4.  This table is included in the PDF attachment.

-Figure 4 has been updated (see attached PDF) and the surrounding discussion will be updated as follows:

>In the ridgeless limit, we can then express the error as : $E_g(k) = s(1-c)F(H, k, \rho, \alpha)$, where
$H \equiv \frac{\eta^2}{s(1-c)}$ is an effective inverse signal-to-noise ratio and
$F(H, k, \rho, \alpha)$ is a rational function of its arguments (see SI for full expressions).  Thus the value
$k^*$ which minimizes error depends on
$\eta$,
$s$, and
$c$ only through the ratio
$H$.
>Using our analytical theory, we plot the optimal number of readouts
$k$ in the parameter space of
$H$ and
$\rho$ (see Fig. 4a).  The resulting phase diagrams are naturally divided into three regions.  In the signal-dominated phase a single fully-connected readout is optimal ($k^* = 1$).  In an intermediate phase, $1<k^*<\infty$ minimizes error.  And in a noise-dominated phase $k^* = \infty$...  As is evident in these phase diagrams, an increase in H (decrease in SNR or increase in c) or an increase in $\rho$ causes an increase in $k^*$.

-We will clarify our motivation for considering a “readout noise” by adding the following paragraph to the introduction:
>“Subsampling from different sets of features introduces variance between members of an ensemble.  However, other sources of variance may also distinguish members of an ensemble of predictors.  For example, random initialization or weight dropout during training may introduce variance in an ensemble of deep networks trained with gradient descent.  Physical neural networks may also have intrinsically noisy neurons which introduce variance between ensemble members. To capture these effects which are not naturally present in the regression setting, we introduce an  explicit “readout noise” which is drawn independently for each ensemble member.   We provide a detailed analysis of the interplay between feature subsampling ensembles and readout noise in the special case of equicorrelated features.”
We will add the following remark after the statement of our main result:
“Remark 4:  “Readout noise” is a noise which is added to the prediction of each member of the ensemble of readouts before they are averaged.  The readout noise parameter $\eta$ represents all additional sources of variance across predictors not accounted for in our simple model.  For example, in an ensemble of deep neural networks trained with gradient descent, there will be additional variance across predictors due to random initialization of the networks (Atanasov 2023).  Additional variance may also be present in physical neural networks, such as an analog neural network or biological neural circuits due to inherent stochasticity in the physical mechanisms of computation.”
And by adding the following sentence the discussion of Figure 4 in section 2.6:
“The resulting phase diagrams demonstrate that partitioning data features amongst multiple readouts can be beneficial when there are sources of variance between ensemble members beyond that induced by sampling from separate sets of features.”

-On the need for real data and model experiments – we have included in the supplementary material an application of heterogeneous ensembles to the CIFAR10 classification task.  While the classification setting does not correspond to the regression setting of our main theorem, the qualitative behavior carries over to this real-world task – we observe that heterogeneity in the number of features seen by each member of the readout ensemble mitigates double-descent even without regularization.  We plan to significantly expand this portion of the paper to demonstrate the benefits of heterogeneous ensembling for image classification with pre-trained feature maps (for example,  a pre-trained ResNet).

-A single fully connected readout with optimal L2 regularization does perform better than a feature subsampling ensemble. However, training a feature-subsampling ensemble is much faster than training a single fully-connected readout layer. (Assuming cubic scaling of the matrix inverse operation, training $k$ models of size $M/k$ scales as  $1/k^2$). Further, introducing heterogeneity of the type we study into the ensemble gives the added benefit of avoiding catastrophic over-fitting without a task-tuned regularization.  This approach is most likely to find applications in settings where training time or computational resources are limited.  If accepted, we will add these points to our discussion.  We will also add curves to panels c.i-vi which show the learning curves for a single model with optimal regularization.

---

### Decision · Program_Chairs · 2023-09-21

**Decision:**

Accept (poster)

**Comment:**

The paper analyses the asymptotic properties of ensembles of ridge regressors trained from heterogeneous (in terms of content and size) feature subsamples. The paper combines solid theory and some interesting preliminary experiments that support this theory. The reviewers highlighted several limitations of the work (justification of the data model and presentation problems) that were partially addressed by the rebuttals. Despite these limitations, the reviewers and I agree that the work is novel, can certainly generate interesting discussions and lead to further theoretical or more practical developments in the future. I therefore recommend acceptance. The authors are urged to make all promised changes in the final version of their paper.